# Towards Principled Representation Learning from Videos for Reinforcement Learning

**Dipendra Misra**[1*]    **Akanksha Saran**[2*]    **Tengyang Xie**[1]    **Alex Lamb**[1]    **John Langford**[1]
[1]Microsoft Research, NY    [2]Sony Research, CA

## Abstract

We study pre-training representations for decision-making using video data, which is abundantly available for tasks such as game agents and software testing. Even though significant empirical advances have been made on this problem, a theoretical understanding remains absent. We initiate the theoretical investigation into principled approaches for representation learning and focus on learning the latent state representations of the underlying MDP using video data. We study two types of settings: one where there is iid noise in the observation, and a more challenging setting where there is also the presence of exogenous noise, which is non-iid noise that is temporally correlated, such as the motion of people or cars in the background. We study three commonly used approaches: autoencoding, temporal contrastive learning, and forward modeling. We prove upper bounds for temporal contrastive learning and forward modeling in the presence of only iid noise. We show that these approaches can learn the latent state and use it to do efficient downstream RL with polynomial sample complexity. When exogenous noise is also present, we establish a lower bound result showing that the sample complexity of learning from video data can be exponentially worse than learning from action-labeled trajectory data. This partially explains why reinforcement learning with video pre-training is hard. We evaluate these representational learning methods in two visual domains, yielding results that are consistent with our theoretical findings.

## 1 Introduction

Representations pre-trained on large amounts of offline data have led to significant advances in machine learning domains such as natural language processing (Liu et al., 2019; Brown et al., 2020) and multi-modal learning (Lin et al., 2021; Radford et al., 2021). This has naturally prompted a similar undertaking in reinforcement learning (RL) with the goal of training a representation model that can be used in a policy to solve a downstream RL task. The natural choice of data for RL problems is trajectory data, which contains the agent's observation along with actions taken by the agent and the rewards received by it (Sutton & Barto, 2018). A line of work has proposed approaches for learning representations with trajectory data in both offline (Uehara et al., 2021; Islam et al., 2022) and online learning settings (Nachum et al., 2018; Bharadhwaj et al., 2022). However, unlike text and image data, which are abundant on the internet or naturally generated by users, trajectory data is comparatively limited and expensive to collect. In contrast, video data, which only contains a sequence of observations (without any action or reward labeling), is often plentiful, especially for domains such as gaming and software. This motivates a line of work considering learning representations for RL using video data (Zhao et al., 2022). *But is there a principled foundation underlying these approaches? Are representations learned from video data as useful as representations learned from trajectory data?* We initiate a theoretical understanding of these approaches to show when and how these approaches yield representations that can be used to solve a downstream RL task efficiently.

Consider a representation learning pipeline shown in Figure 1. We are provided videos, or equivalently a sequence of observations, from agents navigating in the world. We make no assumption

---

*DM and AS contributed equally. Correspondence should be sent to `dimisra@microsoft.com` and `akanksha.saran@sony.com`.

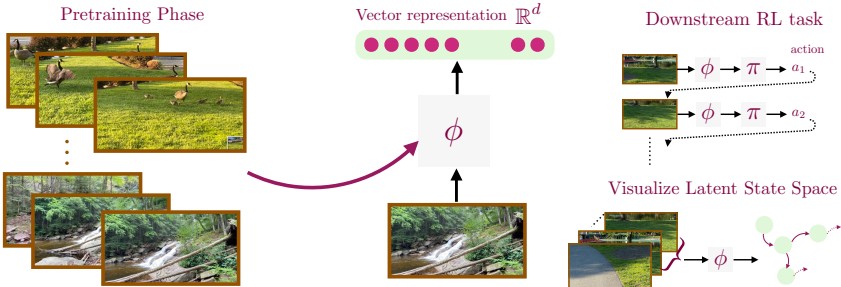

Figure 1: A flowchart of our video pre-training phase. **Left:** We assume access to a large set of videos (or, unlabeled episodes). **Center:** A representation learning method is used to train a model $\phi$ which maps an observation to a vector representation. **Right:** This representation can be used in a downstream task to do reinforcement learning or visualize the latent world state.

about the behavior of the agent in the video data. They can be trying to solve one task, many different tasks, or none at all. This video data is used to learn a model $\phi$ that maps any given observation to a vector representation. This representation is subsequently used to perform downstream RL — defining a policy on top of the learned representation and only training the policy for the downstream task. We can also use this representation to define a dynamics model or a critique model. The representation can also help visualize the agent state space or dynamics for the purpose of debugging.

A suitable representation for performing RL efficiently is aligned with the underlying dynamics of the world. Ideally, the representation captures the latent agent state, which contains information about the world relevant to decision-making while ignoring any noise in the observation. For example, in Figure 1, ignoring noise such as the motion of geese in the background is desirable if the task involves walking on the pavement. We distinguish between two types of noise: (1) temporally independent noise that occurs at each time step independent of the history, (2) temporally dependent noise, or exogenous noise, that can evolve temporally but in a manner independent of the agent's actions (such as the motion of geese in Figure 1).

A range of approaches have been developed that provably recover the latent agent state from observations using trajectory data (Misra et al., 2020; Efroni et al., 2022) which contains actions. However, for many domains there is relatively little trajectory data that exists naturally, making it expensive to scale these learning approaches. In contrast, video data is more naturally available but these prior provable approaches do not work with video data. On the other hand, it is unknown whether approaches that empirically work with video data provably recover the latent representation and lead to efficient RL. Motivated by this, we build a theoretical understanding of three such video-based representation learning approaches: *autoencoder* which trains representations by reconstructing observations, *forward modeling* which predicts future observations, and *temporal contrastive* learning which trains a representation to determine if a pair of observations are causally related or not.

Our first theoretical result shows that in the absence of exogenous noise, forward modeling and temporal contrastive learning approaches both provably work. Further, they lead to efficient downstream RL that is strictly more sample-efficient than solving these tasks without any pre-training. Our second theoretical result establishes a lower bound showing that in the presence of exogenous noise, any compact and frozen representation that is pre-trained using video data cannot be used to do efficient downstream RL. In contrast, if the trajectory data was available, efficient pre-training would be possible. This establishes a statistical gap showing that video-based representation pre-training can be exponentially harder than trajectory-based representation pre-training.

We empirically test our theoretical results in three visual domains: GridWorld (a navigation domain), ViZDoom basic (a first-person 3D shooting game), and ViZDoom Defend The Center (a more challenging first-person 3D shooting game). We evaluate the aforementioned approaches along with ACRO (Islam et al., 2022), a representation pre-trained using trajectory data and designed to filter out exogenous noise. We observe that in accordance with our theory, both forward modeling and temporal contrastive learning succeed at RL when there is no exogenous noise. However, in the presence of exogenous noise, their performance degrades. Specifically, we find that temporal contrastive learning is especially prone to fail in the presence of exogenous noise, as it can rely exclusively on such noise to optimally minimize the contrastive loss. While we find that

forward modeling is somewhat robust to exogenous noise, however, as exogenous noise increases, its performance quickly degrades as well. While any finite-sample guarantees for the autoencoding method remain an open question, empirically, we find that the performance of autoencoder-based representation learning is unpredictable. On the other hand, ACRO continues to perform well, highlighting a disadvantage of video pre-training. The code for all experiments is available as part of the Intrepid codebase at `https://github.com/microsoft/Intrepid`.

## 2    PRELIMINARIES AND OVERVIEW

In this section, we provide a formal overview of our learning setup and problem statement.

**Mathematical Notation.** We use $[N]$ for $N \in \mathbb{N}$ to define the set $\{1, 2, \cdots, N\}$. We assume all sets to be countable. For a given set $\mathcal{U}$, we denote its cardinality by $|\mathcal{U}|$ and define $\Delta(\mathcal{U})$ as the space of all distributions over $\mathcal{U}$. We denote the uniform distribution over $\mathcal{U}$ by $\texttt{Unf}(\mathcal{U})$. Finally, $\texttt{poly}\{\cdot\}$ denotes a term that scales polynomially in the listed quantities.

**Block MDPs.** We study episodic RL in Block Markov Decision Processes (Block MDP) (Du et al., 2019). A Block MDP is defined by the tuple $(\mathcal{X}, \mathcal{S}, \mathcal{A}, T, R, q, \mu, H)$ where $\mathcal{X}$ is a set of observations that can be infinitely large, $\mathcal{S}$ is a finite set of *latent* states, and $\mathcal{A}$ is a set of finite actions. The transition dynamics $T : \mathcal{S} \times \mathcal{A} \rightarrow \Delta(\mathcal{S})$ define transitions in the latent state space. The reward function $R : \mathcal{S} \times \mathcal{A} \rightarrow [0, 1]$ assigns a reward $R(s, a)$ if action $a$ is taken in the latent state $s$. When the agent visits a state $s$, it receives an observation $x \sim q(\cdot \mid s)$ sampled from an emission function $q : \mathcal{S} \rightarrow \Delta(\mathcal{X})$. This emission process contains temporally independent noise but no exogenous noise. Finally, $\mu \in \Delta(\mathcal{S})$ is the distribution over the initial latent state and $H$ is the horizon denoting the number of actions per episode. The agent interacts with a block MDP environment by repeatedly generating an episode $(x_1, a_1, r_1, \cdots, x_H, a_H, r_H)$ where $s_1 \sim \mu$ and for all $h \in [H]$ we have $x_h \sim q(\cdot \mid s_h), r_h = R(s_h, a_h)$, and $s_{h+1} \sim T(\cdot \mid s_h, a_h)$, and all actions $\{a_h\}_{h=1}^{H}$ are taken by the agent. The agent never directly observes the latent states $(s_1, s_2, \cdots, s_H)$.

A key assumption in Block MDPs is that two different latent states cannot generate the same observation. This is called the *disjoint emission property* and holds in many game and OS settings. Formally, this property allows us to define a decoder $\phi^\star : \mathcal{X} \rightarrow \mathcal{S}$ that maps an observation to the unique state that can generate it. The agent does not have access to $\phi^\star$. If the agent had access to $\phi^\star$, one could map each observation from an infinitely large space to the finite latent state space, which allows the use of classical finite RL methods (Kearns & Singh, 2002).

**Exogenous Block MDPs (Ex-Block MDP).** We also consider RL in Exogenous Block MDPs (Ex-Block MDPs) that extend Block MDPs to include exogenous noise (Efroni et al., 2022). An Ex-Block MDP is defined by $(\mathcal{X}, \mathcal{S}, \Xi, \mathcal{A}, T, T_\xi, R, q, H, \mu, \mu_\xi)$ where $\mathcal{X}, \mathcal{S}, \mathcal{A}, T, R, H$ and $\mu$ have the same meaning and type as in Block MDPs. The additional quantities include $\Xi$ which is the space of exogenous noise and can be infinitely large. We use the notation $\xi \in \Xi$ to denote the exogenous noise. For the setting in Figure 1, the exogenous noise variable $\xi$ captures variables such as the position of geese, the position of leaves on the trees in the background, and lighting conditions. The exogenous noise $\xi$ changes with time according to the transition function $T_\xi : \Xi \rightarrow \Delta(\Xi)$ and is at start sampled from $\mu_\xi$. Note that unlike the agent state $s \in \mathcal{S}$, the exogenous noise $\xi \in \Xi$, evolves independently of the agent's action and does not influence the evolution of the agent's state. The emission process $q : \mathcal{S} \times \Xi \rightarrow \Delta(\mathcal{X})$ in Ex-Block MDP uses both the current agent state and exogenous noise, to generate the observation at a given time. For example, the image generated by the agent's camera contains information based on the agent's state (e.g., agent's position and orientation), along with exogenous noise (e.g., the position of geese). Similar to the Block MDP, we assume there exists *unknown* decoders $\phi^\star : \mathcal{X} \rightarrow \mathcal{S}$ and $\phi_\xi^\star : \mathcal{X} \rightarrow \xi$ that can map an observation to the current agent state $s$ and exogenous $\xi$ respectively.

**Provable RL.** We assume access to a policy class $\Pi = \{\pi : \mathcal{X} \rightarrow \mathcal{A}\}$ where a policy $\pi \in \Pi$ allows the agent to take actions. For a given policy $\pi$, we use $\mathbb{E}_\pi[\cdot]$ to denote expectation taken over an episode generated by sampling actions from $\pi$. We define the value of a policy $V(\pi) = \mathbb{E}_\pi\left[\sum_{h=1}^{H} r_h\right]$ as the expected total reward or expected return. Our goal is to learn a near-optimal policy $\widehat{\pi}$, i.e., $\sup_{\pi \in \Pi} V(\pi) - V(\widehat{\pi}) \leq \varepsilon$ with probability at least $1 - \delta$ for a given tolerance parameter $\varepsilon > 0$ and failure probability $\delta \in (0, 1)$, using number of episodes that scale polynomially in $1/\varepsilon$, $1/\delta$, and other relevant quantities. We will call such an algorithm as provably efficient. There exist

several provably efficient RL approaches for solving Block MDPs (Mhammedi et al., 2023; Misra et al., 2020), and Ex-Block MDPs (Efroni et al., 2022). These approaches typically assume access to a decoder class $\Phi = \{\phi : \mathcal{X} \to [N]\}$ and attempt to learn $\phi^\star$ using it. These algorithms don't use any pre-training and instead directly interact with the environment and learn a near-optimal policy by using samples that scale with $\texttt{poly}(S, A, H, \ln |\Phi|, 1/\varepsilon, 1/\delta)$. Crucially, the dependence on $\ln |\Phi|$ cannot be removed. The decoder class $\Phi$ and all other function classes in this work are assumed to have bounded statistical complexity measures. For simplicity, we will assume that these function classes are finite and derive guarantees that scale logarithmically in their size (e.g., $\ln |\Pi|$).[1]

**Representation Pre-training using Videos.** RL algorithms for the above settings require online episodes that scale with $\ln |\Phi|$ which is expensive for real-world problems where $\Phi$ is represented by a complex neural network. Offline RL approaches Uehara et al. (2021) offer a substitute for expensive online interactions but require access to labeled episodes (with actions and rewards) that are not naturally available in many settings such as games and software. In contrast, we focus on pre-training the decoder $\phi$ using video data which is naturally available in these settings.

**Problem Statement.** We are given two hyperparameters $\varepsilon > 0$ and $\delta \in (0, 1)$ and a sufficiently large dataset of videos. We are also given a decoder class $\Phi = \{\phi : \mathcal{X} \to [N]\}$ containing decoders that map an observation to one of the $N$ possible *abstract states*. During the pre-training phase, we learn a decoder $\phi \in \Phi$ using the video data. We then freeze $\phi$ and use it to do RL in a downstream task. Instead of using any particular choice of algorithm for RL, we assume we are given a provably efficient tabular RL algorithm $\mathscr{A}$. We convert the observation-based RL problem to a tabular MDP problem by converting an observation $x$ to its abstract state representation $\phi(x)$ using the frozen learned decoder $\phi$. The algorithm $\mathscr{A}$ uses $\phi(x)$ instead of $x$ and outputs an *abstract policy* $\varphi : [N] \to \mathcal{A}$. We want that $\sup_{\pi \in \Pi} V(\pi) - V(\varphi \circ \phi) \leq \varepsilon$ with probability at least $1 - \delta$, where $\varphi \circ \phi : x \mapsto \varphi(\phi(x))$ is our learned policy. We also require the number of online episodes in the downstream RL phase to not scale with the size of the decoder class $\Phi$. This allows us to minimize expensive online episodes while using naturally available offline video data for pre-training.

## 3 REPRESENTATION LEARNING FOR RL USING VIDEO DATASET

We assume access to a dataset $\mathcal{D}$ of $n$ videos $\mathcal{D} = \{(x_1^{(i)}, x_2^{(i)}, \cdots, x_H^{(i)})\}_{i=1}^n$ where $x_j^{(i)}$ is the $j^{th}$ observation (or frame) of the $i^{th}$ video. We are provided a decoder class $\Phi = \{\phi : \mathcal{X} \to [N]\}$, and our goal is to learn a decoder $\phi \in \Phi$ that captures task-relevant information in the underlying state $\phi^\star(x)$ while throwing away as much exogenous noise as possible. Instead of proposing a new algorithm, we analyze the following three classes of well-known video-based representation learning methods. Our goal is to understand whether these methods provably learn useful representations.

**Autoencoder.** This approach first maps a given observation $x$ to an abstract state $\phi(x)$ using a decoder $\phi \in \Phi$, and then uses it to reconstruct the observation $x$ with the aid of a reconstruction model class $\mathcal{Z} = \{z : [N] \to \mathcal{X}\}$. Formally, we optimize the following loss:

$$\ell_{\text{auto}}(z, \phi) = \frac{1}{nH} \sum_{i=1}^n \sum_{h=1}^H \|z(\phi(x_h^{(i)})) - x_h^{(i)}\|_2^2. \tag{1}$$

In practice, autoencoders are typically implemented using a Vector Quantized bottleneck trained in a Variational AutoEncoder manner, which is called the VQ-VAE approach (Oord et al., 2017).

**Forward Modeling.** This approach is similar to the autoencoder approach but instead of reconstructing the input observation, we reconstruct a future observation using a model class $\mathcal{F} = \{f : [N] \times [K] \to \Delta(\mathcal{X})\}$ where $N$ is the output size of the decoder class $\Phi$ and $K \in \mathbb{N}$ is a hyperparameter representing the forward time steps from the current observation. We collect a dataset of *multistep transitions* $\mathcal{D}_{\text{for}} = \{(x^{(i)}, k^{(i)}, x'^{(i)})\}_{i=1}^n$ sampled iid using the video dataset $\mathcal{D}$ where the observation $x^{(i)}$ is sampled randomly from the $i^{th}$ video, $k^{(i)} \in [K]$, and $x'^{(i)}$ is the frame $k^{(i)}$-steps ahead of $x^{(i)}$ in the $i^{th}$ video. We distinguish between two types of sampling procedures, one where $k^{(i)}$ is always a fixed given value $k \in [K]$, and one where $k^{(i)} \sim \text{Unf}([K])$. Given the dataset $\mathcal{D}_{\text{for}}$, we optimize the following loss:

$$\ell_{\text{for}}(f, \phi) = \frac{1}{n} \sum_{i=1}^n \ln f\left(x'^{(i)} \mid \phi(x^{(i)}), k^{(i)}\right). \tag{2}$$

---

[1]Our theoretical analyses can be extended to other complexity metrics such as Rademacher complexity.

**Temporal Contrastive Learning.** Finally, this approach trains the decoder $\phi$ to learn to separate a pair of temporally causal observations from a pair of temporally *acausal* observations. We collect a dataset of $\mathcal{D}_{\text{temp}} = \{(x^{(i)}, k^{(i)}, x'^{(i)}, z^{(i)})\}_{i=1}^{\lfloor n/2 \rfloor}$ tuples using the multistep transitions dataset $\mathcal{D}_{\text{for}}$. We use 2 multistep transitions to create a single datapoint for $\mathcal{D}_{\text{temp}}$ to keep the datapoints independent. To create the $i^{th}$ datapoint for $\mathcal{D}_{\text{temp}}$, we use the multistep transitions $(x^{(2i)}, k^{(2i)}, x'^{(2i)})$ and $(x^{(2i+1)}, k^{(2i+1)}, x'^{(2i+1)})$ and sample $z^{(i)} \sim \text{Unf}(\{0, 1\})$. If $z^{(i)} = 1$, then our $i^{th}$ datapoint is a causal observation pair $(x^{(2i)}, k^{(2i)}, x'^{(2i)}, z^{(i)})$, otherwise, it is an acausal observation pair $(x^{(2i)}, k^{(2i)}, x'^{(2i+1)}, z^{(i)})$. Depending on how we sample $k$, we collect a different dataset $\mathcal{D}_{\text{for}}$, and accordingly a different dataset $\mathcal{D}_{\text{temp}}$. Given the dataset $\mathcal{D}_{\text{temp}}$, we optimize the following loss using a regression model $g$ belonging to a model class $\mathcal{G} = \{g : \mathcal{X} \times [K] \times \mathcal{X} \to [0, 1]\}$:

$$\ell_{\text{temp}}(g, \phi) = \frac{1}{\lfloor n/2 \rfloor} \sum_{i=1}^{\lfloor n/2 \rfloor} \left(z^{(i)} - g(\phi(x^{(i)}), k^{(i)}, x'^{(i)})\right)^2. \tag{3}$$

**Practical Implementations.** We use the aforementioned description of methods for theoretical analysis. However, their practical implementations differ in a few notable ways. Most importantly we either use a continuous vector representation $\phi : \mathcal{X} \to \mathbb{R}^d$ for modeling $\Phi$, or apply a Vector Quantized (VQ) bottleneck (Oord et al., 2017) on top of the vector representation to model a discrete-representation decoder. We also optimize the loss using minibatches and use square loss for training forward modeling and SimCLR loss (Chen et al., 2020) for contrastive learning. We experimentally show that our theoretical findings extend to these practical implementations.

# 4 IS VIDEO BASED REPRESENTATION LEARNING PROVABLY CORRECT?

In this section, we present our main theoretical results. We first prove that both forward modeling and temporal contrastive methods succeed when there is no exogenous noise. We then establish a lower bound showing that video-based representation learning is exponentially harder than trajectory-based representation learning. We defer all proofs to the Appendix and only provide a sketch here.

## 4.1 UPPER BOUND IN BLOCK MDP SETTING

We start by stating our theoretical setting and our main assumptions.

**Theoretical Setting.** We assume a Block MDP setting and access to a dataset $\mathcal{D} = \left\{(x_1^{(i)}, x_2^{(i)}, \cdots, x_H^{(i)})\right\}_{i=1}^n$ of $n$ independent and identically distributed (iid) videos sampled from data distribution $D$. We denote the probability of a video as $D(x_1, x_2, \cdots, x_H)$. We assume that $D$ is generated by a mixture of Markovian policies $\Pi_D$, i.e., the generative procedure for $D$ is to sample a policy $\pi \in \Pi_D$ with some probability and then generate an entire episode using it. We assume that observations encode time steps. This can be trivially accomplished by simply concatenating the time step information to the observation. We also assume that the video data has good state space coverage and that the data is collected by *noise-free policies.*

**Assumption 1** (Requirements on Data Collection)**.** *There exists an $\eta_{min} > 0$ such that if $s$ is a state reachable at time step $h$ by some policy in $\Pi$, then $D\left(\phi^\star(x_h) = s\right) \geq \eta_{min}$. Further, we assume that every data collection policy $\pi \in \Pi_D$ is noise-free, i.e., $\pi(a \mid x_h) = \pi(a \mid \phi^\star(x_h))$ for all $(a, x_h)$.*

**Justification for Assumption 1** In practice, we expect this assumption to hold for tasks such as gaming, or software debugging, where video data is abundant and, therefore, can be expected to provide good coverage of the underlying state space. This assumption is far weaker than the assumption in batch RL which also requires actions and rewards to be labeled, which makes it more expensive to collect data that has good coverage (Chen & Jiang, 2019). Further, unlike imitation learning from observations (ILO) (Torabi et al., 2019), we don't require that these videos provide demonstrations of the desired behavior. E.g., video streaming of games is extremely common on the internet, and one can get many hours of video data this way. However, this data wouldn't come with actions (which will be mouse or keyboard strokes) or reward labeling, and the game levels or tasks in the data can be different or even unrelated to the downstream tasks we want to solve. As such, the video data do not provide demonstrations of the desired task. Further, as the video data is typically generated by humans, we can expect the data collection policies to be noise-free, as these policies are realized by

humans who would not make decisions based on noise. E.g., a human player is unlikely to turn left due to the background motion of leaves that is unrelated to the game's control or objective.

We analyze the temporal contrastive learning and forward modeling approaches and derive upper bounds for these methods in Block MDPs. While autoencoder-based approaches sometimes do well in practice, it is an open question whether finite-sample bounds exist for them and we leave their theoretical analysis to future work and instead evaluate them empirically. In addition to the decoder class $\Phi$, we assume a function class $\mathcal{F}$ to model $f$ for forward modeling and $\mathcal{G}$ to model $g$ for temporal contrastive learning. We make a realizability assumption on these function classes.

**Assumption 2** (Realizability). *There exists $f^\star \in \mathcal{F}, g^\star \in \mathcal{G}$ and $\phi_{for}, \phi_{temp} \in \Phi$ such that $f^\star(X' \mid \phi_{for}(x), k) = \mathbb{P}_{for}(X' \mid x, k)$ and $g^\star(z \mid \phi_{temp}(x), k, x') = \mathbb{P}_{temp}(z = 1 \mid x, k, x')$ on the appropriate support, and where $\mathbb{P}_{for}$ and $\mathbb{P}_{temp}$ are respectively the Bayes classifier for the forward modeling and temporal contrastive learning methods.*

**Justification for Assumption 2.** Realizability is a typical assumption made in theoretical analysis of RL algorithms (Agarwal et al., 2020). Intuitively, the assumption states that the function classes are expressive enough to represent the Bayes classifier of their problem. In practice, this is usually not a concern as we will use expressive deep neural networks to model these function classes. We will empirically show the feasibility of this assumption in our experiments.

Finally, we assume that our data distribution has the required information to separate the latent states. We state this assumption formally below and then show settings where this is true.

**Assumption 3** (Margin Assumption). *We assume that the margins $\beta_{for}$ and $\beta_{temp}$ defined below:*

$$\beta_{for} = \inf_{s_1, s_2 \in \mathcal{S}, s_1 \neq s_2} \mathbb{E}_k \left[ \|\mathbb{P}_{for}(X' \mid s_1, k) - \mathbb{P}_{for}(X' \mid s_2, k)\|_{TV} \right]$$

$$\beta_{temp} = \inf_{s_1, s_2 \in \mathcal{S}, s_1 \neq s_2} \frac{1}{2} \mathbb{E}_{k, s'} \left[ |\mathbb{P}_{temp}(z = 1 \mid s_1, k, s') - \mathbb{P}_{temp}(z = 1 \mid s_2, k, s')| \right],$$

*are strictly positive, and where in the definition of $\beta_{temp}$, we sample $s'$ from the video data distribution and $k$ is sampled according to our data collection procedure.*

**Justification for Assumption 3.** This assumption states that we need margins ($\beta_{\text{for}}$) for forward modeling and ($\beta_{\text{temp}}$) for temporal contrastive learning. A common scenario where these assumptions are true is when for any pair of different states $s_1, s_2$, there is a third state $s'$ that is reachable from one but not the other. If the video data distribution $D$ supports all underlying transitions, then this immediately implies that $\|\mathbb{P}_{\text{for}}(X' \mid s_1, k) - \mathbb{P}_{\text{for}}(X' \mid s_2, k)\|_{\text{TV}} > 0$ which implies $\beta_{\text{for}} > 0$. This scenario occurs in almost all navigation tasks. Specifically, it occurs in the three domains we experiment with. While it is less clear, under this assumption we also have $\beta_{\text{temp}} > 0$.

We now state our main result for forward modeling under Assumption 1-3.

**Theorem 1** (Forward Modeling Result). *Fix $\varepsilon > 0$ and $\delta \in (0, 1)$ and let $\mathscr{A}$ be any provably efficient RL algorithm for tabular MDPs with sample complexity $n_{samp}(S, A, H, \varepsilon, \delta)$. If $n$ is $\texttt{poly}\{S, H, 1/\eta_{min}, 1/\beta_{for}, 1/\varepsilon, \ln(1/\delta), \ln|\mathcal{F}|, \ln|\Phi|\}$ for a suitable polynomial, then forward modeling learns a decoder $\widehat{\phi} : \mathcal{X} \to [|\mathcal{S}|]$. Further, running $\mathscr{A}$ on the tabular MDP with $n_{samp}(S, A, H, T, \varepsilon/2, \delta/4)$ episodes returns a latent policy $\widehat{\varphi}$. Then there exists a bijective mapping $\alpha : \mathcal{S} \to [|\mathcal{S}|]$ such that with probability at least $1 - \delta$ we have:*

$$\forall s \in \mathcal{S}, \qquad \mathbb{P}_{x \sim q(\cdot|s)} \left( \widehat{\phi}(x) = \alpha(s) \mid \phi^\star(x) = s \right) \geq 1 - \frac{4S^3 H^2}{\eta_{min}^2 \beta_{for}} \sqrt{\frac{1}{n} \ln\left( \frac{|\mathcal{F}| \cdot |\Phi|}{\delta} \right)},$$

*and the learned observation-based policy $\widehat{\varphi} \circ \widehat{\phi} : x \mapsto \widehat{\varphi}(\widehat{\phi}(x))$ is $\varepsilon$-optimal, i.e.,*

$$V(\pi^\star) - V(\widehat{\varphi} \circ \widehat{\phi}) \leq \varepsilon.$$

*Finally, the number of online episodes used in the downstream RL task is given by $n_{samp}(S, A, H, \varepsilon_\circ/2, \delta_\circ/4)$ and doesn't scale with the complexity of function classes $\Phi$ and $\mathcal{F}$.*

The result for temporal contrastive is identical to Theorem 1 but instead of $\beta_{\text{for}}$ we have $\beta_{\text{temp}}$ and instead of $\mathcal{F}$ we have $\mathcal{G}$. These upper bounds provide the desired result which shows that not only can we learn the right representation and near-optimal policy but also do so without online episodes scaling with $\ln|\Phi|$. Typically, the function class for forward modeling $\mathcal{F}$ is much more complex than $\mathcal{G}$, however, as we show in Appendix B.5, the margin for forward modeling $\beta_{\text{for}}$ is larger than for contrastive learning $\beta_{\text{temp}}$ leading to a trade-off between these two approaches.

### 4.2 LEARNING FROM VIDEO IS EXPONENTIALLY HARDER THAN LEARNING FROM TRAJECTORY DATA

When online RL is possible, there exist algorithms Misra et al. (2020); Efroni et al. (2022) that can learn an accurate latent state decoder $\widehat{\phi}$ with high probability and use it to learn near-optimal policies. These methods train the decoder using online trajectory data. This begs the following question: *Is it possible to learn a latent state decoder that is useful for performing RL using offline video data?* As the next result shows, this is not always the case.

**Theorem 2** (Lower Bound for Video). *Suppose $|\mathcal{S}|, |\mathcal{A}|, H \geq 2$. Then, for any $\varepsilon \in (0, 1)$, any algorithm $\mathscr{A}_1$ that outputs a state decoder $\phi$ with $\phi_h : \mathcal{X} \to [L]$, $L \leq 2^{1/4\varepsilon-1}$, $\forall h \in [H]$ given a video dataset $\mathcal{D}$ sampled from some MDP and satisfies Assumption 1, and any online RL algorithm $\mathscr{A}_2$ uses that state decoder $\phi$ in its interaction with such an MDP (i.e., $\mathscr{A}_2$ only observes states through $\phi$) and output a policy $\widehat{\pi}$, there exists an MDP instance $M$ in a class of MDPs which satisfies Assumption 3 and is PAC learnable with $\widetilde{O}(\texttt{poly}(|\mathcal{S}|, |\mathcal{A}|, H, 1/\varepsilon))$ complexity, such that*

$$V_M(\pi_M^\star) - V_M(\widehat{\pi}) > \varepsilon,$$

*regardless of the size of the video dataset $\mathcal{D}$ for algorithm $\mathscr{A}_1$ and the number of episodes of interaction for algorithm $\mathscr{A}_2$.*

The basic idea behind that hard instance construction is that, without the action information, it is impossible for the learning agent to distinguish between endogenous states and exogenous noise. For example, consider an image consisting of $N \times N$ identical mazes but where the agent controls just one maze. Other mazes contain other agents which are exogenous for our purpose. In the absence of actions, we cannot tell which maze is the one we are controlling and must memorize the configuration of all $N \times N$ mazes which grow exponentially with $N$. Another implication from that hard instance is – if the margin condition (Assumption 3) is violated, the exponentially large state decoder is also required for the regular block MDP without exogenous noise; a detailed discussion can also be found in Appendix B.3. We also discuss settings where we may be able to efficient-learning with just video data with additional assumptions in Appendix B.4.

## 5 EXPERIMENTAL RESULTS AND DISCUSSION

We empirically evaluate the above video-based representation learning methods on three visual environments: a gridworld environment and two VizDoom environments. We defer the results on one of the Vizdoom environments along with additional experimental details and results to Appendix C. Our main goal is to validate our theoretical findings by evaluating these methods in the presence and absence of exogenous noise and comparing their performance with a trajectory-based method.

### 5.1 EXPERIMENTAL DETAILS

**GridWorld.** We consider navigation in a $12 \times 12$ Minigrid environment (Chevalier-Boisvert et al., 2023). The agent (red triangle) can only observe an area around itself, and the goal is to reach the key quickly (Figure 3). The position of the agent and key randomizes each episode.

**ViZDoom Defend the Center** This is a first-person shooting game (Wydmuch et al., 2018; Kempka et al., 2016), in which the player needs to kill a variety of monsters to score (Figure 5). The episode ends when the monster is killed or after 500 steps.

**Exogenous Noise.** For all domains, the observation is an RGB image. We add exogenous noise to it by superimposing 10 generated diamonds of a particular size. The color and position of these diamonds are our exogenous state. At the start of each episode, we randomly generate these diamonds, after which they move in a deterministic path. We also test the setting in which there is exogenous noise in the reward. We compute a score based on just the exogenous noise and add it to the reward presented to the agent. However, the agent is still evaluated on the original reward.

**Model and Learning.** Our decoder class $\Phi$ is a convolutional neural network. We use a deconvolutional neural network to model $f$ and $h$. We experimented with both using a vector representation for $\phi$ and also using a VQ-bottleneck to discretize the embeddings. We use PPO to do downstream RL and keep $\phi$ frozen during the RL training. We also visualize the learned representations by

training a decoder on them and fixing $\phi$ to reconstruct the input observations. We then look at the generated images to see what information from the observation is preserved by the representation.

**ACRO.** We also evaluate the learned representations against ACRO (Islam et al., 2022) which uses trajectory data. This approach learns representation $\phi$ by predicting action given a pair of observations $\mathbb{E}\left[\ln p(a_h \mid \phi(x_h), x_{h+k}, k)\right]$. ACRO is designed to filter out exogenous noise as this information is not predictive of the action. Our goal is to test if we get much better representations if we have access to trajectory data instead of video data.

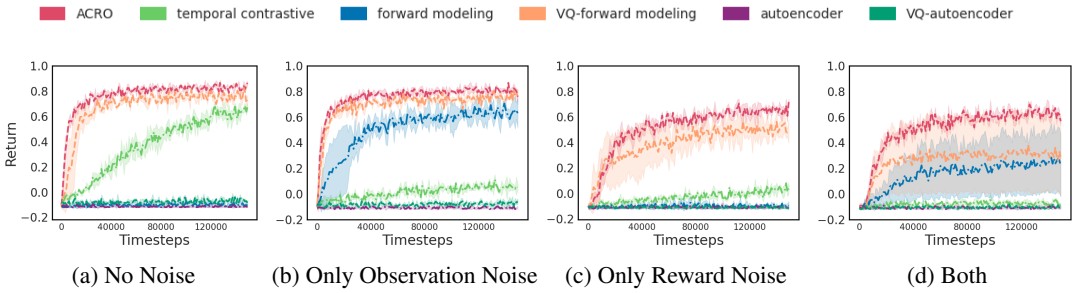

Figure 2: RL experiments in the GridWorld environment.

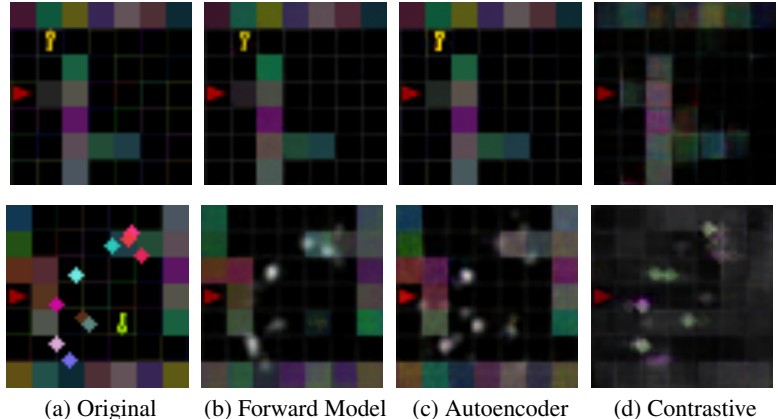

(a) Original  (b) Forward Model  (c) Autoencoder  (d) Contrastive

Figure 3: Decoded image reconstructions for different methods in the GridWorld environment. We train a reconstruction model on top of frozen learned representations $\phi$ trained with a given video-based method. **Top row:** shows an example from the setting where there is no exogenous noise. **Bottom row:** shows an example with exogenous noise (colored diamond shapes).

## 5.2 EMPIRICAL RESULTS AND DISCUSSION

We present our main empirical results in Figure 2 and Figure 4 and discuss the results below.

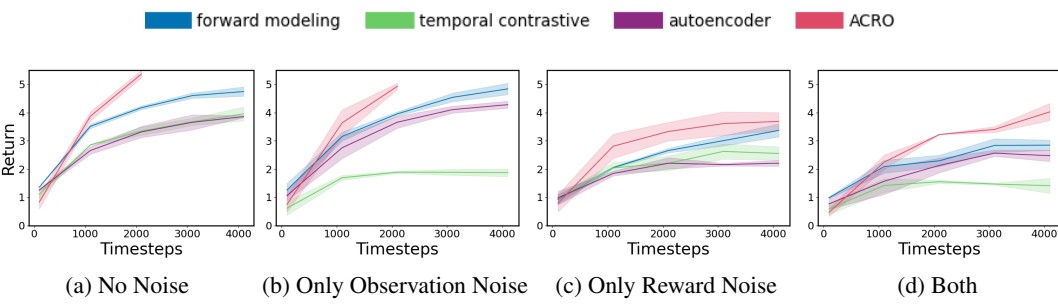

Figure 4: RL experiments using different latent representations for the ViZDoom Defend the Center environment.

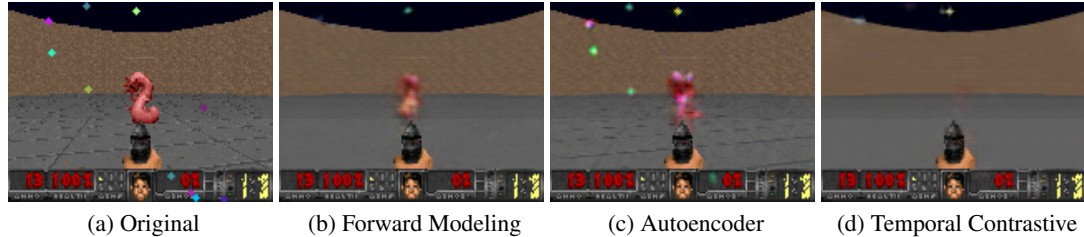

(a) Original      (b) Forward Modeling      (c) Autoencoder      (d) Temporal Contrastive

Figure 5: Decoded image reconstructions for different methods in ViZDoom Defend the Center.

**Forward modeling and temporal contrastive both work when there is no exogenous noise.**
In accordance with Theorem 1, we observe that in the case of both GridWorld (Figure 2) and
ViZDoom Defend the Center (Figure 4), these approaches learn a decoder $\phi$ that lead to success
with RL in the absence of any exogenous noise. For GridWorld, we find support for this result
with VQ bottleneck during representation learning (Figure 2(a)) whereas for ViZDoom Defend the
Center, we find support for this result even without the use of a VQ bottleneck (Figure 4(a)). These
results are further supported via qualitative evaluation through image decoding from the learned
latent representations (Figure 3) which show that these representations can recover critical elements
like walls. We find that autoencoder performs well in ViZDoom Defend the Center but not in
gridworld, which aligns with a lack of any theoretical understanding of autoencoders.

**Performance with exogenous noise.** We find that in the presence of exogenous noise (Figure 2,
Figure 4), representations from forward modeling achieve a lower performance specially in grid-
world, whereas temporal contrastive representations completely fail. One hypothesis for the stark
failure of temporal contrastive learning is that the agent can tell whether two observations are causal
or not, by simply focusing on the noisy diamonds that move in a predictive manner. Therefore,
the contrastive learning loss can be reduced by focusing entirely on the exogenous noise. Whereas,
forward modeling is more robust as it needs to predict future observations, and the agent's state is
more helpful for doing that than noise. This shows in the reconstructions (Figure 3(b)(d), Figure
5(b)(d)). As expected, the reconstructions for forward modeling continue to capture state-relevant
information, whereas for temporal contrastive they focus on noise and miss relevant state informa-
tion. In Appendix B.6, we formally prove that there exists an instance where forward modeling can
recover the latent state for low-levels of exogenous noise, whereas temporal contrastive cannot do
so for any level of exogenous noise.

**Comparison with ACRO.** Finally, we draw
a comparison between the performance of
video-pretrained representation and ACRO
which uses trajectory data. ACRO achieves
the strongest performance across all tasks
(Figure 2, Figure 4). Additionally, we also
observe that as we increase the size of the
exogenous noise elements in the observa-
tion space (Figure 6), the performance of
forward modeling, the overall best video-
based approach, degrades more drastically
compared to ACRO. This agrees with our
theoretical finding (Theorem 2) that learn-
ing representations from video-based data is
significantly harder than trajectory-based data when exogenous noise is present.

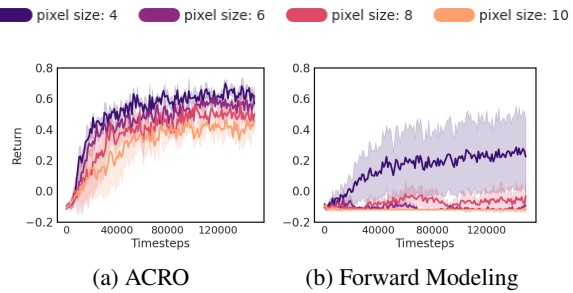

(a) ACRO      (b) Forward Modeling

Figure 6: RL performance with varying size for ex-
ogenous noise in the GridWorld environment.

## 6   CONCLUSION

Videos are a naturally available source of data for training representations for RL. In this work,
we study whether existing video-based representation learning methods are provably effective for
downstream RL tasks. We provide both upper and lower bounds for these methods in two theoretical
settings and provide empirical validation of our findings on 3 visual domains. Using our theoretical
tools to develop better video-based representation learning methods and extending our analysis to
other formal settings are natural future work directions.

ACKNOWLEDGEMENTS.

We thank Sam Devlin, Ching-An Cheng, Andrey Kolobov, and Adith Swaminathan for useful discussions. This work was done while AS was a postdoctoral researcher at Microsoft Research New York.

ETHICS STATEMENT

In our paper, we run experiments on two open-source simulated RL environments. All data was collected in simulation and no real-world dataset was used in this work.

REPRODUCIBILITY STATEMENT

The code is publicly available at `https://github.com/microsoft/Intrepid`. We used publicly available RL environments for our simulated experiments and used pretrained or randomized policies for data collection as described in Appendix C.

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

# Appendix

## Table of Contents

## A  ADDITIONAL RELATED WORK

**Representation Learning for Reinforcement Learning**  A line of research on recurrent state space models is essentially concerned with the next-frame approach, although typically with conditioning on actions. Moreover, to model uncertainty in the observations, a latent variable with a posterior depending on the current observation (or even a sequence of future observations) is typically introduced. (Ke et al., 2019) considered learning such a sequential prediction model which predicts observations and conditions on actions. They used a latent variable with a posterior depending on future observations to model uncertainty. These representations were used for model-predictive control and improved imitation learning. Dreamer (Hafner et al., 2019; 2023) uses the next-frame objective but also conditions on actions. The IRIS algorithm (Micheli et al., 2023) uses the next-frame objective but uses the transformer architecture, again conditioning on actions. The InfoPower approach (Bharadhwaj et al., 2022) combines a one-step inverse model with a temporal contrastive objective. Sobal et al. (2022) explored using semi-supervised objectives for learning representations in RL, yet used action-labeled data. Wang et al. (2022) used a decoupled recurrent neural network approach to learn to extract endogenous states, but relied on action-labeled data to achieve the factorization. Deep Bisimulation for Control (Zhang et al., 2020) introduced an objective to encourage observations with similar value functions to map to similar representations.

Self-prediction methods such as BYOL-explore (Guo et al., 2022) proposed learning reward-free representations for exploration, but depended on open-loop prediction of future states conditioned on actions . An analysis paper studied a simplified action-free version of the self-prediction objective (Tang et al., 2023) and showed results in the absence of using actions, although this has not been instantiated empirically to our knowledge.

A further line of work from theoretical reinforcemnt learning has examined provably efficient objectives for discovering representations. Efroni et al. (2022) explored representation learning in the presence of exogenous noise, establishing a sample efficient algorithm. However Efroni et al. (2022) and the closely related work on filtering exogenous noise required actions (Lamb et al., 2022; Islam et al., 2022). Other theoretical work on learning representations for RL has required access to action-labeled data (Misra et al., 2020).

**Representation Learning from Videos**  Self-supervised representation learning from videos has a long history. Srivastava et al. (2015) used recurrent neural networks with a pixel prediction ob-

jective on future frames. Parthasarathy et al. (2022) explored temporal contrastive objectives for self-supervised learning from videos. They also found that the features learned well aligned with human perceptual priors, despite the model not being explicitly trained to achieve such alignment. Aubret et al. (2023) applied temporal contrastive learning to videos of objects being manipulated in a 3D space, showing that this outperformed standard augmentations used in computer vision.

**Using Video Data for Reinforcement Learning**    The VIPER method (Escontrela et al., 2023) uses a pre-trained autoregressive generative model over action-free expert videos as a reward signal for training an imitation learning agent. The Video Pre-training (VPT) algorithm (Baker et al., 2022) trained an inverse kinematics model on a small dataset of Minecraft videos and used the model to label a large set of unlabeled Minecraft videos from the internet. This larger dataset was then used for imitation learning and reinforcement learning for downstream tasks. Zhao et al. (2022) explicitly studied the challenges in using videos for representation learning in RL, identifying five key factors: task mismatch, camera configuration, visual feature shift, sub-optimal behaviors in the data, and robot morphology. Goo & Niekum (2019) learn reward functions for multi-step tasks from videos by leveraging a single video segmented with action labels (one-shot learning). Sikchi et al. (2022) propose a two-player ranking game between a policy and a reward function to satisfy pairwise performance rankings between behaviors. Their proposed method achieves state-of-the-art sample efficiency and can solve previously unsolvable tasks in the learning from observation (no actions) setting.

Recently some approaches have also considered recovering *latent actions* from video data using an encoder-decoder approach (Ye et al., 2022). In general, the lower bound in Theorem 2 applies to these methods and they do not provably work in the hard instances with exogenous noise. For example, the latent actions can capture *exogenous noise* instead of actions, if the former is more predictive of changes in the observations. However, in simpler cases such as 3D games, where the agent's action is typically most predictive of changes in observations, or in settings with no exogenous noise, one can expect these approaches to do well.

## B    PROOFS OF THEORETICAL STATEMENTS

We state our setting and general assumptions before presenting method specific results. We also include a table of notations in Table 1.

| Notation | Description |
|---|---|
| $[N]$ | Denotes the set $\{1, 2, \cdots, N\}$ |
| $\Delta(\mathcal{U})$ | Denotes the set of all distributions over a set $\mathcal{U}$ |
| $\mathrm{Unf}(\mathcal{U})$ | Uniform distribution over $\mathcal{U}$ |
| $\mathrm{supp}(\mathbb{P})$ | Support of a distribution $\mathbb{P} \in \Delta(\mathcal{U})$, i.e., $\mathrm{supp}(\mathbb{P}) = \{x \in \mathcal{U} \mid \mathbb{P}(x) > 0\}$. |
| $\mathcal{X}$ | Observation space |
| $\mathcal{S}$ | Latent endogenous state |
| $\mathcal{A}$ | Action space |
| $T : \mathcal{S} \to \mathcal{A} \to \Delta(\mathcal{S})$ | Transition dynamics |
| $R : \mathcal{S} \times \mathcal{A} \to [0, 1]$ | Reward function |
| $\mu$ | Start state distribution |
| $H$ | Horizon indicating the maximum number of actions per episode |
| $\phi^\star : \mathcal{X} \to \mathcal{S}$ | Endogenous state decoder |

Table 1: Description for mathematical notations.

We are given a dataset $\mathcal{D} = \left\{ (x_1^{(i)}, x_2^{(i)}, \cdots, x_H^{(i)}) \right\}_{i=1}^{n}$ of $n$ independent and identically distributed (iid) unlabeled episodes. We will use the word video and unlabeled episodes interchangeably. We assume the underlying data distribution is $D$. We denote the probability of an unlabeled episode as $D(x_1, x_2, \cdots, x_H)$. We assume that $D$ is generated by a mixture of Markovian policies $\Pi_D$, i.e., the generative procedure for $D$ is to sample a policy $\pi \in \Pi_D$ with probability $\Theta_\pi$ and then generate an entire episode using it. For this reason, we will denote $D = \Theta \circ \Pi_D$ where $\Theta$ is the

mixture distribution. We assume *no direct knowledge* about either $\Pi_D$ or $\Theta$, other than that the set of policies in $\Pi_D$ are Markovian. We define the *underlying* distribution over the action-labeled episode as $D(x_1, a_1, x_2, \cdots, x_H, a_H)$, of which the agent *only* gets to observe the $(x_1, x_2, \cdots, x_H)$. We will use the notation $D$ to refer to any distribution that is derived from the above joint distribution.

We assume that observations encode time steps. This can be trivially accomplished by simply concatenating the time step information to the observation. This also implies that observations from different time steps are different. Because of this property, we can assume that the Markovian policies used to realize $D$ were time homogenous, i.e., they only depend on observation and not observation and timestep pair (this is because we include timesteps in the observation). Therefore, for all $h \in [H]$ and $k \in \mathbb{N}$ we have:

$$D(x_{h+k} = x' \mid x_h = x) = D(x_{k+1} = x' \mid x_1 = x) \tag{4}$$

We denote $D(x_h)$ to define the marginal distribution over an observation $x_h$, and $D(x_h, x_{h+k})$ to denote the marginal distribution over a pair of observations $(x_h, x_{h+k})$ in the episode. We similarly define $D(x_h, a_h)$ as the distribution over observation action pairs $(x_h, a_h)$.

We assume that the video data has good coverage. This is stated formally below:

**Assumption 4** (State Coverage by $D$). *Given our policy class $\Pi$, there exists an $\eta_{min} > 0$ such that if $\sup_{\pi \in \Pi} \mathbb{P}_\pi(s_h = s) > 0$ for some $s \in \mathcal{S}$, then we assume $D(\phi^\star(x_h) = s) \geq \eta_{min}$.*

In practice, Assumption 4 can be satisfied since videos are more easily available than labeled episodes and we can hope that a large diverse collection of videos can provide reasonable coverage over the underlying state action space. E.g., for tasks like gaming, one can use hours of streaming data from many users.

Further, we also assume that the data policy depends only on the endogenous state. Recall that for an observation $x \in \mathcal{X}$, its endogenous state is given by $\phi^\star(x) \in \mathcal{S}$.

**Assumption 5** (Noise-Free Video Distribution). *For any $h$, $\pi \in \Pi_D$, $x_h \in supp\ \mathbb{P}_\pi$ and $a \in \mathcal{A}$, we have*

$$\pi(a \mid x_h) = \pi(a \mid \phi^\star(x_h)).$$

**Justification of Noise-Free Policy.** Typically, video data is created by humans. E.g., a human may be playing a game and the video data is collected by recording the user's screen. A user is unlikely to take actions relying on iid or exogenous noise in the observation process. Therefore, the collected data can be expected to obey the noise-free assumption.

**Multi-step transition.** We choose to analyze a multi-step variant of standard temporal contrastive and forward modeling algorithms that train on a dataset of pairs of observations $(x, x')$ that can be variable time steps apart. As our proof will show, this gives the algorithms more expressibility and allows them to learn correct representations for some problems that their single-step variants (i.e., the observations are adjacent) or fixed time-step variants (i.e., the observations are fixed time steps apart) cannot solve. We will use the variable $k$ to denote the time steps by which these observations differ. Formally, we will call $(x, k, x')$ as a multi-step transition where $x$ was observed at some time step $h$, and $x'$ was observed at $h + k$. For the single-step variant of the algorithms, we have $k = 1$. For the fixed multi-step variant, we have $k > 1$ but $k$ is fixed. Finally, in the general multi-step variant, we will assume that $k$ is picked from $\text{Unf}([K])$ where $K$ is a fixed upper bound.

**Extending episode to $H + K$.** When using $k > 1$, we may want to collect a multi-step transition $(x, k, x')$ where $x = x_H$ to allow learning state representation for time step $H$. However, at this point, we don't have time steps left to observe $x_{H+k}$. We alleviate this by assuming that we can allow an episode to run till $H + K$ if necessary. In practice, this is not a problem where the algorithm sets the horizon and not the environment. However, if we cannot go past $H$, then we can instead assume that all states are reachable by the time step $H - K$ and so their state representation can be learned when $x$ is selected at $x_{H-K}$. In our analysis ahead, we make the former setting that the episodes can be extended to $H + K$, but it can be easily rephrased to work with the other setting.

For both the forward model and the temporal contrastive approach, we assume access to a dataset $\mathcal{D}_{\text{for}} = \left((x^{(i)}, k^{(i)}, x'^{(i)})\right)_{i=1}^n$ of pairs of observations. We define a few different distributions that

can be used to generate this set. For a given $k \in [K]$, we define a distribution $D_k$ over $k$-step separate observations as:

$$D_k\left(X = x, X' = x'\right) = \frac{1}{H} \sum_{h=1}^{H} D(x_h = x, x_{h+k} = x') \tag{5}$$

We can sample $(x, k, x') \sim D_k(X, X')$ by sampling an episode $(x_1, x_2, \cdots, x_H) \sim D$, and then sampling a $h \sim \text{Unf}([H])$, and choosing $x = x_h$ and $x' = x_{h+k}$.

We also define a distribution $D_{\text{unf}}$ where we also sample $k$ uniformly over available choices:

$$D_{\text{unf}}\left(X = x, k, X' = x'\right) = \frac{1}{K} D_k(x_h = x, x_{h+k} = x') \tag{6}$$

We can sample $(x, k, x') \sim D_{\text{unf}}(X, X')$ by sampling an episode $(x_1, x_2, \cdots, x_H) \sim D$, and then sampling $h \in [H]$, and sampling $k \in [K]$, and choosing $(x_h, x_{h+k})$ as the selected pair.

We define a useful notation $\rho \in \Delta(\mathcal{X})$ as:

$$\rho(X = x) = \frac{1}{H} \sum_{h=1}^{H} D(x_h = x). \tag{7}$$

The distribution $\rho(X)$ is a good distribution to sample from as it covers states across all time steps. Finally, because of Assumption 4, we have the following:

$$\forall s \in \mathcal{S}, \qquad \rho(s) \geq \frac{\eta_{\min}}{H} \tag{8}$$

This is because we assume every state $s \in \mathcal{S}$, is visited at some time step $t$, and so we have $D(s_t = s) \geq \eta_{\min}$, and $\rho(s) = \frac{1}{H} \sum_{h=1}^{H} D(s_h = s) \geq \frac{1}{H} D(s_t = s) \geq \frac{\eta_{\min}}{H}$.

It can be easily verified that for both $D_k(X, X')$ and $D_{\text{unf}}(X, X')$, their marginals over $X$ is given by $\rho(X)$. Both $D_k$ and $D_{\text{unf}}$ satisfy the noise-free property. We prove this using the next two Lemma.s

**Lemma 1** (Property of Noise-Free policy). *Let $\pi$ be a policy such that for any $x \in \mathcal{X}$, we have $\pi(a \mid x) = \pi(a \mid \phi^\star(x))$. Then for any $h \in [H]$ and $k \in [K]$ we have $\mathbb{P}_\pi(x_{h+k} = x' \mid x_h = x)$ only depend on $\phi^\star(x)$ and this common value is defined by $\mathbb{P}_\pi(x_{h+k} \mid s_h = \phi^\star(x))$.*

*Proof.* The proof is by induction on $k$. For $k = 1$ we have:

$$\mathbb{P}_\pi(x_{h+1} = x' \mid x_h = x) = \sum_{a \in \mathcal{A}} T(x' \mid x, a)\pi(a \mid x_h = x) = \sum_{a \in \mathcal{A}} T(x' \mid \phi^\star(x), a)\pi(a \mid x_h = \phi^\star(x)),$$

and as the right hand side only depends on $\phi^\star(x)$, the base case is proven. For the general case, we have:

$$\begin{aligned}
\mathbb{P}_\pi(x_{h+k} = x' \mid x_h = x) &= \sum_{\tilde{x} \in \mathcal{X}} \mathbb{P}_\pi(x_{h+k} = x', x_{h+k-1} = \tilde{x} \mid x_h = x) \\
&= \sum_{\tilde{x} \in \mathcal{X}} \mathbb{P}_\pi(x_{h+k} = x' \mid x_{h+k-1} = \tilde{x})\mathbb{P}_\pi(x_{h+k-1} = \tilde{x} \mid x_h = x) \\
&= \sum_{\tilde{x} \in \mathcal{X}} \mathbb{P}_\pi(x_{h+k} = x' \mid x_{h+k-1} = \tilde{x})\mathbb{P}_\pi(x_{h+k-1} = \tilde{x} \mid x_h = \phi^\star(x)),
\end{aligned}$$

where the second step uses the fact that $\pi$ is Markovian and the last step uses the inductive case for $k - 1$. $\qquad\square$

**Lemma 2** (Distribution over Pairs). *Let $k \in [K]$, $x \in \text{supp } \rho(X)$, then the distribution $D_k(X' \mid x)$ only depends on $\phi^\star(x)$. This allows us to define $D_k(X' \mid \phi^\star(x))$ as this common value. Similarly, the distribution $D_{unf}(X' \mid x, k)$ depends only on $\phi^\star(x)$ and $k$. We define this common value as $D_{unf}(X' \mid \phi^\star(x), k)$.*

*Proof.* For any $k$ we have:

$$D_k(X = x, X' = x') = \frac{1}{H} \sum_{h=1}^{H} D(x_h = x, x_{h+k} = x')$$

$$= \frac{1}{H} \sum_{h=1}^{H} \sum_{\pi \in \Pi_D} \Theta_\pi \mathbb{P}_\pi(x_h = x, x_{h+k} = x')$$

$$= \frac{1}{H} \sum_{h=1}^{H} \sum_{\pi \in \Pi_D} \Theta_\pi \mathbb{P}_\pi(x_h = x) \mathbb{P}(x_{h+k} = x' \mid x_h = x)$$

$$= \frac{1}{H} \sum_{h=1}^{H} \sum_{\pi \in \Pi_D} \Theta_\pi \mathbb{P}_\pi(x_h = x) \mathbb{P}_\pi(x_{h+k} = x' \mid s_h = \phi^\star(x)), \quad \text{(using Lemma 1)}$$

$$= \frac{q(x \mid \phi^\star(x))}{H} \sum_{h=1}^{H} \sum_{\pi \in \Pi_D} \Theta_\pi \mathbb{P}_\pi(s_h = \phi^\star(x)) \mathbb{P}_\pi(x_{h+k} = x' \mid s_h = \phi^\star(x))$$

The marginal $D_k(X = x)$ is given by:

$$D_k(X = x) = \frac{1}{H} \sum_{h=1}^{H} \sum_{\pi \in \Pi_D} \Theta_\pi q(x \mid \phi^\star(x)) \mathbb{P}_\pi(s_h = \phi^\star(x)) = \frac{q(x \mid \phi^\star(x))}{H} \sum_{h=1}^{H} D_k(s_h = \phi^\star(x)).$$

The conditional $D_k(X' = x' \mid X = x)$ is given by:

$$D_k(X' = x' \mid X = x) = \frac{D_k(X = x, X' = x')}{D_k(x)}$$

$$= \frac{\sum_{h=1}^{H} \sum_{\pi \in \Pi_D} \Theta_\pi \mathbb{P}_\pi(s_h = \phi^\star(x)) \mathbb{P}_\pi(x_{h+k} = x' \mid s_h = \phi^\star(x))}{\sum_{h=1}^{H} D_k(s_h = \phi^\star(x))}$$

Therefore, the conditional $D_k(X' = x' \mid X = x$ only depends on $\phi^\star(x)$, and we define this common value as $D_k(X' = x' \mid s = \phi^\star(x))$.

The proof for $D_{\text{unf}}$ is similar. We can use the property of $D_k$ that we have proven to get:

$$D_{\text{unf}}(X' = x' \mid X = x, k) = \frac{D_{\text{unf}}(X = x, k, X' = x')}{\sum_{\tilde{x} \in \mathcal{X}} D_{\text{unf}}(X = x, k, X' = \tilde{x})}$$

$$= \frac{D_k(X = x, X' = x')}{\sum_{\tilde{x} \in \mathcal{X}} D_k(X = x, X' = \tilde{x})}$$

$$= \frac{D_k(X' = x' \mid X = x)}{\sum_{\tilde{x} \in \mathcal{X}} D_k(X' = \tilde{x} \mid X = x)}$$

$$= \frac{D_k(X' = x' \mid X = \phi^\star(x))}{\sum_{\tilde{x} \in \mathcal{X}} D_k(X' = \tilde{x} \mid X = \phi^\star(x))}.$$

Therefore, $D_{\text{unf}}(X' = x' \mid X = x, k)$ only depends on $\phi^\star(x)$. We will define the common values as $D_{\text{unf}}(X' = x' \mid s = \phi^\star(x), k)$. □

Lemma 2 allows us to define $D_k(x' \mid \phi^\star(x))$ and $D_{\text{unf}}(x' \mid \phi^\star(x), k)$, as the distribution only depends on the latent state.

### B.1 Upper Bound for the Forward Model Baseline

Let $\mathcal{D}_{\text{for}} = \{(x^{(i)}, k^{(i)}, x'^{(i)})\}_{i=1}^{n}$ be a pair of iid multi-step observations. We will collect this dataset in one of three ways:

1. Single step ($k = 1$), in this case we will sample $(x^{(i)}, x'^{(i)}) \sim D_k(X, X')$. As explained before, we can get this sample using the episode data. We save $(x^{(i)}, k, x'^{(i)})$ as our sample.

2. Fixed multi-step. We use a fixed $k > 1$, and sample $(x^{(i)}, x'^{(i)}) \sim D_k(X, X')$. We save $(x^{(i)}, k, x'^{(i)})$ as our sample.

3. Variable multi-step. We sample $(x, k, x') \sim D_{unf}(X, k, X')$ and use it as our sample.

We will abstract these three choices using a general notion of $D_{pr} \in \Delta(\mathcal{X} \times [K] \times \mathcal{X})$. In the first two cases, we assume we have point-mass distribution over $k$ and given this $k$, we sample from $D_k(X, X')$. We will assume $(x^{(i)}, k^{(i)}, x'^{(i)}) \sim D_{pr}$. We can create $\mathcal{D}_{for}$ from the dataset $\mathcal{D}$ of $n$ episodes sampled from $D$ using the sampling procedures explained earlier. Note that as marginals over both $D_k(X)$ and $D_{unf}(X)$ is $\rho(X)$, therefore, the marginals over $D_{pr}(X)$ is also $\rho(X)$. Additionally, we will define $D_{pr}(k)$ as the marginal over $k$ which is either point-mass in the first two sampling procedures and $Unf([K])$ in the third procedure.

We assume access to two function classes. The first is a decoder class $\Phi_N : \mathcal{X} \to [N]$ where $N$ is a given number that satisfies $N \geq |\mathcal{S}|$. The second is a conditional probability class $\mathcal{F} : [N] \times [K] \to \Delta(\mathcal{X})$.

**Assumption 6.** *(Realizability of $\Phi$ and $\mathcal{F}$) We assume that there exists $\phi^\circ \in \Phi_N$ and $f^\circ \in \mathcal{F}$ such that $f^\circ(x' \mid \phi^\circ(x), k) = D_{pr}(x' \mid x, k) = D_{pr}(x' \mid \phi^\star(x), k)$ for all $(x, k) \sim D_{pr}(\cdot, \cdot)$.*

This assumption firstly is non-vacuous as $D_{pr}(x' \mid x) = D_{pr}(x' \mid \phi^\star(x))$, and therefore, we can apply a bottleneck function $\phi$ and still assume realizability. For example, we can assume that $\tilde{\phi}$ is the same as $\phi^\star$ up to the relabeling of its output, and $\tilde{f}(x' \mid i) = D_{pr}(x' \mid s)$.

Let $\widehat{f}, \widehat{\phi}$ be the empirical solution to the following maximum likelihood problem.

$$\widehat{f}, \widehat{\phi} = \arg \max_{f \in \mathcal{F}, \phi \in \Phi_N} \frac{1}{n} \sum_{i=1}^{n} \ln f\left(x'^{(i)} \mid \phi(x^{(i)}), k^{(i)}\right) \tag{9}$$

Note that when $k$ is fixed (we sample from $D_k$), then information theoretically there is no advantage of condition on $k$ and it can be dropped from optimization.

As we are in a realizable setting (Assumption 6), we can use standard maximum likelihood guarantees to get the following result.

**Proposition 3** (Generalization Bound). *Fix $\delta \in (0, 1)$, then with probability at least $1 - \delta$, we have:*

$$\mathbb{E}_{(x,k) \sim D_{pr}} \left[ \left\| D_{pr}(X' \mid x, k) - \widehat{f}(X' \mid \widehat{\phi}(x), k) \right\|_{TV}^2 \right] \leq \Delta^2(n; \delta),$$

*where $\Delta^2(n; \delta) = \frac{2}{n} \ln\left(\frac{|\Phi| \cdot |\mathcal{F}|}{\delta}\right)$.*

For proof see Chapter 7 of Geer (2000).

Finally, we assume that the forward modeling objective is expressive to allow the separation of states. While, this seems like assuming that the objective works, our goal is to establish a formal notion of the margin so we can verify it later in different settings to see when it holds.

**Assumption 7.** *(Forward Modeling Margin). We assume there exists a $\beta_{for} \in (0, 1)$ such that:*

$$\inf_{s_1, s_2 \in \mathcal{S}, s_1 \neq s_2} \mathbb{E}_{k \sim D_{pr}} \left[ \| D_{pr}(X' \mid s_1, k) - D_{pr}(X' \mid s_2, k) \|_{TV} \right] \geq \beta_{for}$$

Note that this defines two types of margin depending on $D_{pr}$. When $k$ is a fixed value, the margin is given by:

$$\beta_{for}^{(k)} = \inf_{s_1, s_2 \in \mathcal{S}, s_1 \neq s_2} \| D_{pr}(X' \mid s_1, k) - D_{pr}(X' \mid s_2, k) \|_{TV}.$$

When we sample $k \sim Unf([K])$ then the margin is given by:

$$\beta_{for}^{(u)} = \inf_{s_1, s_2 \in \mathcal{S}, s_1 \neq s_2} \frac{1}{K} \sum_{k=1}^{K} \| D_{pr}(X' \mid s_1, k) - D_{pr}(X' \mid s_2, k) \|_{TV}.$$

We will use the abstract notion $\beta_{for}$ for forward margin which will be equal to $\beta_{for}^{(k)}$ or $\beta_{for}^{(u)}$ depending on our sampling procedure. It is easy to see that $\beta_{for}^{(u)} = \frac{1}{K} \sum_{k=1}^{K} \beta_{for}^{(k)}$.

We are now ready to state our first main result.

**Proposition 4** (Recovering Endogenous State.). *Fix $\delta \in (0, 1)$, then with probability at least $1 - \delta$ we learn $\widehat{\phi}$ that satisfies:*

$$\mathbb{P}_{x_1, x_2 \sim \rho} \left( \phi^\star(x_1) \neq \phi^\star(x_2) \wedge \widehat{\phi}(x_1) = \widehat{\phi}(x_2) \right) \leq \frac{2\Delta(n, \delta)}{\beta_{for}}.$$

*Proof.* We start with a coupling argument where we sample $x_1, x_2$ independently from $D_{pr}(X)$ which is the same as $\rho(X)$.

$$\mathbb{E}_{x_1, x_2 \sim D_{pr}, k \sim D_{pr}} \left[ \mathbf{1}\left\{ \widehat{\phi}(x_1) = \widehat{\phi}(x_2) \right\} \|D_{pr}(X' \mid x_1, k) - D_{pr}(X' \mid x_2, k)\|_{\mathrm{TV}} \right]$$

$$\leq \mathbb{E}_{x_1, x_2 \sim D_{pr}, k \sim D_{pr}} \left[ \mathbf{1}\left\{ \widehat{\phi}(x_1) = \widehat{\phi}(x_2) \right\} \left\| \widehat{f}(X' \mid \widehat{\phi}(x_1), k) - D_{pr}(X' \mid x_1, k) \right\|_{\mathrm{TV}} \right]$$

$$+ \mathbb{E}_{x_1, x_2 \sim D_{pr}, k \sim D_{pr}} \left[ \mathbf{1}\left\{ \widehat{\phi}(x_1) = \widehat{\phi}(x_2) \right\} \left\| \widehat{f}(X' \mid \widehat{\phi}(x_1), k) - D_{pr}(X' \mid x_2, k) \right\|_{\mathrm{TV}} \right]$$

We bound these two terms separately

$$\mathbb{E}_{x_1, x_2 \sim D_{pr}, k \sim D_{pr}} \left[ \mathbf{1}\left\{ \widehat{\phi}(x_1) = \widehat{\phi}(x_2) \right\} \left\| \widehat{f}(X' \mid \widehat{\phi}(x_1), k) - D_{pr}(X' \mid x_1, k)) \right\|_{\mathrm{TV}} \right]$$

$$\leq \sqrt{\mathbb{E}_{x_1, x_2 \sim D_{pr}, k \sim D_{pr}} \left[ \mathbf{1}\left\{ \widehat{\phi}(x_1) = \widehat{\phi}(x_2) \right\} \right]} \cdot \sqrt{\mathbb{E}_{x_1, x_2 \sim D_{pr}, k \sim D_{pr}} \left[ \left\| \widehat{f}(X' \mid \widehat{\phi}(x_1), k) - D_{pr}(X' \mid x_1, k)) \right\|_{\mathrm{TV}}^2 \right]}$$

$$= \sqrt{\mathbb{E}_{x_1, x_2 \sim D_{pr}} \left[ \mathbf{1}\left\{ \widehat{\phi}(x_1) = \widehat{\phi}(x_2) \right\} \right]} \cdot \sqrt{\mathbb{E}_{(x, k) \sim D_{pr}} \left[ \left\| \widehat{f}(X' \mid \widehat{\phi}(x) - D_{pr}(X' \mid x)) \right\|_{\mathrm{TV}}^2 \right]}$$

$$\leq b \cdot \Delta,$$

where $b = \sqrt{\mathbb{E}_{x_1, x_2 \sim D_{pr}} \left[ \mathbf{1}\left\{ \widehat{\phi}(x_1) = \widehat{\phi}(x_2) \right\} \right]}$ and the second step uses Cauchy-Schwarz inequality. It is straightforward to verify that $b \in [0, 1]$. We bound the second term similarly

$$\mathbb{E}_{x_1, x_2 \sim D_{pr}, k \sim D_{pr}} \left[ \mathbf{1}\left\{ \widehat{\phi}(x_1) = \widehat{\phi}(x_2) \right\} \left\| \widehat{f}(X' \mid \widehat{\phi}(x_1), k) - D_{pr}(X' \mid x_2, k) \right\|_{\mathrm{TV}} \right]$$

$$= \mathbb{E}_{x_1, x_2 \sim D_{pr}} \left[ \mathbf{1}\left\{ \widehat{\phi}(x_1) = \widehat{\phi}(x_2) \right\} \left\| \widehat{f}(X' \mid \widehat{\phi}(x_2), k) - D_{pr}(X' \mid x_2, k) \right\|_{\mathrm{TV}} \right]$$

$$\leq b \cdot \Delta,$$

where the second step uses the crucial coupling argument that we can replace $x_1$ with $x_2$ because of the indicator $\mathbf{1}\left\{ \widehat{\phi}(x_1) = \widehat{\phi}(x_2) \right\}$, and the last step follows as we reduce it to the first term except we switch the names of $x_1$ and $x_2$. Combining the two upper bounds we get:

$$\mathbb{E}_{x_1, x_2 \sim D_{pr}, k \sim D_{pr}} \left[ \mathbf{1}\left\{ \widehat{\phi}(x_1) = \widehat{\phi}(x_2) \right\} \|D_{pr}(X' \mid x_1, k) - D_{pr}(X' \mid x_2, k)\|_{\mathrm{TV}} \right] \leq 2b \cdot \Delta$$

or, equivalently,

$$\mathbb{E}_{x_1, x_2 \sim D_{pr}} \left[ \mathbf{1}\left\{ \widehat{\phi}(x_1) = \widehat{\phi}(x_2) \right\} \underbrace{\mathbb{E}_{k \sim D_{pr}} \left[ \|D_{pr}(X' \mid x_1, k) - D_{pr}(X' \mid x_2, k)\|_{\mathrm{TV}} \right]}_{:=\Gamma(x_1, x_2)} \right] \leq 2b \cdot \Delta$$

Let $\Gamma(x_1, x_2) = \mathbb{E}_{k \sim D_{pr}} \left[ \|D_{pr}(X' \mid x_1, k) - D_{pr}(X' \mid x_2, k)\|_{\mathrm{TV}} \right]$. For any two observations, if $\phi^\star(x_1) = \phi^\star(x_2)$, then $\|D_{pr}(X' \mid x_1) - D_{pr}(X' \mid x_2)\|_{\mathrm{TV}} = 0$, and therefore, $\Gamma(x_1, x_2) = 0$ because of Lemma 2. Otherwise, $\Gamma(x_1, x_2)$ is at least $\beta_{\mathrm{for}}$, by Assumption 6. Combining these two observations we get:

$$\Gamma(x_1, x_2) \geq \beta_{\mathrm{for}} \mathbf{1}\{\phi^\star(x_1) \neq \phi^\star(x_2)\}$$

Combining the previous two inequalities we get:

$$\mathbb{E}_{x_1, x_2 \sim D_{pr}} \left[ \mathbf{1}\left\{ \widehat{\phi}(x_1) = \widehat{\phi}(x_2) \wedge \phi^\star(x_1) \neq \phi^\star(x_2) \right\} \right] \leq \frac{2b \cdot \Delta}{\beta_{\mathrm{for}}}$$

This directly gives

$$\mathbb{P}_{x_1, x_2 \sim D_{pr}} \left( \widehat{\phi}(x_1) = \widehat{\phi}(x_2) \wedge \phi^\star(x_1) \neq \phi^\star(x_2) \right) \leq \frac{2b\Delta}{\beta_{\text{for}}} \leq \frac{2\Delta}{\beta_{\text{for}}}.$$

The proof is completed by recalling that marginal $D_{pr}(X)$ is the same as $\rho(X)$. $\qquad \square$

Proposition 4 shows that the learned $\widehat{\phi}$ has one-sided error. If it merges two observations, then with high probability they are not from the same state. As $N = |\mathcal{S}|$, we will show below that the reverse is also true.

**Theorem 5.** *If $N = |\mathcal{S}|$, then there exists a bijection $\alpha : [N] \to \mathcal{S}$ such that for any $s \in \mathcal{S}$ we have:*

$$\mathbb{P}_{x \sim q(\cdot|s)} \left( \widehat{\phi}(x) = \alpha(s) \mid \phi^\star(x) = s \right) \geq 1 - \frac{4N^3 H^2 \Delta}{\eta_{min}^2 \beta_{for}},$$

*provided $\Delta < \frac{\eta_{min}^2 \beta_{for}}{N^2 H^2}$.*

*Proof.* We define a few shorthand below for any $j \in [N]$ and $\tilde{s} \in \mathcal{S}$

$$\mathbb{P}(j, \tilde{s}) = \mathbb{P}_{x \sim \rho} \left( \widehat{\phi}(x) = j \wedge \phi^\star(x) = \tilde{s} \right)$$

$$\rho(j) = \mathbb{P}_{x \sim \rho} \left( \widehat{\phi}(x) = j \right)$$

$$\rho(\tilde{s}) = \mathbb{P}_{x \sim \rho} \left( \phi^\star(x) = \tilde{s} \right).$$

It is easy to verify that $\mathbb{P}(j, \tilde{s})$ is a joint distribution with $\rho(j)$ and $\rho(\tilde{s})$ as its marginals.

Fix $i \in [N]$ and $s \in \mathcal{S}$.

$$\mathbb{P}_{x_1, x_2 \sim \rho} \left( \widehat{\phi}(x_1) = \widehat{\phi}(x_2) \wedge \phi^\star(x_1) \neq \phi^\star(x_2) \right)$$

$$= \mathbb{P}_{x_1, x_2 \sim \rho} \left( \cup_{\tilde{s} \in \mathcal{S}, j \in [N]} \left\{ \widehat{\phi}(x_1) = j \wedge \widehat{\phi}(x_2) = j \wedge \phi^\star(x_1) = \tilde{s} \wedge \phi^\star(x_2) \neq \tilde{s} \right\} \right)$$

$$\geq \mathbb{P}_{x_1, x_2 \sim \rho} \left( \widehat{\phi}(x_1) = i \wedge \widehat{\phi}(x_2) = i \wedge \phi^\star(x_1) = s \wedge \phi^\star(x_2) \neq s \right)$$

$$= \mathbb{P}_{x_1 \sim \rho} \left( \widehat{\phi}(x_1) = i \wedge \phi^\star(x_1) = s \right) \mathbb{P}_{x_2 \sim \rho} \left( \widehat{\phi}(x_2) = i \wedge \phi^\star(x_2) \neq s \right)$$

$$= \mathbb{P}_{x \sim \rho} \left( \widehat{\phi}(x) = i \wedge \phi^\star(x) = s \right) \left( \sum_{s' \in \mathcal{S}} \mathbb{P}_{x \sim \rho} \left( \widehat{\phi}(x) = i \wedge \phi^\star(x) = s' \right) - \mathbb{P}_{x \sim \rho}(\widehat{\phi}(x) = i \wedge \phi^\star(x) = s) \right)$$

$$= \mathbb{P}(i, s) \left( \sum_{s' \in \mathcal{S}} \mathbb{P}(i, s') - \mathbb{P}(i, s) \right)$$

$$= \mathbb{P}(i, s) \left( \rho(i) - \mathbb{P}(i, s) \right).$$

Combining this with Proposition 4, we get:

$$\forall i \in [N], s \in \mathcal{S}, \qquad \mathbb{P}(i, s) \left( \rho(i) - \mathbb{P}(i, s) \right) \leq \Delta' := \frac{2\Delta}{\beta_{\text{for}}}$$

where we have used a shorthand $\Delta' = 2\Delta/\beta_{\text{for}}$. We define a mapping $\alpha : \mathcal{S} \to [N]$ where for any $s \in \mathcal{S}$:

$$\alpha(s) = \arg \max_{j \in [N]} \mathbb{P}(j, s) \tag{10}$$

We immediately have:

$$\mathbb{P}(\alpha(s), s) = \max_{j \in [N]} \mathbb{P}(j, s) \geq \frac{1}{N} \sum_{j=1}^{N} \mathbb{P}(j, s) = \frac{1}{N} \rho(s) \geq \frac{\eta_{min}}{NH}, \tag{11}$$

where we use the fact that max is greater than average in the first inequality, and Equation 8. Further, for every $s \in \mathcal{S}$, we have:

$$\mathbb{P}(\alpha(s), s) \left( \rho(\alpha(s)) - \mathbb{P}(\alpha(s), s) \right) \leq \Delta'.$$

Plugging the lower bound $\mathbb{P}(\alpha(s), s) \geq \frac{\eta_{\min}}{NH}$, we get:

$$\mathbb{P}(\alpha(s), s) \geq \rho(\alpha(s)) - \frac{NH\Delta'}{\eta_{\min}}. \tag{12}$$

We now show that if $\Delta' < \frac{\eta_{\min}^2}{2N^2H^2}$, then $\alpha(s)$ is a bijection. Let $s_1$ and $s_2$ be such that $\alpha(s_1) = \alpha(s_2) = i$. Then using the above Equation 12 we get $\mathbb{P}(i, s_1) \geq \rho(i) - \frac{NH\Delta'}{\eta_{\min}}$ and $\mathbb{P}(i, s_2) \geq \rho(i) - \frac{NH\Delta'}{\eta_{\min}}$. We have:

$$\rho(i) = \sum_{\tilde{s} \in \mathcal{S}} \mathbb{P}(i, \tilde{s}) \geq \mathbb{P}(i, s_1) + \mathbb{P}(i, s_2) \geq 2\rho(i) - \frac{2NH\Delta'}{\eta_{\min}}$$

This implies $\frac{2N\Delta'}{\eta_{\min}} \geq \rho(i)$ but as $\rho(i) = \rho(\alpha(s_1)) \geq \mathbb{P}(\alpha(s_1), s_1) \geq \frac{\eta_{\min}}{NH}$ (Equation 11), we get $\frac{2NH\Delta'}{\eta_{\min}} \geq \frac{\eta_{\min}}{NH}$ or $\Delta' \geq \frac{\eta_{\min}^2}{2N^2H^2}$. However, as we assume that $\Delta' < \frac{\eta_{\min}^2}{2N^2H^2}$, therefore, this is a contradiction. This implies $\alpha(s_1) \neq \alpha(s_2)$ for any two different states $s_1$ and $s_2$. Since we assume $|N| = |\mathcal{S}|$, this implies $\alpha$ is a bijection.

Fix $s \in \mathcal{S}$ and let $i \neq \alpha(s)$. As $\alpha$ is a bijection, let $\tilde{s} = \alpha^{-1}(i)$, we can show that $\mathbb{P}(i, s)$ is small:

$$\mathbb{P}(i, s) \leq \rho(i) - \mathbb{P}(i, \tilde{s}) = \rho(\alpha(\tilde{s})) - \mathbb{P}(\alpha(\tilde{s}), \tilde{s}) \leq \frac{NH\Delta'}{\eta_{\min}} \tag{13}$$

where we use $s \neq \tilde{s}$ and Equation 12.

This allows us to show that $\mathbb{P}(\alpha(s) \mid s)$ is high as follows:

$$\begin{aligned}
\mathbb{P}(\alpha(s) \mid s) = \frac{\mathbb{P}(\alpha(s), s)}{\rho(s)} &= \frac{\mathbb{P}(\alpha(s), s)}{\mathbb{P}(\alpha(s), s) + \sum_{i=1, i\neq\alpha(s)}^{N} \mathbb{P}(i, s)} \\
&\geq \frac{\mathbb{P}(\alpha(s), s)}{\rho(\alpha(s)) + \frac{N^2H\Delta'}{\eta_{\min}}}, \\
&\geq \frac{\rho(\alpha(s)) - \frac{NH\Delta'}{\eta_{\min}}}{\rho(\alpha(s)) + \frac{N^2H\Delta'}{\eta_{\min}}} \\
&= 1 - \frac{\left(\frac{N^2H\Delta'}{\eta_{\min}} + \frac{NH\Delta'}{\eta_{\min}}\right)}{\rho(\alpha(s)) + \frac{N^2H\Delta'}{\eta_{\min}}} \\
&\geq 1 - \frac{2N^2H^2\Delta'}{\eta_{\min}\rho(\alpha(s))} \\
&\geq 1 - \frac{2N^3H^2\Delta'}{\eta_{\min}^2},
\end{aligned}$$

where the first inequality uses Eq. (13) and $\rho(\alpha(s)) \geq \mathbb{P}(\alpha(s), s)$, second inequality uses Eq. (12), and the last step uses $\rho(\alpha(s)) \geq \mathbb{P}(\alpha(s), s) \geq \frac{\eta_{\min}}{NH}$.

The proof is completed by noting that:

$$\mathbb{P}_{x \sim q(\cdot|s)} \left( \widehat{\phi}(x) = \alpha(s) \right) = \mathbb{P}_{x \sim \rho} \left( \widehat{\phi}(x) = \alpha(s) \mid \phi^\star(x) = s \right) = \mathbb{P}(\alpha(s) \mid s).$$

$\square$

Let $\mathscr{A}$ be a PAC RL algorithm for tabular MDPs. We assume that this algorithm's sample complexity is given by $n_{\mathrm{samp}}(S, A, H, \varepsilon, \delta)$ where $S$ and $A$ are the size of the state space and action space of the tabular MDP, $H$ is the horizon, and $(\varepsilon, \delta)$ are the typical PAC RL hyperparameters denoting tolerance and failure probability. Formally, the algorithm $\mathscr{A}$ interacts with a tabular MDP $\mathbb{M}$ for $n_{\mathrm{samp}}(S, A, H, \varepsilon, \delta)$ episodes and outputs a policy $\widehat{\varphi} : \mathcal{S} \times [H] \to \mathcal{A}$ such that with probability at least $1 - \delta$ we have:

$$\sup_{\varphi \in \Psi_{\mathrm{all}}} V_{\mathbb{M}}(\varphi) - V_{\mathbb{M}}(\widehat{\varphi}) \leq \varepsilon,$$

where $\Psi_{\text{all}}$ is the space of all policies of the type $\mathcal{S} \times [H] \to \mathcal{A}$.

We assume that we are given knowledge of the desired $(\varepsilon, \delta)$ hyperparameters in the downstream RL task during the representation pre-training phase so we can use the right amount of data.

**Induced Finite MDP.** The latent MDP inside a block MDP is a tabular MDP with state space $\mathcal{S}$, action space $\mathcal{A}$, horizon $H$, transition dynamics $T$, reward function $R$, and a start state distribution of $\mu$. If we directly had access to this latent MDP, say via the true decoding function $\phi^\star$, then we can apply the algorithm $\mathscr{A}$ and learn the optimal latent policy $\varphi^\star$ which we can couple with $\phi^\star$ and learn the optimal observation-based policy. Formally, we write this observation-based policy as $\varphi \circ \phi^\star : \mathcal{X} \times [H] \to \mathcal{A}$ given by $\varphi(\phi^\star(x), h)$. We dont have access to $\phi^\star$, but we have access to $\widehat{\phi}$ that with high probability for a given $x$ outputs a state which is same as $\phi^\star(x)$ up to the learned $\alpha$-bijection. We, therefore, define the induced MDP $\mathbb{M}$ as the finite MDP with state space $\widehat{\mathcal{S}}$, action space $\mathcal{A}$, transition function $\widehat{T}$, reward function $\widehat{R}$ and start state distribution $\widehat{\mu}$. These same as the latent Block MDP but where the true state $s$ is replaced by $\alpha(s)$. It is this induced $\mathbb{M}$ that the tabular MDP algorithm $\mathscr{A}$ will see with high probability.

**Proposition 6** (PAC RL Bound). *Let $\mathscr{A}$ be a PAC RL algorithm for tabular MDPs and $n_{samp}$ is its sample complexity. Let $\widehat{\phi} : \mathcal{X} \to [N]$ be a decoder pre-trained using video data and $\alpha : \mathcal{S} \to [N]$ is a bijection such that:*

$$\forall s \in \mathcal{S}, \qquad \mathbb{P}_{x \sim q(\cdot|s)} \left( \widehat{\phi}(x) = \alpha(s) \right) \geq 1 - \vartheta,$$

*then let $\widehat{\varphi}$ be the policy returned by $\mathscr{A}$ on the tabular MDP induced by $\widehat{\phi}(x)$. Then we have with probability at least $1 - \delta - n_{samp}(S, A, H, \varepsilon, \delta)H\vartheta$:*

$$\sup_{\pi \in \Pi} V(\pi) - V(\varphi \circ \widehat{\phi}) \leq \varepsilon + 2H^2\vartheta$$

*Proof.* The algorithm runs for $n_{\text{samp}}(S, A, H, \varepsilon, \delta)$ episodes. This implies the agent visits $n_{\text{samp}}(S, A, H, \varepsilon, \delta)H$ many latent states. If the decoder maps every such state $s$ to the correct permutation $\alpha(s)$, then the tabular MDP algorithm is running as if it ran on the induced MDP $\mathbb{M}$. The probability of failure is bounded by $n_{\text{samp}}(S, A, H, \varepsilon, \delta)H\vartheta$ as all these failures are independent given the state. Further, the failure probability of the tabular MDP algorithm itself is $\delta$. This leads to the total failure probability of $\delta + n_{\text{samp}}(S, A, H, \varepsilon, \delta)H\vartheta$.

Let $\Pi$ be the set of observation-based policies we are competing with and which includes the optimal observation-based policy $\pi^\star$. We can write $\sup_{\pi \in \Pi} V(\pi) = V_{\mathbb{M}}(\varphi^\star)$ where we use the subscript $\mathbb{M}$ to denote that the latent policy is running in the induced MDP $\mathbb{M}$. Further, for any latent policy $\varphi$ we have $V(\varphi \circ \alpha \circ \phi^\star) = V_{\mathbb{M}}(\varphi)$ as the decoder $\alpha \circ \phi^\star : x \mapsto \alpha(\phi^\star(x))$ give me access to the true state of the induced MDP $\mathbb{M}$. Then with probability at least $1 - \delta$, we have:

$$V_{\mathbb{M}}(\varphi^\star) - V_{\mathbb{M}}(\widehat{\varphi}) \leq \varepsilon$$

This allows us to bound the sub-optimality of the learned observation-based policy $\widehat{\varphi} \circ \widehat{\phi}$ as:

$$\sup_{\pi \in \Pi} V(\pi) - V(\widehat{\varphi} \circ \widehat{\phi}) = V(\varphi^\star \circ \alpha \circ \phi^\star) - V(\widehat{\varphi} \circ \alpha \circ \phi^\star) + V(\widehat{\varphi} \circ \alpha \circ \phi^\star) - V(\widehat{\varphi} \circ \widehat{\phi})$$

$$= V_{\mathbb{M}}(\varphi^\star) - V_{\mathbb{M}}(\widehat{\varphi}) + V(\widehat{\varphi} \circ \alpha \circ \phi^\star) - V(\widehat{\varphi} \circ \widehat{\phi})$$

$$\leq \varepsilon + V(\widehat{\varphi} \circ \alpha \circ \phi^\star) - V(\widehat{\varphi} \circ \widehat{\phi})$$

Here we use $\widehat{\varphi} \circ \alpha \circ \phi^\star$ to denote an observation-based policy that takes action as $\widehat{\varphi}(\alpha(\phi^\star(x)), h)$.

We bound $V(\widehat{\varphi} \circ \alpha \circ \phi^\star) - V(\widehat{\varphi} \circ \widehat{\phi})$ below. Let $\mathcal{E}_h = \{\widehat{\phi}(x_h) = \alpha(\phi^\star(x_h))\}$ and $\mathcal{E} = \cap_{h=1}^{H} \mathcal{E}_h$ be two events. We have $\mathbb{P}(\mathcal{E}_h) \geq 1 - \vartheta$. Further, using union bound we have $\mathbb{P}(\mathcal{E}^c) = \mathbb{P}(\cup_{h=1}^{H} \mathcal{E}_h^c) \leq \sum_{h=1}^{H} \mathbb{P}(\mathcal{E}_h^c) \leq H\vartheta$.

We first prove an upper bound on $V(\widehat{\varphi} \circ \alpha \circ \phi^\star)$:

$$V(\widehat{\varphi} \circ \alpha \circ \phi^\star) = \mathbb{E}_{\widehat{\varphi} \circ \alpha \circ \phi^\star} \left[ \sum_{h=1}^{H} r_h \right]$$

$$= \mathbb{E}_{\widehat{\varphi} \circ \alpha \circ \phi^\star} \left[ \sum_{h=1}^{H} r_h \mid \mathcal{E} \right] \mathbb{P}_{\widehat{\varphi} \circ \alpha \circ \phi^\star}(\mathcal{E}) + \mathbb{E}_{\widehat{\varphi} \circ \alpha \circ \phi^\star} \left[ \sum_{h=1}^{H} r_h \mid \mathcal{E}^c \right] \mathbb{P}_{\widehat{\varphi} \circ \alpha \circ \phi^\star}(\mathcal{E}^c)$$

$$\leq \mathbb{E}_{\widehat{\varphi} \circ \alpha \circ \phi^\star} \left[ \sum_{h=1}^{H} r_h \mid \mathcal{E} \right] + H^2 \vartheta$$

$$= \mathbb{E}_{\widehat{\varphi} \circ \widehat{\phi}} \left[ \sum_{h=1}^{H} r_h \mid \mathcal{E} \right] + H^2 \vartheta$$

Here we have used the fact that value of any policy is in $[0, H]$ since the horizon is $H$ and the rewards are in $[0, 1]$.

We next prove a lower bound on $V(\widehat{\varphi} \circ \widehat{\phi})$:

$$V(\widehat{\varphi} \circ \widehat{\phi}) = \mathbb{E}_{\widehat{\varphi} \circ \widehat{\phi}} \left[ \sum_{h=1}^{H} r_h \right]$$

$$= \mathbb{E}_{\widehat{\varphi} \circ \widehat{\phi}} \left[ \sum_{h=1}^{H} r_h \mid \mathcal{E} \right] \mathbb{P}_{\widehat{\varphi} \circ \widehat{\phi}}(\mathcal{E}) + \mathbb{E}_{\widehat{\varphi} \circ \widehat{\phi}} \left[ \sum_{h=1}^{H} r_h \mid \mathcal{E}^c \right] \mathbb{P}_{\widehat{\varphi} \circ \widehat{\phi}}(\mathcal{E}^c)$$

$$\geq \mathbb{E}_{\widehat{\varphi} \circ \widehat{\phi}} \left[ \sum_{h=1}^{H} r_h \mid \mathcal{E} \right] \mathbb{P}_{\widehat{\varphi} \circ \widehat{\phi}}(\mathcal{E})$$

$$\geq \mathbb{E}_{\widehat{\varphi} \circ \widehat{\phi}} \left[ \sum_{h=1}^{H} r_h \mid \mathcal{E} \right] - \mathbb{E}_{\widehat{\varphi} \circ \widehat{\phi}} \left[ \sum_{h=1}^{H} r_h \mid \mathcal{E} \right] H \vartheta$$

$$\geq \mathbb{E}_{\widehat{\varphi} \circ \widehat{\phi}} \left[ \sum_{h=1}^{H} r_h \mid \mathcal{E} \right] - H^2 \vartheta$$

Combining the two upper bounds we get:

$$V(\widehat{\varphi} \circ \alpha \circ \phi^\star) - V(\widehat{\varphi} \circ \widehat{\phi}) \leq \mathbb{E}_{\widehat{\varphi} \circ \widehat{\phi}} \left[ \sum_{h=1}^{H} r_h \mid \mathcal{E} \right] + H^2 \vartheta - \mathbb{E}_{\widehat{\varphi} \circ \widehat{\phi}} \left[ \sum_{h=1}^{H} r_h \mid \mathcal{E} \right] + H^2 \vartheta \leq 2H^2 \vartheta$$

Therefore, with probability at least $1 - \delta - n_{\mathrm{samp}}(S, A, H, \varepsilon, \delta) H \vartheta$, learn a policy $\widehat{\varphi} \circ \widehat{\phi}$ such that:

$$\sup_{\pi \in \Pi} V(\pi) - V(\widehat{\varphi} \circ \widehat{\phi}) \leq \varepsilon + 2H^2 \vartheta.$$

$\square$

**Theorem 7** (Wrapping up the proof.). *Fix $\varepsilon_\circ > 0$ and $\delta_\circ \in (0, 1)$ and let $\mathscr{A}$ be any PAC RL algorithm for tabular MDPs with sample complexity $n_{samp}(S, A, H, \varepsilon, \delta)$. If $n$ satisfies:*

$$n = \mathcal{O}\left( \left\{ \frac{N^4 H^4}{\eta_{min}^4 \beta_{for}^2} + \frac{N^6 H^8}{\varepsilon_\circ^2 \eta_{min}^4 \beta_{for}^2} + \frac{N^6 H^6 n_{samp}^2(S, A, H, \varepsilon_\circ/2, \delta_\circ/4)}{\delta_\circ^2 \eta_{min}^4 \beta_{for}^2} \right\} \ln\left( \frac{|\mathcal{F}||\Phi|}{\delta_\circ} \right) \right),$$

*then forward modeling learns a decoder $\widehat{\phi} : \mathcal{X} \to N$. Further, running $\mathscr{A}$ on the tabular MDP with induced by $\widehat{\phi}$ with hyperparameters $\varepsilon = \varepsilon_\circ/2$, $\delta = \delta_\circ/4$, returns a latent policy $\widehat{\varphi}$. Then there exists a bijective mapping $\alpha : \mathcal{S} \to [|\mathcal{S}|]$ such that with probability at least $1 - \delta$ we have:*

$$\forall s \in \mathcal{S}, \qquad \mathbb{P}_{x \sim q(\cdot | s)} \left( \widehat{\phi}(x) = \alpha(s) \mid \phi^\star(x) = s \right) \geq 1 - \frac{4 N^3 H^2 \Delta}{\eta_{min}^2 \beta_{for}},$$

*and*

$$V(\pi^\star) - V(\widehat{\varphi} \circ \widehat{\phi}) \leq \varepsilon_\circ$$

*Further, the amount of online interactions in the downstream RL is given by $n_{samp}(S, A, H, \varepsilon_\circ/2, \delta_\circ/4)$ and doesn't scale with $\ln |\Phi|$.*

*Proof.* We showed in Theorem 5 that we learn a $\widehat{\phi}$ such that:

$$\mathbb{P}_{x \sim q(\cdot|s)}\left(\widehat{\phi}(x) = \alpha(s) \mid \phi^{\star}(x) = s\right) \geq 1 - \frac{4N^3 H^2 \Delta}{\eta_{\min}^2 \beta_{\text{for}}},$$

provided $\Delta < \frac{\eta_{\min}^2 \beta_{\text{for}}}{N^2 H^2}$.

Let $\vartheta = \frac{4N^3 H^2 \Delta}{\eta_{\min}^2 \beta_{\text{for}}}$. Then from Proposition 6 we learn a $\widehat{\varphi}$ such that:

$$V(\pi^{\star}) - V(\widehat{\varphi} \circ \widehat{\phi}) \leq \varepsilon + 2H^2 \vartheta,$$

with probability at least $1 - \delta - n_{\text{samp}}(S, A, H, \varepsilon, \delta)H\vartheta$. The failure probability $\delta - n_{\text{samp}}(S, A, H, \varepsilon, \delta)H\vartheta$ was when condition in Theorem 5 holds which holds with $\delta$ probability. Hence, total failure probability is:

$$2\delta + n_{\text{samp}}(S, A, H, \varepsilon, \delta)H\vartheta.$$

We set $\delta$ both in our representation learning analysis and in PAC RL to $\delta_\circ/4$. We also set $\varepsilon$ in the PAC RL algorithm to $\varepsilon_\circ/2$. This means the PAC RL algorithm runs for $n_{\text{samp}}(S, A, H, \varepsilon_\circ/2, \delta_\circ/4)$ episodes.

We enforce $\vartheta \leq \frac{\delta_\circ}{2n_{\text{samp}}(S, A, H, \varepsilon_\circ/2, \delta_\circ/4)H}$. Then the total failure probability becomes:

$$2\delta_\circ/4) + \delta_\circ/4 + \delta_\circ/2 \leq \delta_\circ$$

We also enforce $2H^2 \vartheta \leq \varepsilon_\circ/2$. The sub-optimality of the PAC RL policy is given by:

$$\varepsilon_\circ/2 + \varepsilon_\circ/2 \leq \varepsilon_\circ$$

This gives us our derived PAC RL bound.

We now accumulate all conditions:

$$\Delta = \sqrt{\frac{2}{n} \ln\left(\frac{4|\mathcal{F}||\Phi|}{\delta_\circ}\right)}$$

$$\vartheta = \frac{4N^3 H^2 \Delta}{\eta_{\min}^2 \beta_{\text{for}}}$$

$$\Delta < \frac{\eta_{\min}^2 \beta_{\text{for}}}{N^2 H^2}$$

$$\vartheta \leq \frac{\delta_\circ}{2n_{\text{samp}}(S, A, H, \varepsilon_\circ/2, \delta_\circ/4)H}$$

$$2H^2 \vartheta \leq \varepsilon_\circ/2$$

This simplifies to

$$\Delta \leq \frac{\eta_{\min}^2 \beta_{\text{for}}}{N^2 H^2}$$

$$\Delta \leq \frac{\delta_\circ \eta_{\min}^2 \beta_{\text{for}}}{8N^3 H^3 n_{\text{samp}}(S, A, H, \varepsilon_\circ/2, \delta_\circ/4)}$$

$$\Delta \leq \frac{\varepsilon_\circ \eta_{\min}^2 \beta_{\text{for}}}{16N^3 H^4}$$

Or,

$$n = \mathcal{O}\left(\left\{\frac{N^4 H^4}{\eta_{\min}^4 \beta_{\text{for}}^2} + \frac{N^6 H^8}{\varepsilon_\circ^2 \eta_{\min}^4 \beta_{\text{for}}^2} + \frac{N^6 H^6 n_{\text{samp}}^2(S, A, H, \varepsilon_\circ/2, \delta_\circ/4)}{\delta_\circ^2 \eta_{\min}^4 \beta_{\text{for}}^2}\right\} \ln\left(\frac{|\mathcal{F}||\Phi|}{\delta_\circ}\right)\right)$$

This completes the proof. $\square$

## B.2 UPPER BOUND FOR THE TEMPORAL CONTRASTIVE APPROACH

We first convert our video dataset $\mathcal{D}$ into a dataset suitable for contrastive learning. We first split the datasets into $\lfloor n/2 \rfloor$ pairs of videos. For each video pair $\left\{ \left( x_1^{(2l)}, x_2^{(2l)}, \cdots, x_H^{(2k)} \right), \left( x_1^{(2l+1)}, x_2^{(2l+1)}, \cdots, x_H^{(2l+1)} \right) \right\}$, we create a tuple $(x, x', k, z)$ where $z \in \{0, 1\}$ as follows. As in forward modeling, we will either use a fixed value of $k$, or sample $k \in \text{Unf}([K])$. We denote this general distribution over $k$ by $\omega \in \Delta([K])$ which is either point mass, or $\text{Unf}([K])$. We sample $k \sim \omega$ and $z \sim \text{Unf}(\{0, 1\})$ and $h \in \text{Unf}([H])$. We set $x = x_h^{(2l)}$. If $z = 1$, then we set $x' = x_{h+k}^{(2l)}$, otherwise, we sample $h' \sim \text{Unf}(\{0, 1\})$ and select $x' = x_{h'}^{(2l)}$. This way, we collect a dataset $\mathcal{D}_{\text{cont}}$ of $\lfloor n/2 \rfloor$ tuples $(x, k, x', z)$. We view a tuple $(x, k, x', z)$ as a *real observation pair* when $z = 1$, and a *fake observation pair* when $z = 0$. Note that our sampling process leads to all data points being iid.

We define the distribution $D_{\text{cont}}(X, k, X', Z)$ as the distribution over $(x, k, x', z)$. We can express this distribution as:

$$D_{\text{cont}}(X = x, k, X' = x', Z = 1) = \frac{\omega(k)}{2H} \sum_{h=1}^{H} D(x = x_h, x' = x_{h+k})$$

$$= \frac{\omega(k)}{2} \rho(x) D(x_{k+1} = x' \mid x_1 = x)$$

$$D_{\text{cont}}(X = x, X' = x', Z = 0) = \frac{\omega(k)}{2H^2} \sum_{h=1}^{H} D(x = x_h) \sum_{h'=1}^{H} D(x' = x_{h'})$$

$$= \frac{\omega(k)}{2} \rho(x) \rho(x')$$

where we use the time homogeneity of $D$ and definition of $\rho$. We will use a shorthand to denote $D(x_{k+1} = x' \mid x_1 = x)$ as $D(x' \mid x, k)$ in this analysis. It is easy to verify that $D(x' \mid x, k) = D(x' \mid \phi^\star(x), k)$. The marginal distribution $D_{\text{cont}}(x, k, x')$ is given by:

$$D_{\text{cont}}(x, k, x') = \frac{\omega(k)\rho(x)}{2} \left( D(x' \mid x, k) + \rho(x') \right) \tag{14}$$

Note that $D_{\text{cont}}(X)$ is the same as $\rho(X)$.

We will use $D_{\text{cont}}$ for any marginal and conditional distribution derived from $D_{\text{cont}}(X, k, X', Z)$. We assume a model class $\mathcal{G} : \mathcal{X} \times [K] \times \times [N] \rightarrow [0, 1]$ that we use for solving the prediction problem. We will also reuse the decoder class $\phi : \mathcal{X} \rightarrow [N]$ that we defined earlier, and we will assume that $N = |\mathcal{S}|$. This can be relaxed by doing clustering or working with a different induced MDP (e.g., see the clustering algorithm in Misra et al. (2020)). However, this is not the main point of the analysis.

We define the expected risk minimizer of the squared loss problem below:

$$\widehat{g}, \widehat{\phi} = \arg \min_{g \in \mathcal{G}, \phi \in \Phi} \frac{1}{\lfloor n/2 \rfloor} \sum_{i=1}^{\lfloor n/2 \rfloor} \left( g(\phi(x^{(i)}), k^{(i)}, x'^{(i)}) - z^{(i)} \right)^2 \tag{15}$$

We express the Bayes classifier of this problem below:

**Lemma 3** (Bayes Classifier). *The Bayes classifier of the problem posed in Equation 15 is given by $D_{cont}(z = 1 \mid x, k, x')$ which satisfies:*

$$D_{cont}(z = 1 \mid x, k, x') = \frac{D(\phi^\star(x') \mid \phi^\star(x), k)}{D(\phi^\star(x') \mid \phi^\star(x), k) + \rho(\phi^\star(x'))}.$$

*Proof.* We can express the Bayes classifier as:

$$D_{\text{cont}}(z = 1 \mid x, k, x') = \frac{D_{\text{cont}}(x, k, x', z = 1)}{D_{\text{cont}}(x, k, x', z = 1) + D_{\text{cont}}(x, k, x', z = 0)}$$

$$= \frac{\omega(k)/2\rho(x)D(x' \mid x)}{\omega(k)/2\rho(x)D(x' \mid x) + \omega(k)/2\rho(x)\rho(x')}$$

$$= \frac{D(x' \mid x, k)}{D(x' \mid x, k) + \rho(x')}$$

$$= \frac{D(x' \mid \phi^\star(x), k)}{D(x' \mid \phi^\star(x), k) + \rho(x')}$$

$$= \frac{q(x' \mid \phi^\star(x))D(\phi^\star(x') \mid \phi^\star(x), k)}{q(x' \mid \phi^\star(x))D(\phi^\star(x') \mid \phi^\star(x), k) + q(x' \mid \phi^\star(x))\rho(\phi^\star(x'))}$$

$$= \frac{D(\phi^\star(x') \mid \phi^\star(x), k)}{D(\phi^\star(x') \mid \phi^\star(x), k) + \rho(\phi^\star(x'))}.$$

$\square$

**Assumption 8** (Realizability). *There exists $g^\star \in \mathcal{G}$ and $\phi^\circ \in \Phi$ such that for all $(x, k, x') \in$ supp $D_{cont}(X, k, X')$, we have $D_{cont}(z = 1 \mid x, k, x') = g^\star(\phi^\circ(x), k, x')$.*

We will use the shorthand to denote $g^\star(x, k, x') = g^\star(\phi^\circ(x), k, x')$.

As before, we start with typical square loss guarantees in the realizable setting.

**Theorem 8.** *Fix $\delta \in (0, 1)$. Under realizability (Assumption 8), the ERM solution of $\widehat{f}, \widehat{\phi}$ in Eq. (15) satisfies:*

$$\mathbb{E}_{(x,k,x')\sim D_{cont}} \left[ \left( \widehat{g}(\widehat{\phi}(x), k, x') - g^\star(x, k, x') \right)^2 \right] \leq \Delta_{cont}^2 = \frac{2}{n} \ln \frac{|\mathcal{G}| . |\Phi|}{\delta}$$

For proof see Proposition 12 in Misra et al. (2020).

We will prove a coupling result similar to the case for forward modeling. However, to do this, we need to define a coupling distribution:

$$D_{\text{coup}}(X_1 = x_1, X_2 = x_2, k, X' = x') = \omega(k)D_{\text{cont}}(X = x_1)D_{\text{cont}}(X = x_2)D_{\text{cont}}(X' = x')$$

We will derive a useful importance ratio bound.

$$\frac{D_{\text{coup}}(x_1, k, x')}{D_{\text{cont}}(x_1, k, x')} = \frac{2\rho(x_1)\rho(x')}{\rho(x_1)D(x' \mid x_1, k) + \rho(x_1)\rho(x')} \leq 2 \qquad (16)$$

We now prove an analogous result to Proposition 4.

**Theorem 9** (Coupling for Temporal Contrastive Learning). *With probability at least $1 - \delta$ we have:*

$$\mathbb{E}_{(x_1,x_2,k,x')\sim D_{coup}} \left[ \mathbf{1}\left\{ \widehat{\phi}(x_1) = \widehat{\phi}(x_2) \right\} |g^\star(x_1, k, x') - g^\star(x_2, k, x')| \right] < 4\Delta_{cont}(n, \delta)$$

*Proof.* We start with triangle inequality:

$$\mathbb{E}_{(x_1,x_2,k,x')\sim D_{\text{coup}}} \left[ \mathbf{1}\left\{ \widehat{\phi}(x_1) = \widehat{\phi}(x_2) \right\} |g^\star(x_1, k, x') - g^\star(x_2, k, x')| \right]$$

$$\leq \mathbb{E}_{(x_1,x_2,k,x')\sim D_{\text{coup}}} \left[ \mathbf{1}\left\{ \widehat{\phi}(x_1) = \widehat{\phi}(x_2) \right\} \left| g^\star(x_1, k, x') - \widehat{g}(\widehat{\phi}(x_1), k, x') \right| \right] +$$

$$\mathbb{E}_{(x_1,x_2,k,x')\sim D_{\text{coup}}} \left[ \mathbf{1}\left\{ \widehat{\phi}(x_1) = \widehat{\phi}(x_2) \right\} \left| \widehat{g}(\widehat{\phi}(x_1), k, x') - g^\star(x_2, k, x') \right| \right]$$

We bound the first term as:

$$\mathbb{E}_{(x_1,x_2,k,x')\sim D_{\text{coup}}} \left[ \mathbf{1}\left\{ \widehat{\phi}(x_1) = \widehat{\phi}(x_2) \right\} \left| g^\star(x_1, k, x') - \widehat{g}(\widehat{\phi}(x_1), k, x') \right| \right]$$

$$\leq \underbrace{\sqrt{\mathbb{E}_{(x_1,x_2,k,x')\sim D_{\text{coup}}} \left[ \mathbf{1}\left\{ \widehat{\phi}(x_1) = \widehat{\phi}(x_2) \right\} \right]}}_{:=b} \cdot \sqrt{\mathbb{E}_{(x_1,x_2,k,x')\sim D_{\text{coup}}} \left[ \left| g^\star(x_1, k, x') - \widehat{g}(\widehat{\phi}(x_1), k, x') \right|^2 \right]}$$

$$= b\sqrt{\mathbb{E}_{(x_1,k,x')\sim D_{\text{coup}}}\left[\left(g^\star(x_1,k,x') - \widehat{g}(\widehat{\phi}(x_1),k,x')\right)^2\right]}$$

$$= b\sqrt{\mathbb{E}_{(x_1,k,x')\sim D_{\text{cont}}}\left[\frac{D_{\text{coup}}(x_1,k,x')}{D_{\text{cont}}(x_1,k,x')}\left(g^\star(x_1,k,x') - \widehat{g}(\widehat{\phi}(x_1),k,x')\right)^2\right]}$$

$$\leq b\sqrt{2\mathbb{E}_{(x_1,k,x')\sim D_{\text{cont}}}\left[\left(g^\star(x_1,k,x') - \widehat{g}(\widehat{\phi}(x_1),k,x')\right)^2\right]}$$

$$\leq \sqrt{2}b\Delta_{\text{cont}},$$

where we use Cauchy-Schwartz's inequality in the first step and Equation 16 in the second inequality.
The second term is bounded as:

$$\mathbb{E}_{(x_1,x_2,k,x')\sim D_{\text{coup}}}\left[\mathbf{1}\left\{\widehat{\phi}(x_1) = \widehat{\phi}(x_2)\right\}\left|\widehat{g}(\widehat{\phi}(x_1),k,x') - g^\star(x_2,k,x')\right|\right]$$

$$= \mathbb{E}_{(x_1,x_2,k,x')\sim D_{\text{coup}}}\left[\mathbf{1}\left\{\widehat{\phi}(x_1) = \widehat{\phi}(x_2)\right\}\left|\widehat{g}(\widehat{\phi}(x_2),k,x') - g^\star(x_2,k,x')\right|\right]$$

$$= \mathbb{E}_{(x_1,x_2,k,x')\sim D_{\text{coup}}}\left[\mathbf{1}\left\{\widehat{\phi}(x_1) = \widehat{\phi}(x_2)\right\}\left|\widehat{g}(\widehat{\phi}(x_1),k,x') - g^\star(x_1,k,x')\right|\right]$$

$$\leq \sqrt{2}b\Delta_{\text{cont}},$$

where we use the coupling argument in the first step and then reduce it to the first term using symmetric of $(x_1, x_2)$ in $D_{\text{coup}}$. Combining the upper bounds of the two terms and using $b \leq 1$ and $2\sqrt{2} < 4$ completes the proof. $\qquad\square$

**Assumption 9** (Temporal Contrastive Margin). *We assume that there exists a $\beta_{temp} > 0$ such that for any two different states $s_1$ and $s_2$:*

$$\frac{1}{2}\mathbb{E}_{k\sim\omega,s'\sim\rho}\left[|g^\star(s_1,k,s') - g^\star(s_2,k,s')|\right] \geq \beta_{temp}$$

The factor of $\frac{1}{2}$ is chosen for comparison with forward modeling as will become clear later at the end of the proof. As before, if $k$ is fixed, the margin is given by

$$\beta_{\text{temp}}^{(k)} := \frac{1}{2}\inf_{s_1\neq s_2;s_1,s_2\in\mathcal{S}}\mathbb{E}_{s'\sim\rho}\left[|g^\star(s_1,k,s') - g^\star(s_2,k,s')|\right]$$

and when $k \sim \text{Unf}([K])$ the margin is given by

$$\beta_{\text{temp}}^{(u)} := \frac{1}{2}\inf_{s_1\neq s_2;s_1,s_2\in\mathcal{S}}\mathbb{E}_{k\sim\text{Unf}([K]),s'\sim\rho}\left[|g^\star(s_1,k,s') - g^\star(s_2,k,s')|\right]$$

We directly have $\beta_{\text{temp}}^{(u)} \geq \frac{1}{K}\sum_{k=1}^{K}\beta_{\text{temp}}^{(k)}$.

**Lemma 4.**

$$\mathbb{P}_{x_1,x_2\sim\rho}\left(\widehat{\phi}(x_1) = \widehat{\phi}(x_2) \wedge \phi^\star(x_1) \neq \phi^\star(x_2)\right) \leq \frac{2\Delta_{cont}(n,\delta)}{\beta_{temp}}$$

*Proof.* We start with the left-hand side in Theorem 9.

$$\mathbb{E}_{(x_1,k,x_2,x')\sim D_{\text{coup}}}\left[\mathbf{1}\left\{\widehat{\phi}(x_1) = \widehat{\phi}(x_2)\right\}|g^\star(x_1,k,x') - g^\star(x_2,k,x')|\right]$$

$$= \mathbb{E}_{(x_1,x_2)\sim D_{\text{coup}}}\left[\mathbf{1}\left\{\widehat{\phi}(x_1) = \widehat{\phi}(x_2)\right\}\mathbb{E}_{k\sim\omega,x'\sim\rho}\left[|g^\star(x_1,k,x') - g^\star(x_2,k,x')|\right]\right]$$

$$= \mathbb{E}_{(x_1,x_2)\sim\rho}\left[\mathbf{1}\left\{\widehat{\phi}(x_1) = \widehat{\phi}(x_2)\right\}\mathbb{E}_{k\sim\omega,s'\sim\rho}\left[|g^\star(x_1,k,s') - g^\star(x_2,k,s')|\right]\right]$$

$$\geq 2\beta_{\text{temp}}\mathbb{E}_{(x_1,x_2)\sim\rho}\left[\mathbf{1}\left\{\widehat{\phi}(x_1) = \widehat{\phi}(x_2) \wedge \phi^\star(x_1) \neq \phi^\star(x_2)\right\}\right]$$

$$= 2\beta_{\text{temp}}\mathbb{P}_{(x_1,x_2)\sim\rho}\left[\widehat{\phi}(x_1) = \widehat{\phi}(x_2) \wedge \phi^\star(x_1) \neq \phi^\star(x_2)\right],$$

where we use the definition of $\beta_{\text{temp}}$, the fact that marginal over $D_{\text{coup}}(X)$ is $\rho$, and that $g^\star(x,k,x')$ only depends on $\phi^\star(x')$ and $\phi^\star(x)$ (Lemma 3). Combining with the inequality proved in Theorem 9, completes the proof. $\qquad\square$

We have now reduced this analysis to an almost identical one to the forward analysis case (Proposition 4). We can, therefore, use the same steps and derive identical bounds. All what changes is that $\beta_{\text{for}}$ is replaced by $\beta_{\text{temp}}$ and in $\Delta$ we replace $\ln|\mathcal{F}|$ with $\ln|\mathcal{G}|$. At this point, we can clarify that the factor of $\frac{1}{2}$ was chosen in the definition of $\beta_{\text{temp}}$ so that $\beta_{\text{for}}$ can be replaced by $\beta_{\text{temp}}$ rather than $\frac{\beta_{\text{temp}}}{2}$ which will make it harder to compare margins, as we will do later.

## B.3  PROOF OF LOWER BOUND FOR EXOGENOUS BLOCK MDPS

***Proof of Theorem 2.*** We present a hard instance using a family of exogenous block MDPs, with $H = 2$, $\mathcal{A} = \{1, 2\}$, and a single binary endogenous factor and $d - 1$ exogenous binary factors for each level, where each endogenous and exogenous factor. We first fix an absolute constant $p \in [0, 1]$.

Each MDP $M_i$ is indexed by $i \in [d]$, and is specified as follows:

- **State space:** The state is represented by $x_h := [s_h^{(1)}, s_h^{(2)}, \ldots, s_h^{(d)}]$, where the superscript denotes different factors. For MDP $M_i$, only the $i$-th factor $s_h^{(i)}$ is an endogenous state for all $h$, and the other factors are exogenous. Each factor has values of $\{0, 1\}$.

- **Transition:** For the MDP instance $M_i$: it has

  1. For the $i$-th factor (endogenous factor), $\mathbb{P}(s_2^{(i)} \mid s_1^{(i)}, a) = \mathbb{1}[s_2^{(i)} = \mathbb{1}(s_1^{(i)} = a)]$. That is, the endogenous states have deterministic dynamics. If $s_1^{(i)} = a$, then it transitions to $s_2^{(i)} = 1$, otherwise it transitions to $s_2^{(i)} = 0$.

  2. For the $j$-th factor with $j \neq i$ (exogenous factor), $\mathbb{P}(s_2^{(j)} \mid s_1^{(j)}) = (1 - p)\mathbb{1}(s_2^{(j)} = s_1^{(j)}) + p\mathbb{1}(s_2^{(j)} \neq s_1^{(j)})$ for any $s_2^{(j)}$ and $s_1^{(j)}$. That is, the $j$-th factor has probability of $1 - p$ of transiting to the same state (i.e., $s_1^{(j)} = 0 \to s_2^{(j)} = 0$ or $s_1^{(j)} = 1 \to s_2^{(j)} = 1$), and probability of $p$ of transiting to the different state (i.e., $s_1^{(j)} = 0 \to s_2^{(j)} = 1$ or $s_1^{(j)} = 1 \to s_2^{(j)} = 0$).

  Note that the MDP terminates at $h = 2$.

- **Initial state distribution and reward:** The marginal distribution of $s_1^{(j)}$ is uniformly distributed at random over $\{0, 1\}$ for all $j \in [d]$, and all factors are independent from each other. For MDP $M_i$, the agent only receive reward signal after taking action at $h = 2$, with $R(s_2^{(i)}, a) = s_2^{(i)}$. That is, it always reward 1 at $s_2^{(i)} = 1$ and reward 0 at $s_2^{(i)} = 0$ no matter which action it takes.

- **Data collection policy for video data:** We assume that the data collection policy always pick action 0 with probability $p$ and action 1 with probability $1 - p$ for all states.

Now we use the following two steps to establish the proof.

**Uninformative video data for learning the state decoder**  Since video data only contains state information, from the MDP family construction above, we can easily verify that all MDP instances in such a family will have an identical video data distribution, *regardless of the choice of constant $p$*. This implies that the video data is uninformative for the agent to distinguish the MDP instance from the MDP family. Now, we assume $\mathcal{D}^{(i)}$ is the video data from the instance $M^{(i)}$, and $\phi^{(i)}$ is the state decoder learned from an arbitrary algorithm $\mathscr{A}_1$ with $\mathcal{D}^{(i)}$. Then, for any arbitrary algorithm $\mathscr{A}_2$ that uses the state decoder $\phi^{(i)}$ in its execution, it is equivalent to such an $\mathscr{A}_2$ that uses the state decoder $\phi^{(j)}$ in its execution, where $j$ can be selected arbitrarily from $[d]$.

**State decoder requiring exponential length**  Without loss of generality, we further restrict the state decoder $\phi$ used in the execution of $\mathscr{A}_2$ for all MDP instance to be some $\phi_h : \mathcal{X} \to [L]$, where $h \in \{1, 2\}$ and $L \leq 2^d$. Then we will argue that there must exists a $k \in [d]$, such that

$$\sum_{x_1, \widetilde{x}_1 \in \mathcal{X}} \mathbb{P}\left(\phi_1(x_1) = \phi_1(\widetilde{x}_1) \vee \left(s_1^{(k)} \neq \widetilde{s}_1^{(k)}\right)\right) > \frac{2^d - L}{d2^d}, \tag{17}$$

where $x_1 := [s_1^{(1)}, s_1^{(2)}, \ldots, s_1^{(d)}]$ and $\widetilde{x}_1 := [\widetilde{s}_1^{(1)}, \widetilde{s}_1^{(2)}, \ldots, \widetilde{s}_1^{(d)}]$. Note that, Eq. (17) means there must be a probability of at least $2^d{-}L/d2^d$ that $\phi_1$ will incorrectly group two different $s_1^{(k)}$ together.

We now prove Eq. (17). Based on the construct above, we know that $|\mathcal{X}| = 2^d$, and each state in $\mathcal{X}$ has the same occupancy for $x_1$ based on the defined initial state distribution (this holds for all instances in the MDP family, as we are now only talking about the initial state $x_1$). Thus, we have

$$\sum_{x_1, \widetilde{x}_1 \in \mathcal{X}} \mathbb{P}\left[\phi_1(x_1) = \phi_1(\widetilde{x}_1) \vee \left(s_1^{(1)} = \widetilde{s}_1^{(1)}\right) \vee \left(s_1^{(2)} = \widetilde{s}_1^{(2)}\right) \vee \cdots \vee \left(s_1^{(d)} = \widetilde{s}_1^{(d)}\right)\right] \leq \frac{L}{2^d}, \quad (18)$$

because we defined $\phi_1 : \mathcal{X} \to [L]$, it means that such $\phi_1$ is only able to distinguish the number of $L$ different states from $\mathcal{X}$. Then, we obtain

$$\sum_{j \in [d]} \sum_{x_1, \widetilde{x}_1 \in \mathcal{X}} \mathbb{P}\left(\phi_1(x_1) = \phi_1(\widetilde{x}_1) \vee \left(s_1^{(j)} \neq \widetilde{s}_1^{(j)}\right)\right)$$

$$= \sum_{x_1, \widetilde{x}_1 \in \mathcal{X}} \mathbb{P}\left(\phi_1(x_1) = \phi_1(\widetilde{x}_1)\right)$$

$$- \sum_{x_1, \widetilde{x}_1 \in \mathcal{X}} \mathbb{P}\left[\phi_1(x_1) = \phi_1(\widetilde{x}_1) \vee \left(s_1^{(1)} = \widetilde{s}_1^{(1)}\right) \vee \left(s_1^{(2)} = \widetilde{s}_1^{(2)}\right) \vee \cdots \vee \left(s_1^{(d)} = \widetilde{s}_1^{(d)}\right)\right]$$

$$= \frac{2^d - L}{2^d}. \qquad \text{(by Eq. (18))}$$

$$\implies \max_{j \in [d]} \sum_{x_1, \widetilde{x}_1 \in \mathcal{X}} \mathbb{P}\left(\phi_1(x_1) = \phi_1(\widetilde{x}_1) \vee \left(s_1^{(j)} \neq \widetilde{s}_1^{(j)}\right)\right) > \frac{2^d - L}{d2^d}.$$

So this proves Eq. (17).

From Eq. (17), we know that for the MDP instance $M^{(k)}$, $\phi_1$ will have probability at least $2^d{-}L/2 \cdot d2^d$ to mistake the endogenous state, which implies that for any policy that is represented using the state decoder $\phi$, it must have sub-optimality at least $2^d{-}L/2 \cdot d2^d$. Therefore, it is easy to verify that, for any $\varepsilon > 0$, we can simply pick $d = 1/4\varepsilon$, and obtain

$$\text{sub-optimality} > \frac{2^d - L}{2 \cdot d2^d} \geq \varepsilon, \quad \forall L \leq 2^{1/4\varepsilon - 1}.$$

Then, any arbitrary algorithm $\mathscr{A}_2$ that uses the state decoder $\phi$ in its execution, where $\phi_h : \mathcal{X} \to [L]$ can be chosen arbitrarily for $h \in \{1, 2\}$ and $L \leq 2^{1/4\varepsilon - 1}$, must have sub-optimality larger than $\varepsilon$.

**Additional characteristics of MDP family and video data**  Note that, by combining the arguments of uninformative video data and a state decoder requiring exponential length, we obtain impossible results. We now discuss the following:

1. The margin condition defined in Assumption 3 regarding the constructed MDPs

2. The PAC learnability of the constructed MDPs

3. The coverage condition of video data.

For the defined margin condition of forward modeling, we have: for the MDP instance $M_i$ with constant $p$, we can bound the forward margin as below ($\mathbb{P}_{\text{for}}$ denotes the video distribution)

$$\left\|\mathbb{P}_{\text{for}}(X_2 \mid s_1^{(i)} = 0) - \mathbb{P}_{\text{for}}(X_2 \mid s_1^{(i)} = 1)\right\|_{\text{TV}}$$

$$= \frac{1}{2} \sum_{X_2} \left|\mathbb{P}_{\text{for}}(X_2 \mid s_1^{(i)} = 0) - \mathbb{P}_{\text{for}}(X_2 \mid s_1^{(i)} = 1)\right|$$

$$= \frac{1}{2} \sum_{X_2} \left|\mathbb{P}_{\text{for}}(s_2^{(i)} = 0 \mid s_1^{(i)} = 0)\mathbb{P}(X_2 \mid s_2^{(i)} = 0) + \mathbb{P}_{\text{for}}(s_2^{(i)} = 1 \mid s_1^{(i)} = 0)\mathbb{P}(X_2 \mid s_2^{(i)} = 1)\right.$$

$$\left. - \mathbb{P}_{\text{for}}(s_2^{(i)} = 0 \mid s_1^{(i)} = 1)\mathbb{P}(X_2 \mid s_2^{(i)} = 0) + \mathbb{P}_{\text{for}}(s_2^{(i)} = 1 \mid s_1^{(i)} = 1)\mathbb{P}(X_2 \mid s_2^{(i)} = 1)\right|$$

$$= \frac{1}{2} \sum_{X_2} \left| (1 - 2p) \left[ \mathbb{P}(X_2 \mid s_2^{(i)} = 0) - \mathbb{P}(X_2 \mid s_2^{(i)} = 1) \right] \right|.$$

$$\stackrel{(a)}{=} \frac{|1 - 2p|}{2} \sum_{X_2} \mathbb{P}(X_2 \mid s_2^{(i)} = 0) + \frac{|1 - 2p|}{2} \sum_{X_2} \mathbb{P}(X_2 \mid s_2^{(i)} = 1)$$

$$= |1 - 2p|,$$

where step (a) is because $s_2^{(i)}$ is a part of $X_2$, and then we know $\mathbb{P}(X_2 \mid s_2^{(i)} = 0)$ and $\mathbb{P}(X_2 \mid s_2^{(i)} = 1)$ cannot be nonzero simultaneously. So picking $p \neq 0.5$ implies positive forward margin.

For the temporal contrastive learning, it is easy to verify that $|\mathbb{P}_{\text{for}}(z = 1 \mid s_1^{(i)} = 1, X_2) - \mathbb{P}_{\text{for}}(z = 1 \mid s_1^{(i)} = 1, X_2)| = |1 - 2p|$, so picking $p \neq 0.5$ also implies positive margin for temporal contrastive learning.

As for the PAC learnability, since the latent dynamics of our constructed MDPs are deterministic, they are provably PAC learnable by Efroni et al. (2022).

As for the coverage property of the video data, it is easy to verify

$$\max_{\pi \in \Pi, x_1 \in \mathcal{X}} \frac{\mathbb{P}_\pi(x_1, a_1)}{\mathbb{P}_{\text{for}}(x_1, a_1)} = \max_{\pi \in \Pi, x_2 \in \mathcal{X}} \frac{\mathbb{P}_\pi(x_2)}{\mathbb{P}_{\text{for}}(x_2)} = \max \left\{ 1/p, 1/1-p \right\}.$$

Therefore, we can simply pick $p = 1/3$ and obtain the desired MDP and video data properties. This completes the proof. $\square$

**Addition remark of Theorem 2** In the proof of Theorem 2, if we pick $p = 0.5$ for that hard instance, the constructed MDP family reduces to a block MDP without exogenous noise, but the margin becomes 0 for both forward modeling and temporal contrastive learning. Therefore, it implies that either the exogenous noise or zero forward margin could make the learnability of the problem impossible.

### B.4 CAN WE GET EFFICIENT LEARNING UNDER ADDITIONAL ASSUMPTIONS?

Our lower bound suggests that one can in general not learn efficient and correct representations with just video data. However, it may be possible in some cases to do so with an additional assumption. We highlight one example here and defer a proper formal analysis to future work. One path to success is when the gold decoder results in the best-in-class error. A domain where this can happen is when the endogenous state is more predictive of $x'$ than any other $\ln |\mathcal{S}|$ bits of information in $x$. E.g., in a navigation domain, there can be many sources of noise in the background, but memorizing all of them can easily overwhelm the decoder's model capacity. Instead focusing solely on modeling the agent's state can simplify the task of predicting the future.

Recently some approaches have also considered recovering *latent actions* from video data using an encoder-decoder approach (Ye et al., 2022). In general, the lower bound in Theorem 2 applies to these methods and they do not provably work in the hard instances with exogenous noise. For example, the latent actions can capture *exogenous noise* instead of actions, if the former is more predictive of changes in the observations. However, in simpler cases such as 3D games, where the agent's action is typically most predictive of changes in observations, or in settings with no exogenous noise, one can expect these approaches to do well.

### B.5 RELATION BETWEEN MARGINS

We defined margins $\beta_{\text{for}}$ for forward modeling and $\beta_{\text{temp}}$ for temporal contrastive learning. The larger the values of these margins, the more easy it is to separate observations from different endogenous states. This can be directly inferred from the sample complexity bounds which scale inversely with these margins. In particular, both $\beta_{\text{for}}$ and $\beta_{\text{temp}}$ depend on the way we sample the multi-step variable $k$. We consider two special cases: one where $k \in [K]$ is fixed, we instantiate these margins as $\beta_{\text{for}}^{(k)}$ and $\beta_{\text{temp}}^{(k)}$, and second where $k$ is uniformly sampled from $[K]$ and we instantiate those margins as $\beta_{\text{for}}^{(u)}$ and $\beta_{\text{temp}}^{(u)}$.

A natural question is how these margins are related. The sample complexity bounds of forward modeling and temporal contrastive are almost identical except for the difference in margins ($\beta_{\text{for}}$ vs $\beta_{\text{temp}}$) and the function classes ($\mathcal{F}$ vs $\mathcal{G}$). If the function classes were of similar complexity, then having a larger margin will make it easier to learn the right representation.[2]

**Theorem 10** (Margin Relation). *For any Block MDP and $K \in \mathbb{N}$, the margins $\beta_{for}^{(k)}, \beta_{for}^{(u)}, \beta_{temp}^{(k)}, \beta_{temp}^{(u)} > 0$ are related as:*

$$\frac{1}{K}\beta_{for}^{(k)} \le \beta_{for}^{(u)}$$

$$\frac{1}{K}\beta_{temp}^{(k)} \le \beta_{temp}^{(u)}$$

$$\frac{\eta_{min}^2}{4H^2}\beta_{for}^{(k)} \le \beta_{temp}^{(k)} \le \beta_{for}^{(k)}$$

$$\frac{\eta_{min}^2}{4H^2}\beta_{for}^{(u)} \le \beta_{temp}^{(u)} \le \beta_{for}^{(u)}.$$

*Proof.* We first prove the first two relations. Fix any $k \in [K]$ then,

$$\beta_{\text{for}}^{(u)} = \inf_{s_1 \neq s_2, s_1, s_2 \in \mathcal{S}} \mathbb{E}_{k' \sim \text{Unf}([K])} \left[ \|D_{pr}(X' \mid s_1, k') - D_{pr}(X' \mid s_2, k')\|_{\text{TV}} \right],$$

$$\ge \frac{1}{K} \sum_{k'=1}^K \inf_{s_1 \neq s_2, s_1, s_2 \in \mathcal{S}} \|D_{pr}(X' \mid s_1, k') - D_{pr}(X' \mid s_2, k')\|_{\text{TV}},$$

$$\ge \frac{1}{K} \inf_{s_1 \neq s_2, s_1, s_2 \in \mathcal{S}} \|D_{pr}(X' \mid s_1, k) - D_{pr}(X' \mid s_2, k)\|_{\text{TV}},$$

$$= \frac{1}{K}\beta_{\text{for}}^{(k)}.$$

Similarly,

$$\beta_{\text{temp}}^{(u)} = \frac{1}{2} \inf_{s_1 \neq s_2, s_1, s_2 \in \mathcal{S}} \mathbb{E}_{k' \sim \text{Unf}([K]), s' \sim \rho} \left[ |g^\star(s_1, k', s') - g^\star(s_2, k', s')| \right],$$

$$\ge \frac{1}{2K} \sum_{k'=1}^K \inf_{s_1 \neq s_2, s_1, s_2 \in \mathcal{S}} \mathbb{E}_{s' \sim \rho} \left[ |g^\star(s_1, k', s') - g^\star(s_2, k', s')| \right],$$

$$\ge \frac{1}{2K} \inf_{s_1 \neq s_2, s_1, s_2 \in \mathcal{S}} \mathbb{E}_{s' \sim \rho} \left[ |g^\star(s_1, k, s') - g^\star(s_2, k, s')| \right],$$

$$= \frac{1}{K}\beta_{\text{temp}}^{(k)}.$$

We now prove the next two relations. We will prove these bounds for a generic distribution $\omega \in \Delta([K])$ over $k$. Recall that $\omega$ is point-mass over $k$ for $\beta_{\text{temp}}^{(k)}$ and $\text{Unf}([K])$ for $\beta_{\text{temp}}^{(u)}$. We denote our generic margins as $\beta_{\text{for}}$ and $\beta_{\text{temp}}$ for $k \sim \omega$. We use a shorthand notation $W_k(s, s') = \frac{\rho(s')}{D_{pr}(s' \mid s, k) + \rho(s')}$ for a given pair of states $s, s'$ and integer $k \in [K]$. It is easy to see that $W_k(s, s') \le 1$ as $D_{pr}(s' \mid s, k), \rho(s') \in (0, 1]$. Further, we have $W_k(s, s') \ge \frac{\rho(s')}{2} \ge \frac{\eta_{\min}}{2H}$ where we use $D_{pr}(s' \mid s, k), \rho(s') \in (0, 1]$, and [Equation 8](#).

We have $g^\star(s, k, s') = D_{\text{cont}}(z = 1 \mid s, k, s') = g^\star(s, k, s') = \frac{D_{pr}(s' \mid s, k)}{D_{pr}(s' \mid s, k) + \rho(s')}$ using the definition of $D_{\text{cont}}$ in [Lemma 3](#) and [Assumption 8](#). We can use the shorthand $W_k$ and the definition of $g^\star$ to show

$$\beta_{\text{temp}} = \frac{1}{2} \inf_{s_1 \neq s_2, s_1, s_2 \in \mathcal{S}} \mathbb{E}_{k \sim \omega, s' \sim \rho} \left[ |g^\star(s_1, k, s') - g^\star(s_2, k, s')| \right],$$

$$= \frac{1}{2} \inf_{s_1 \neq s_2, s_1, s_2 \in \mathcal{S}} \sum_{k=1}^K \omega(k) \sum_{s' \in \mathcal{S}} \rho(s') |g^\star(s_1, k, s') - g^\star(s_2, k, s')|,$$

---

[2]This inference has to be made with a caveat that since we are comparing upper bounds, we cannot guarantee this to hold.

$$= \frac{1}{2} \inf_{s_1 \neq s_2, s_1, s_2 \in \mathcal{S}} \sum_{k=1}^{K} \omega(k) \sum_{s' \in \mathcal{S}} W_k(s_1, s') W_k(s_2, s') \left| D_{pr}(s' \mid s_1, k) - D_{pr}(s' \mid s_2, k) \right|. \tag{19}$$

As $W_k(s_1, s') \leq 1$ and $W_k(s_2, s') \leq 1$ we have

$$\beta_{\text{for}} = \frac{1}{2} \inf_{s_1 \neq s_2, s_1, s_2 \in \mathcal{S}} \sum_{k=1}^{K} \omega(k) \sum_{s' \in \mathcal{S}} \underbrace{W_k(s_1, s')}_{\leq 1} \underbrace{W_k(s_2, s')}_{\leq 1} \left| D_{pr}(s' \mid s_1, k) - D_{pr}(s' \mid s_2, k) \right|,$$

$$\leq \frac{1}{2} \inf_{s_1 \neq s_2, s_1, s_2 \in \mathcal{S}} \sum_{k=1}^{K} \omega(k) \sum_{s' \in \mathcal{S}} \left| D_{pr}(s' \mid s_1, k) - D_{pr}(s' \mid s_2, k) \right|,$$

$$= \inf_{s_1 \neq s_2, s_1, s_2 \in \mathcal{S}} \mathbb{E}_{k \sim \omega} \left[ \| D_{pr}(s' \mid s_1, k) - D_{pr}(s' \mid s_2, k) \|_{\text{TV}} \right]$$

$$= \beta_{\text{for}}.$$

This gives us $\beta_{\text{temp}}^{(k)} \leq \beta_{\text{for}}^{(k)}$ and $\beta_{\text{temp}}^{(u)} \leq \beta_{\text{for}}^{(u)}$. Finally, we prove the lower bounds. Starting from Equation 19 and using $W_k(s_1, s') \geq \frac{\eta_{\min}}{2H}$ and $W_k(s_2, s') \leq \frac{\eta_{\min}}{2H}$ we get the following:

$$\beta_{\text{for}} = \frac{1}{2} \inf_{s_1 \neq s_2, s_1, s_2 \in \mathcal{S}} \sum_{k=1}^{K} \omega(k) \sum_{s' \in \mathcal{S}} \underbrace{W_k(s_1, s')}_{\geq \eta_{\min}/2H} \underbrace{W_k(s_2, s')}_{\geq \eta_{\min}/2H} \left| D_{pr}(s' \mid s_1, k) - D_{pr}(s' \mid s_2, k) \right|,$$

$$\geq \frac{\eta_{\min}^2}{4H^2} \cdot \frac{1}{2} \inf_{s_1 \neq s_2, s_1, s_2 \in \mathcal{S}} \sum_{k=1}^{K} \omega(k) \sum_{s' \in \mathcal{S}} \left| D_{pr}(s' \mid s_1, k) - D_{pr}(s' \mid s_2, k) \right|,$$

$$= \frac{\eta_{\min}^2}{4H^2} \inf_{s_1 \neq s_2, s_1, s_2 \in \mathcal{S}} \mathbb{E}_{k \sim \omega} \left[ \| D_{pr}(s' \mid s_1, k) - D_{pr}(s' \mid s_2, k) \|_{\text{TV}} \right]$$

$$= \frac{\eta_{\min}^2}{4H^2} \beta_{\text{for}}.$$

This gives us $\beta_{\text{temp}}^{(k)} \geq \frac{\eta_{\min}^2}{4H^2} \beta_{\text{for}}^{(k)}$ and $\beta_{\text{temp}}^{(u)} \geq \frac{\eta_{\min}^2}{4H^2} \beta_{\text{for}}^{(u)}$ which completes the proof. $\qquad \square$

The main finding of the above theorem is that forward modeling has a higher margin than temporal contrastive learning. However, typically the function class used for forward modeling has a higher statistical complexity than those for temporal contrastive learning as the latter is solving a simpler binary classification problem than generating an observation.

### B.6 WHY TEMPORAL CONTRASTIVE LEARNING IS MORE SUSCEPTIBLE TO EXOGENOUS NOISE THAN FORWARD MODELING

Theorem 2 shows that in the presence of exogenous noise, no video-based representation learning approach can be efficient in the worst case. However, this result only presents a worst-case analysis. In this section, we show an instance-dependent analysis. The main finding is that the temporal contrastive approach is very susceptible to even the smallest amount of exogenous noise, while forward modeling is more robust to the presence of exogenous noise. However, both approaches fail when there is a significant amount of exogenous noise, consistent with Theorem 2.

**Problem Instance.** We consider a Block MDP with exogenous noise with a state space of $\mathcal{S} = \{0, 1\}$, action space of $\mathcal{A} = \{0, 1\}$ and exogenous noise space of $\xi = \{0, 1\}$. We consider $H = 1$ with a uniform distribution over $s_1$ and $\xi_1$, i.e., the start state $s_1$ and the start exogenous noise variable $\xi_1$ are chosen uniformly from $\{0, 1\}$. The transition dynamics are deterministic and given as follows: given action $a_1 \in \{0, 1\}$ and state $s_1 \in \{0, 1\}$, we deterministically transition to $s_2 = 1 - s_1$ if $s_1 = a_1$, otherwise, we remain in $s_2 = s_1$. The exogenous noise variable deterministically transitions from $\xi_1$ to $\xi_2 = 1 - \xi_1$. The reward function is given by $R(s_2, s_1) = \mathbf{1}\{s_2 = s_1\}$. We use the indicator notation $\mathbf{1}\{\mathcal{E}\}$ to denote 1 if the condition $\mathcal{E}$ is true and 0 otherwise. The

observation space is given by $\mathcal{X} = \{0,1\}^{m+2}$ where $(m+2)$ is the dimension of observation space. Given the endogenous state $s$ and exogenous noise $\xi$, the environment generates an observation stochasticaly as $x = [\xi, v_1, \cdots, v_l, w_1, \cdots, w_{m-l}, s]$ where $v_i \sim p_{\text{samp}}(\cdot \mid \xi)$ and $w_j \sim p_{\text{samp}}(\cdot \mid s)$ for all $i \in [l]$ and $j \in [m-l]$. The distribution $p_{\text{samp}}(u \mid s)$ generates $u = s$ with a probability 0.8 and $u = 1-s$ with a probability 0.2. The hyperparameter $l$ is a fixed integer controlling what portion of the observation is generated by the exogenous noise compared to the endogenous state. If $l = 1$, we only have a small amount of exogenous noise, while if $l = m - 1$ we have the maximal amount of exogenous noise. The state $s$ and exogenous noise $\xi$ are both decodable from the observation $x$. The optimal policy achieves a return of 1 and takes action $a_1 = 1$ if $s_1 = 0$ and $a_1 = 0$ if $s_1 = 1$. As the optimal policy depends on the value of $s_1$, we must learn the latent state to realize the optimal policy.

**Learning Setting.** We assume a decoder class $\Phi = \{\phi^\star, \phi_\xi^\star\}$ consisting of the true decoder $\phi^\star$ and the incorrect decoder $\phi_\xi^\star$ which maps observation to the exogenous noise $\xi$. Both decoders take an observation and map it to a value in $\{0,1\}$. We assume access to an arbitrarily large dataset $\mathcal{D}$ consisting of tuples $(x_1, x_2)$ collecting iid using a fixed data policy $\pi_{\text{data}}$. This policy takes action $a_1 = 0$ in $s_1 = 0$ and action $a_1 = 1$ in $s_1 = 1$. Let $D(x_1, x_2)$ be the data distribution induced by $\pi_{\text{data}}$. We will use $D$ to define other distributions induced by $D(x_1, x_2)$, for example $D(x_2)$ or $D(s_2)$. We also assume access to two model classes $\mathcal{F} : \{0,1\} \to \Delta(\mathcal{X})$ and $\mathcal{G} : \{0,1\}^2 \to [0,1]$. We assume these model classes are finite and contain certain constructions that we define later.

**Overview:** As we increase the value of $l$, the amount of exogenous noise in the environment increases. We will prove that irrespective of the value of $l$, temporal contrastive learning assigns the same loss for both the correct decoder $\phi^\star$ and the incorrect decoder $\phi_\xi^\star$. In contrast, the forward modeling approach is able to prefer $\phi^\star$ over $\phi_\xi^\star$ when the noise is limited, specifically, when $l < m/2$. This will establish that temporal contrastive is very susceptible to exogenous noise whereas forward modeling is more robust. However, both approaches provably fail when there is $l \geq m/2$.

As we have $H = 1$, we will denote $x_2, s_2, \xi_2$ by $x', s', \xi'$ and $x_1, s_1, \xi_1$ by $x, s, \xi$ respectively. Note that unless specified otherwise, $s$ and $\xi$ are the endogenous state and exogenous noise of the observation $x$. Similarly, $s'$ and $\xi'$ are the endogenous state and exogenous noise of $x'$. We will also use a shorthand $q(x')$ to denote the emission probability $q(x' \mid \phi_\xi^\star(x'), \phi^\star(x'))$ given its endogenous state and exogenous noise. We first state the conditional data distribution $D(x' \mid x)$.

$$D(x' \mid x) = q(x') T_\xi(\xi' \mid \xi) \sum_{a \in \mathcal{A}} T(s' \mid s, a) \pi_{\text{data}}(a \mid s),$$
$$= q(x') \mathbf{1}\{\xi' = 1 - \xi\} \mathbf{1}\{s' = 1 - s\}, \tag{20}$$

where we use $T_\xi(\xi' \mid \xi) = \mathbf{1}\{\xi' = 1 - \xi\}$ and $\sum_{a \in \mathcal{A}} T(s' \mid s, a) \pi_{\text{data}}(a \mid s) = \mathbf{1}\{s' = 1 - s\}$ which follows from the definition of $\pi_{\text{data}}$. Note that $D(x' \mid x)$ only depends on $x$ via $s, \xi$, therefore, we can define $D(x' \mid x) = D(x' \mid s, \xi)$.

Let $\tilde{x}$ be an observation variable with endogenous state $\tilde{s}$ and exogenous noise $\tilde{\xi}$, i.e., $\tilde{s} = \phi^\star(\tilde{x})$ and $\tilde{\xi} = \phi_\xi^\star(\tilde{x})$. We use this to derive the marginal data distribution $\rho$ over $x'$ as follows:

$$\rho(x') = \sum_{s,\xi \in \{0,1\}} D(x', s, \xi) = \sum_{s,\xi \in \{0,1\}} D(x' \mid s, \xi) \mu(s) \mu_\xi(\xi),$$
$$= \frac{q(x')}{4} \sum_{s,\xi \in \{0,1\}} \mathbf{1}\{\xi' = 1 - \xi\} \mathbf{1}\{s' = 1 - s\},$$
$$= \frac{q(x')}{4}, \tag{21}$$

where in the second step uses the fact that $\mu$ and $\mu_\xi$ are uniform and Eq. (20). We are now ready to prove our desired result.

**Temporal contrastive learning cannot distinguish between good and bad decoder for all $l \in [m-1]$.** We first recall that temporal contrastive learning approach use the given observed data $(x_1, x_2)$ to compute a set of real and fake observation tuples. This is collected into a dataset $(x, x', z)$

where $z = 1$ indicates that $(x_1 = x, x_2 = x')$ was observed in the dataset, and $z = 0$ indicates that $(x_1 = x, x_2 = x')$ was not observed, or is an imposter. We sample $z$ uniformly in $\{0, 1\}$. The fake data is constructed by take $x = x_1$ from one tuple and $x' = x_2$ from another observed tuple. We start by computing the optimal Bayes classifier for the temporal contrastive learning approach using the definition of Bayes classifier in Lemma 3.

$$D_{\text{cont}}(z = 1 \mid x, x') = \frac{D(x' \mid x)}{D(x' \mid x) + \rho(x')} = \frac{\mathbf{1}\{s' = 1 - s\}\mathbf{1}\{\xi' = 1 - \xi\}}{\mathbf{1}\{s' = 1 - s\}\mathbf{1}\{\xi' = 1 - \xi\} + 1/4},$$

where we use Lemma 3 in the first step and Eqs. (20) and (21) in the second step. Recall that $z = 1$ denotes whether a given observation tuple $(x, x')$ is real rather than an imposter/false. Note that since we have $k = 1$, as it is a $H = 1$ problem, we drop the notation $k$ from all terms.

The marginal distribution over $(x, x')$ for the temporal contrastive is given by Eq. (14) which in our case instantiates to:

$$D_{\text{cont}}(x, x') = \frac{D(x)}{2} \left\{ D(x' \mid x) + \rho(x') \right\},$$

$$= \frac{1}{8} q(x') q(x) \left\{ \mathbf{1}\{s' = 1 - s\}\mathbf{1}\{\xi' = 1 - \xi\} + 1/4 \right\}, \tag{22}$$

where we use Eqs. (20) and (21), and $D(x) = q(x)\mu(s)\mu_\xi(\xi) = {q(x)}/{4}$.

Let $g \in \mathcal{G}$ be any classifier head. Given a decoder $\phi$, we define $g \circ \phi : (x, x') \mapsto g(\phi(x), \phi(x'))$ as a model for temporal contrastive learning, with an expected contrastive loss of:

$\ell_{\text{cont}}(g, \phi^\star)$

$= \mathbb{E}_{(x,x') \sim D_{\text{cont}}, z \sim D_{\text{cont}}(\cdot | x, x')} \left[ \left( z - g\left( \phi^\star(x), \phi^\star(x') \right) \right)^2 \right]$

$= \mathbb{E}_{(x,x') \sim D_{\text{cont}}} \left[ D_{\text{cont}}(z = 1 \mid x, x') \left( 1 - 2g\left( \phi^\star(x), \phi^\star(x') \right) \right) + g\left( \phi^\star(x), \phi^\star(x') \right)^2 \right]$

$= \frac{1}{8} \sum_{s, \xi, s', \xi'} \left\{ \mathbf{1}\{s' = 1 - s\}\mathbf{1}\{\xi' = 1 - \xi\} + \frac{1}{4} \right\} \left( \frac{\mathbf{1}\{s' = 1 - s\}\mathbf{1}\{\xi' = 1 - \xi\}}{\mathbf{1}\{s' = 1 - s\}\mathbf{1}\{\xi' = 1 - \xi\} + \frac{1}{4}} (1 - g(s, s')) + g(s, s')^2 \right)$

Similarly, the expected temporal contrastive loss of the model $g \circ \phi^\star$ with the bad decoder $\phi_\xi^\star$ is given by:

$\ell_{\text{cont}}(g, \phi_\xi^\star)$

$= \mathbb{E}_{(x,x') \sim D_{\text{cont}}, z \sim D_{\text{cont}}(\cdot | x, x')} \left[ \left( z - g\left( \phi_\xi^\star(x), \phi_\xi^\star(x') \right) \right)^2 \right]$

$= \frac{1}{8} \sum_{s, \xi, s', \xi'} \left\{ \mathbf{1}\{s' = 1 - s\}\mathbf{1}\{\xi' = 1 - \xi\} + \frac{1}{4} \right\} \left( \frac{\mathbf{1}\{s' = 1 - s\}\mathbf{1}\{\xi' = 1 - \xi\}}{\mathbf{1}\{s' = 1 - s\}\mathbf{1}\{\xi' = 1 - \xi\} + \frac{1}{4}} (1 - g(\xi, \xi')) + g(\xi, \xi')^2 \right)$

Note that by interchanging $s$ with $\xi$ and $s'$ with $\xi'$, we can show $\ell_{\text{cont}}(g, \phi_\xi^\star) = \ell_{\text{cont}}(g, \phi^\star)$. Therefore, $\inf_{g \in \mathcal{G}} \ell_{\text{cont}}(g, \phi_\xi^\star) = \inf_{g \in \mathcal{G}} \ell_{\text{cont}}(g, \phi^\star)$. This implies that for any value of $l$, the temporal contrastive loss assigns the same loss to the good decoder $\phi^\star$ and the bad decoder $\phi_\xi^\star$. Hence, in practice, temporal contrastive cannot distinguish between the good and bad decoder and may converge to the latter leading to poor downstream performance. This convergence to the bad decoder may happen if it is easier to overfit to noise. For example, in our gridworld example, it is possibly easier for the model to overfit to the predictable motion of noise than understand the underlying dynamics of the agent. This is observed in Figure 3 where the representation learned via temporal contrastive tends to overfit to the noisy exogenous pixels and perform poorly on downstream RL tasks (Figure 2).

**Forward modeling learns the good decoder if $l < \lfloor m/2 \rfloor$.** We likewise analyze the expected forward modeling loss of the good and bad decoder. For any $f \in \mathcal{F}$, we have $f(x' \mid u)$ as the generator head that acts on a given decoder's output $u \in \{0, 1\}$ and generates the next observation $x'$.

If we use the good decoder $\phi^\star$, then we cannot predict the exogenous noise $\xi$ or $\xi'$ which can be either 0 or 1 with equal probability. This implies that for the $l$ noisy bits $v_1, \cdots, v_l$ in $x'$, the best

prediction is that each one has an equal probability of taking 0 or 1. To see this, fix $i \in [l]$ and recall that $\mathbb{P}(v_i = \xi' \mid \xi') = 0.8$ and $\mathbb{P}(v_i = 1 - \xi' \mid \xi') = 0.2$. As $\xi'$ has equal probability of taking value 0 or 1, therefore, $\mathbb{P}(v_i = u) = \sum_{\xi' \in \{0,1\}} \mathbb{P}(v_i = u \mid \xi')^{1/2} = \frac{0.8+0.2}{2} = 0.5$. However, since we can deterministically predict $s'$, therefore, we can predict the true distribution over $w_j$ for all $j \in [m - l]$. Let $f_{\text{good}}$ be this generator head. Formally, we have:

$$f_{\text{good}}(x' \mid \phi^\star(x)) = \underbrace{(1/2)}_{\text{due to } x'_1 = \xi'} \cdot \underbrace{(1/2)^l}_{\text{due to } v_{1:l}} \cdot \underbrace{\prod_{j=l+2}^{m+1} p_{\text{samp}}(x'_j \mid 1 - \phi^\star(x))}_{\text{due to } w_{1:m-l}} \cdot \underbrace{\mathbf{1}\{x'_{m+2} = 1 - \phi^\star(x)\}}_{\text{due to } x'_{m+2} = s'}$$

The Bayes distribution is given by:

$$D(x' \mid x)$$
$$= q(x') \cdot \mathbf{1}\{\phi^\star(x') = 1 - \phi^\star(x)\} \cdot \mathbf{1}\{\phi^\star_\xi(x') = 1 - \phi^\star_\xi(x)\}$$
$$= \mathbf{1}\{x'_1 = 1 - \phi^\star_\xi(x)\} \cdot \prod_{i=1}^{l} p_{\text{samp}}(x'_{i+1} \mid 1 - \phi^\star_\xi(x)) \cdot \prod_{j=l+2}^{m+1} p_{\text{samp}}(x'_j \mid 1 - \phi^\star(x)) \mathbf{1}\{x'_{m+2} = 1 - \phi^\star(x)\}.$$

As we are optimizing the log-loss, we look at the expected KL divergence $\ell_{kl}$ between the $D(x' \mid x)$ and $f_{\text{good}}(x' \mid \phi^\star(x))$ which gives:

$$\ell_{kl}(f_{\text{good}}, \phi^\star)$$
$$= \mathbb{E}_x \left[ \sum_{x'} D(x' \mid x) \ln \frac{D(x' \mid x)}{f_{\text{good}}(x' \mid \phi^\star(x))} \right]$$
$$= \mathbb{E}_x \left[ \sum_{x'} D(x' \mid x) \ln \frac{\mathbf{1}\{x'_1 = 1 - \phi^\star_\xi(x)\} \cdot \prod_{i=1}^{l} p_{\text{samp}}(x'_{i+1} \mid 1 - \phi^\star_\xi(x))}{(1/2)^{l+1}} \right]$$
$$= (l+1)\ln(2) + \mathbb{E}_x \left[ \sum_{x'} D(x' \mid x) \ln \left( \mathbf{1}\{x'_1 = 1 - \phi^\star_\xi(x)\} \cdot \prod_{i=1}^{l} p_{\text{samp}}(x'_{i+1} \mid 1 - \phi^\star_\xi(x)) \right) \right]$$
$$= (l+1)\ln(2) + \mathbb{E}_x \left[ \sum_{i=1}^{l} \sum_{x'_{i+1} \in \{0,1\}} p_{\text{samp}}(x'_{i+1} \mid 1 - \phi^\star(x)) \ln p_{\text{samp}}(x'_{i+1} \mid 1 - \phi^\star(x)) \right]$$
$$= (l+1)\ln(2) - lH(p_{\text{samp}}),$$

where $H(p_{\text{samp}})$ denotes the conditional entropy given by $-1/2 \sum_{s \in \{0,1\}} \sum_{v \in \{0,1\}} p_{\text{samp}}(v \mid s) \ln p_{\text{samp}}(v \mid s)$. As $p_{\text{samp}}(u \mid u) = 0.8$ and $p_{\text{samp}}(1 - u \mid u) = 0.2$, we have $H(p_{\text{samp}}) = -0.8 \ln(0.8) - 0.2 \ln(0.2) \approx 0.500$. Plugging this in, we get $\ell_{kl}(f_{\text{good}}, \phi^\star) = l \ln(2) - 0.5l + \ln(2) = \ln(2) + 0.193l$.

Finally, the analysis when we use the $\phi^\star_\xi$ decoder is identical to above. In this case, we can predict $\phi^\star_\xi(x')$ and correctly predict the $p_{\text{samp}}$ distribution over all the $l$-noisy bits $v_{1:l}$. However, for the $w_{1:m-l}$ bits and the $x'[m+2]$, our best bet is to predict a uniform distribution. We capture this by the generator $f_{\text{bad}}$ which gives:

$$f_{\text{bad}}(x' \mid \phi^\star(x)) = \underbrace{(1/2)}_{\text{due to } x'_{m+2} = s'} \cdot \underbrace{(1/2)^{m-l}}_{\text{due to } w_{1:m-l}} \cdot \underbrace{\prod_{i=2}^{l+1} p_{\text{samp}}(x'_i \mid 1 - \phi^\star_\xi(x))}_{\text{due to } v_{1:l}} \cdot \underbrace{\mathbf{1}\{x'_1 = 1 - \phi^\star_\xi(x)\}}_{\text{due to } x'_1 = \xi'}$$

The expected KL loss $\ell_{kl}(f_{\text{bad}}, \phi^\star_\xi)$ can be computed almost exactly as before and is equal to $\ln(2) + 0.193(m - l)$. We can see that for $\ell_{kl}(f_{\text{good}}, \phi^\star) < \ell_{kl}(f_{\text{bad}}, \phi^\star_\xi)$ we must have $\ln(2) + 0.193l < \ln(2) + 0.193(m - l)$, or equivalently, $l < m/2$. This completes the analysis.

| Hyperparameter | Value |
|---|---|
| batch size | 128 |
| learning rate | 0.001 |
| epochs | 400 |
| # of exogenous variables | 10 |
| exogenous pixel size | 4 |
| # of VQ heads | 2 |
| VQ codebook size | 100 |
| VQ codebook temperature | 0 |
| VQ codebook dimension | 32 |
| VQ bottleneck dimension | 1024 |

Table 2: Hyperparameters used for experiments with the GridWorld and ViZDoom domains.

## C  ADDITIONAL EXPERIMENTAL DETAILS

### C.1  DETAILS OF EXPERIMENTAL SETUP

All results are reported with mean and standard error computed over 3 seeds. All the code for this work was run on A100, V100, P40 GPUs, with a compute time of approx. 12 hours for grid world experiments and 6 hours for ViZDoom experiments. Data collection for gridworld was done using a mixture of random walks, optimal trajectories, deviation from optimal trajectories, and walks to randomly chosen goal positions. Data collection for Vizdoom was done via pretrained PPO policies along with random walks for diversity in the observation space.

**GridWorld Details.** We consider navigation in a $12 \times 12$ Minigrid environment (Chevalier-Boisvert et al., 2023). The agent is represented as a red triangle and can take three actions: move forward, turn left, and turn right (Figure 3). The agent needs to reach a yellow key. The position of the agent and key randomizes each episode. The agent only observes an area around itself (as an agent-centric-view). Horizon $H = 12$, and the agent gets a reward of +1.0 for reaching the goal and -0.01 in other cases.

**ViZDoom Defend The Center Details.** We test with a ViZDoom environment called Defend the Center (Wydmuch et al., 2018; Kempka et al., 2016), which is a first-person shooting game (Figure 5). The map is a large circle. A player is spawned in the exact center. 5 monsters are spawned along the wall. Monsters are killed after a single shot. After dying, each monster is respawned after some time. The episode ends when the player dies. The reward scheme is as follows: +1 for killing a monsterand -1 for death.

**Hyperparameters.**   In Table 2, we report the hyperaparameter values used for experiments in this work with the GridWorld and ViZDoom environments.

### C.2  RESULTS ON AN ADDITIONAL DOMAIN

**ViZDoom Basic.**  We use an additional basic ViZDoom environment (Wydmuch et al., 2018; Kempka et al., 2016), which is a first-person shooting game (Figure 8). The player needs to kill a monster to win. The map of the environment is a rectangle with gray walls, ceiling, and floor. The player is spawned along the longer wall in the center. A red, circular monster is spawned randomly somewhere along the opposite wall. The player can take one of three actions at each time step (left, right, shoot). One hit is enough to kill the monster. The episode finishes when the monster is killed or on timeout. The reward scheme is as follows: +101 for shooting the enemy, -1 per time step, and -5 for missed shots. Results for this environment are shown in Figure 7 and Figure 8 and further validate our findings from theory and experiments.

### C.3  ADDITIONAL ABLATIONS

**Harder Exogenous Noise.**   Figure 6 showed the results when we increase the size of the exogenous noise variables (diamond shapes overlayed on the image) in the gridworld domain while keeping the

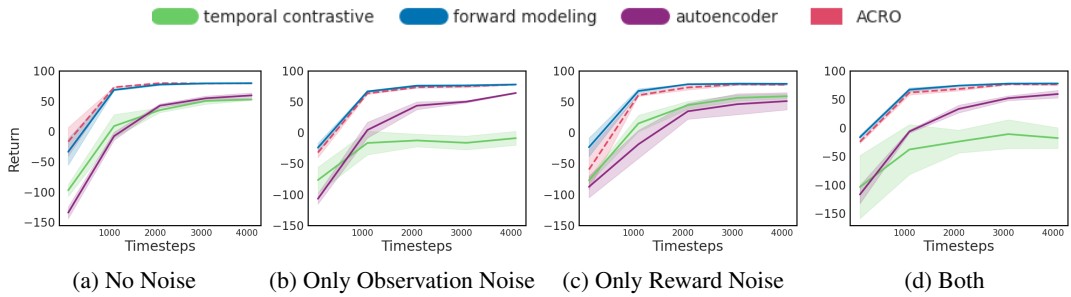

(a) No Noise    (b) Only Observation Noise    (c) Only Reward Noise    (d) Both

Figure 7: RL experiments using different latent representations for the ViZDoom environment.

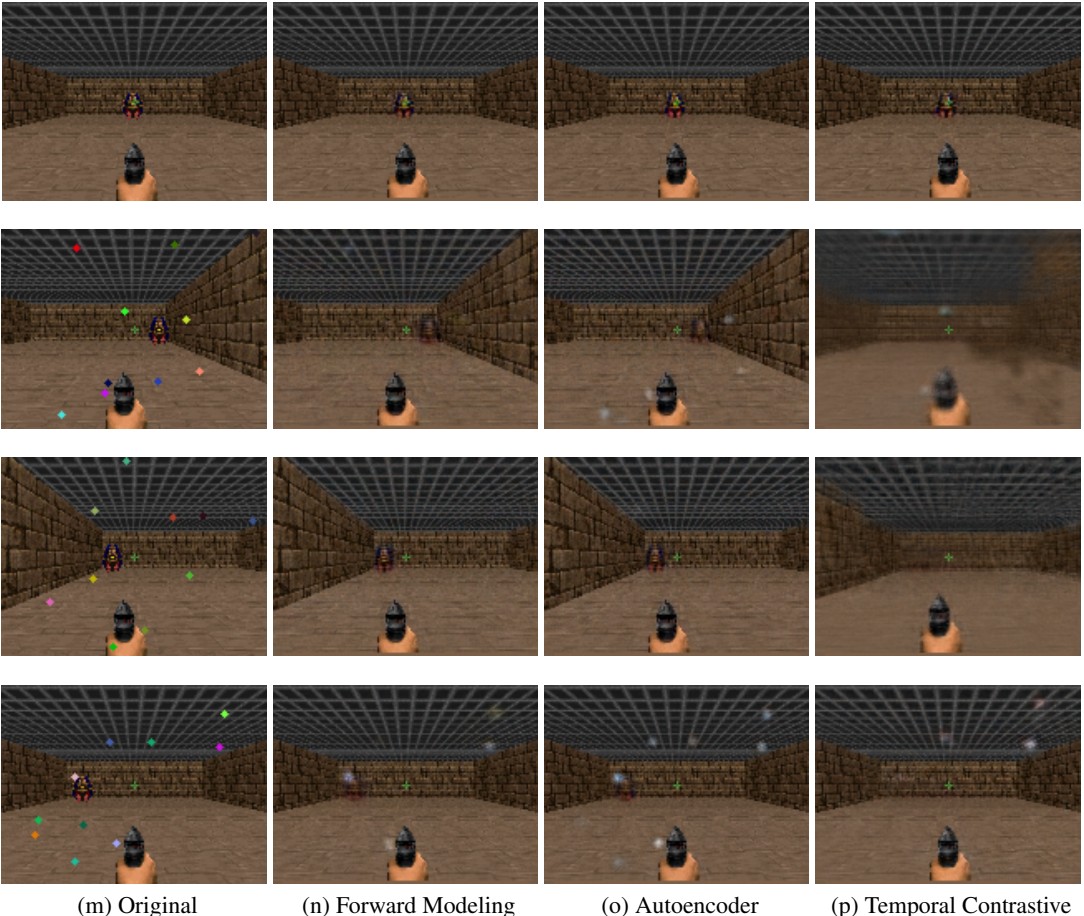

(m) Original    (n) Forward Modeling    (o) Autoencoder    (p) Temporal Contrastive

Figure 8: Decoded image reconstructions from different latent representation learning methods in the ViZDoom environment. We train a decoder on top of frozen representations trained with the three video pre-training approaches.

number of exogenous variables fixed at 10. We also increase the number of exogenous noise variables in the gridworld domain, while keeping their sizes fixed at 4 pixels and present the results in Figure 9. Both results show significant degradation in the performance of video-based representation learning methods whereas ACRO which uses trajectory data continues to perform well. This supports one of our main theoretical results that exogenous noise poses a challenge for video-based representation learning.

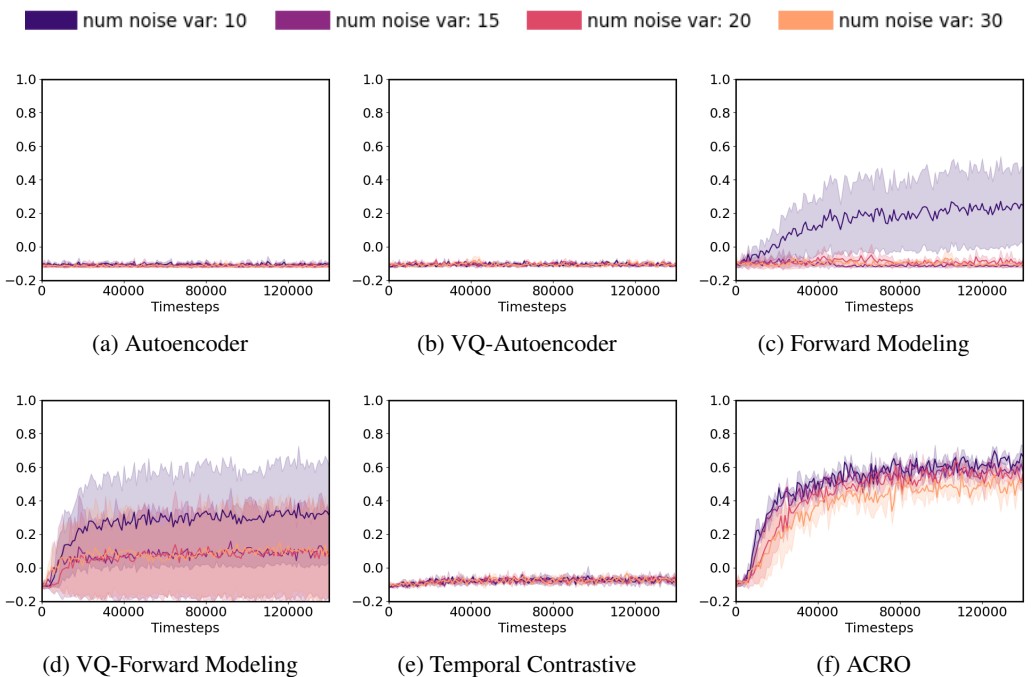

(a) Autoencoder      (b) VQ-Autoencoder      (c) Forward Modeling

(d) VQ-Forward Modeling      (e) Temporal Contrastive      (f) ACRO

Figure 9: Gridworld experiments with exogenous noise of size 4 and different the number of exogenous noise variables. Several video-based representation learning methods struggle to learn as the number of exogenous noise variables increases, whereas ACRO which uses trajectory data, still performs well.

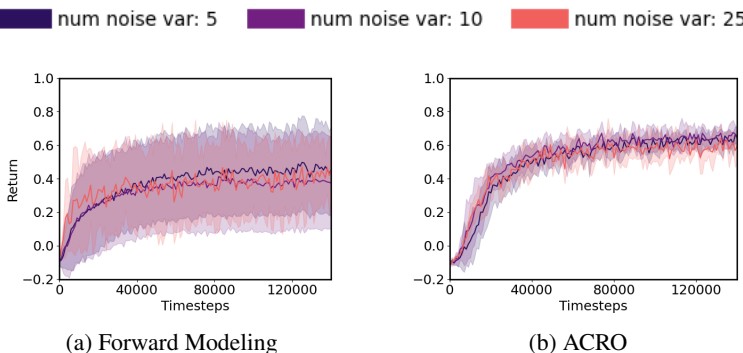

(a) Forward Modeling      (b) ACRO

Figure 10: Experiments with iid noise for the Gridworld environment. 'Num noise var' denotes the number of noisy diamonds constituting the exogenous noise.

**I.I.D. Noise in Gridworld.** We evaluate iid noise in the gridworld domain. We use the diamond-shaped exogenous noise that we used in Figure 2, however, at each time step, we randomly sample the color and position of each diamond, independent of the agent's history. Figure 10(a) shows the result for forward modeling and Figure 10(b) shows the same for ACRO. We also ablate the number of noisy diamonds. As expected, forward modeling and ACRO can learn a good policy while the increase in the number of noisy diamonds (num noise var) only slightly decreases their performance.

**I.I.D. Noise in the Basic ViZDoom environment.** We evaluate the representation learning methods on the basic ViZDoom domain but with independent and identically distributed (iid) noise. We add iid Gaussian noise to each pixel sampled from a 0 mean Gaussian distribution with a standard deviation of 0.001. Based on theory, we expect temporal contrastive objectives to be substantially better at filtering out Gaussian iid noise, which is validated experimentally for the basic ViZDoom

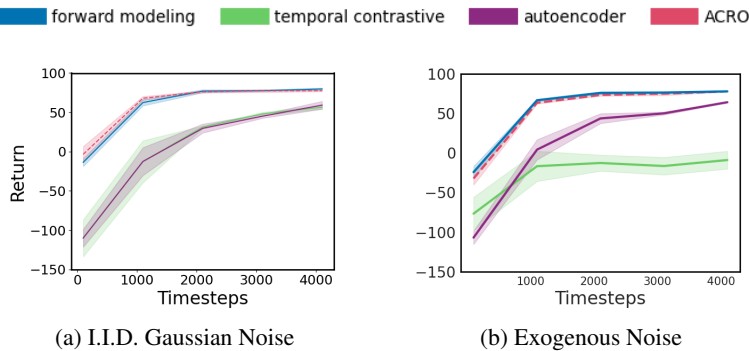

(a) I.I.D. Gaussian Noise       (b) Exogenous Noise

Figure 11: Experiments with (a) Guassian iid noise for the ViZDoom environment and (b) exogenous noise.

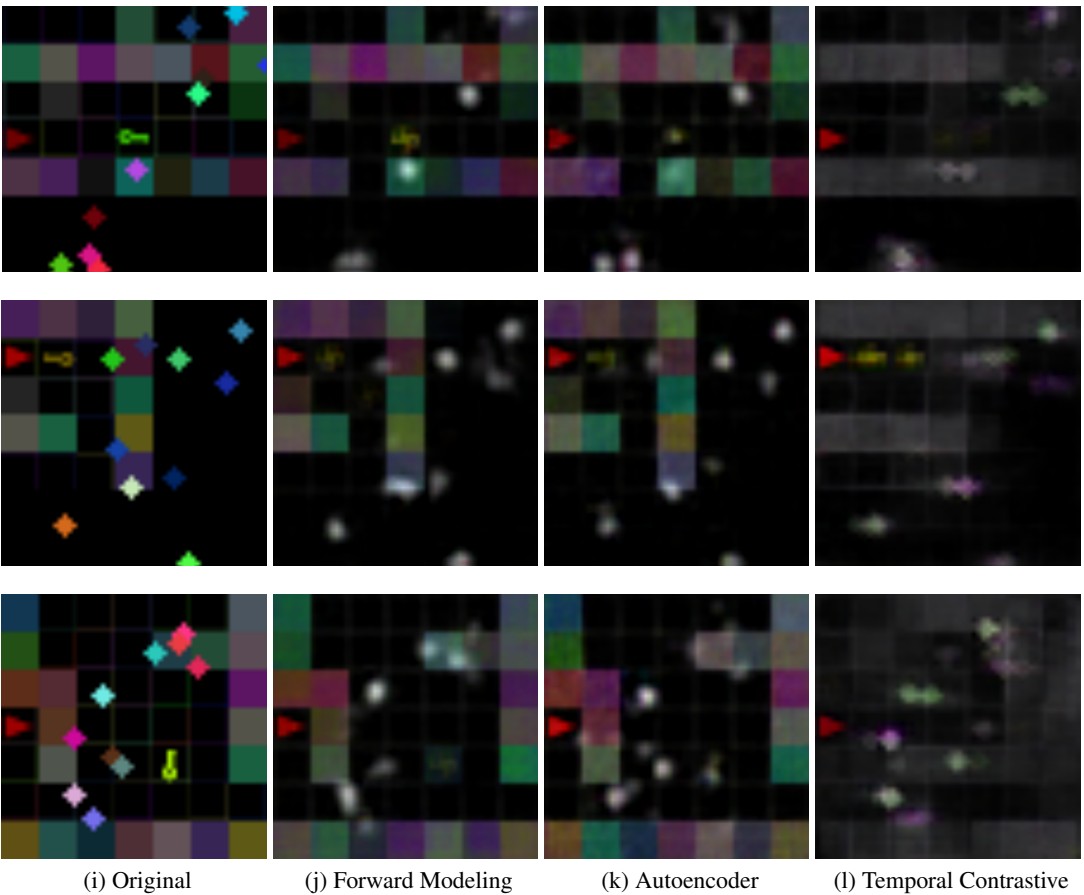

(i) Original     (j) Forward Modeling     (k) Autoencoder     (l) Temporal Contrastive

Figure 12: Decoded image reconstructions from different latent representation learning methods in the GridWorld environment. We train a decoder on top of frozen representations trained with the three video pre-training approaches.

Environment (Figure 11(a)). Figure 11(b) refers to the basic ViZDoom result for convenient comparison.

**Additional reconstructions.** We show additional image reconstructions Figure 12 for the GridWorld environment and in Figure 13 for the ViZDoom Defend the Center environment. We highlight

that important parts of the observation space are recovered successfully by the forward modeling approach under varying levels of exogenous noise, whereas temporal contrastive learning often fails.

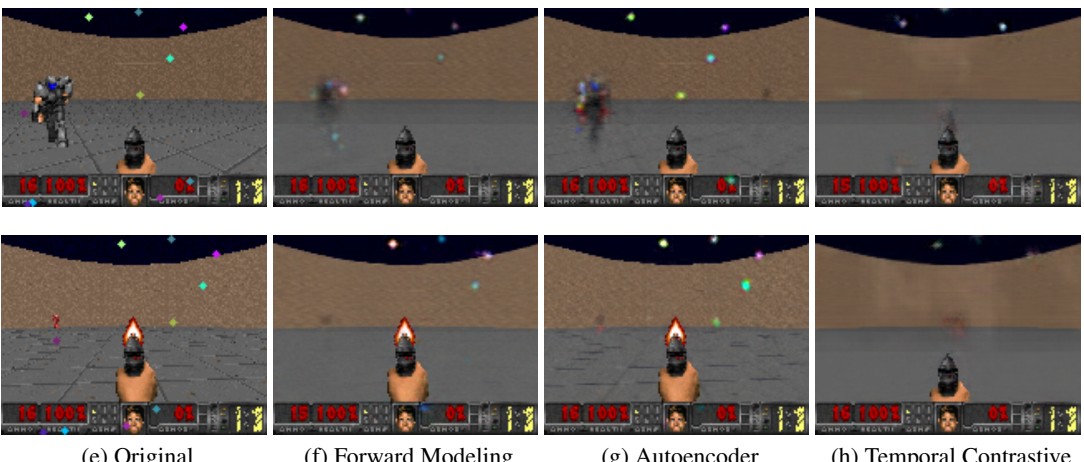

  (e) Original   (f) Forward Modeling  (g) Autoencoder  (h) Temporal Contrastive

Figure 13: Decoded image reconstructions from different latent representation learning methods in the ViZDoom Defend the Center environment. We train a decoder on top of frozen representations trained with the three video pre-training approaches.

