# OpenReview forum: "Towards Principled Representation Learning from Videos for Reinforcement Learning"
_ICLR.cc/2024/Conference — ICLR 2024 spotlight_

### Official Review · Reviewer_ZrZp · 2023-10-30

**Soundness:** 4 excellent
**Presentation:** 3 good
**Contribution:** 4 excellent
**Rating:** 8
**Confidence:** 4

**Summary:**

This study offers a theoretical analysis of representation learning from video-based observations without labeled actions. The problem is formulated as a block Markov Decision Process with exogenous noise. Endogenous states are acquired through an encoder trained with one of three methods: a temporal contrastive loss, a single-state reconstruction loss (autoencoder), or a future state reconstruction loss.

The primary contribution of this paper is the establishment of a theorem that sets an upper bound on representation learning for future state prediction and the contrastive learning approach without noise. It also provides a lower bound in the presence of exogenous noise for these approaches, indicating that agents cannot distinguish between exogenous and endogenous noise. These results are further validated through experiments conducted in both a grid world and a visual environment. In cases without exogenous noise, representation learning proves successful, but it fails in its presence.

**Strengths:**

- The paper is well-written, and the proofs are clear and concise.
- The results address a significant and novel problem, namely, representation learning from noisy video-based data, which is of great interest in the current Reinforcement Learning (RL) community.
- The theoretical results quantitatively address a major challenge in current methods.
- The empirical results align well with the theoretical findings.
- The study explores multiple approaches for representation learning.
- Assumptions are thoroughly explained and justified.

In conclusion, I believe this work should be accepted, as it offers significant and relevant insights to the action-free RL research community. The paper's strengths and contributions make it a valuable addition to the field.

**Weaknesses:**

- While the paper is strong in many aspects, it would be beneficial to expand the experimental evaluation to a wider variety of environments to further validate the results.
- Minor errors, such as unclosed brackets in equations under Assumption 3, should be corrected for clarity and correctness.

**Questions:**

Could we see some additional experiments in the revision?

---

> ### Author Response · Authors · 2023-11-22
> **Additional Experiments and Fixed the Typos**
>
> We thank the reviewer for their helpful feedback and encouraging remarks.
>
> > While the paper is strong in many aspects, it would be beneficial to expand the experimental evaluation to a wider variety of environments to further validate the results.- Could we see some additional experiments in the revision?
>
> **We have added multiple additional experiments and ablations in the revision.** Firstly, we have added results using IID noise (see Figure 7). Secondly, we have added results for a **new Vizdoom environment called Defend the Center** (see Figure 9). Thirdly, we have added ablations where we evaluate the representation learning methods in the presence of an increasing number of exogenous noise variables (Figure 8). These results further buttress the findings of our paper. All these figures and the experimental details are in Appendix C.
>
> The IID experiments show the difference between exogenous noise and IID noise with the former being more harmful to video-based methods as predicted by our theory. Evaluations with an increasing number of exogenous noise allow us to show that video-based representation learning methods quickly fail as the exogenous noise increases whereas ACRO which is a trajectory-based representation learning method keeps working well. This is also consistent with our theoretical findings.
>
> We have also added a new analysis that shows why temporal contrastive learning is especially more susceptible to failure even in the presence of a small amount of exogenous noise, compared to forward modeling. See Appendix B.4 for a proof.
>
> > Minor errors, such as unclosed brackets in equations under Assumption 3, should be corrected for clarity and correctness.
>
> Thanks for a lot pointing out these typos. We have corrected the typos mentioned in the review.

---

### Official Review · Reviewer_TsKD · 2023-10-31

**Soundness:** 2 fair
**Presentation:** 3 good
**Contribution:** 2 fair
**Rating:** 5
**Confidence:** 3

**Summary:**

The paper provides theoretical and empirical results for representations learning for decision-making using video data (without explicit knowledge of the actions). Two settings are studied: one where there is iid noise in the observation, and a setting where there is "exogenous noise, which is non-iid noise that is temporally correlated, such as the motion of people or cars in the background". Three techniques are compared: autoencoding, temporal contrastive learning, and forward modeling. Theoretical and empirical results are provided.

**Strengths:**

- Interesting research questions that can have a big scientific impact
- overall well-written
- experiments follow good practice

**Weaknesses:**

- Some parts of the text are not clear/accurate (see the remarks and questions below in the "questions" section). There are also typos, e.g. "(...) temporal contrastive learning is probne to fail (...)"
- It is unclear how the key messages that are supposed to come from the theorems are actually deduced (see questions below).

**Questions:**

Unclarities in the text:
- The abstract mentions "We evaluate these representational learning methods in two visual domains, proving our theoretical findings." Empirical evaluation can never prove theoretical results except if it looks at all possible cases for instance. In general, it can only illustrate them.
- Beginning of Section 3, it is mentioned that "Our goal is to learn a decoder $\phi : X \rightarrow [N]$ that learns information in the underlying endogenous state $\phi^*(x)$ while throwing away as much irrelevant information as possible.". I don't understand this sentence. Isn't it an encoder that is learnt? What is an endogenous state and what is $phi^*$?
- "ACRO achieves optimal performance across all tasks.": do the authors mean better than other algorithms instead of optimal? (the optimal is not exactly reached and also basically not known.

Theorems
- What is $\alpha$ in Theorem 1?
- For Theorem 1, unless I'm mistaken, the only discussion that is directly about the theorem mentions "These upper bound provide the desired result which shows that not only can we learn the right representation and near-optimal policy but also do without the online episodes scaling with ln |Φ|." How can that interpretation be made from the theorem?
- For theorem 2, it also unclear how the interpretations can be deduced from the theorem itself.

---

> ### Author Response · Authors · 2023-11-22
> **Major clarification, Explaining the setup and theorems, and Writing Improvements**
>
> We thank the reviewer for their feedback. We address the major concerns raised by the reviewer below:
>
> > ...it is mentioned that "Our goal is to learn a decoder  that learns information in the underlying endogenous state  while throwing away as much irrelevant information as possible.". I don't understand this sentence. Isn't it an encoder that is learnt? What is an endogenous state and what is phi*?
>
> **Major Clarification:** We want to learn a representation using video data that can be used to define policies for downstream RL. In real-world problems, there are 3 important variables: observation $(x)$ which is what is received by the agent, and which is generated by two underlying latent variables: endogenous state  $s$ and a noise variable $\xi$ (which can be either IID or exogenous).  The endogenous state $s$ captures information that can be modified by the agent or that affects the agent’s dynamics. Whereas, exogenous noise $\xi$ captures information that neither affects the agent nor is affected by the agent’s action.
>
> For example, consider camera-based robot navigation. Then, $x$ is the image generated by the camera. The endogenous state $s$ contains the position of the robot and nearby obstacles. Whereas, exogenous noise $\xi$ can be the motion of trees in the background or changes in sunlight.
>
> *We assume there is an _unknown_ endogenous state decoder $\phi^\star$ that maps $x$ to $s = \phi^\star(x)$.* As the dynamics of the agent, and reward is controlled by the endogenous state, a useful and parsimonous representation will capture the endogenous state $s$ of the observation $x$ and ignore the noise in $x$. Ideally, we would want our decoder $\phi$ to be equivalent to $\phi^\star$ up to some permutation of the labels of $\phi^\star$. If the representation contains noise or it doesn't fully capture $s$, then it will hurt the downstream RL task. We show this empirically where reconstructions in Fig 3 and Fig 5 show what is captured by the representation and Fig 2 and Fig 4 show how these representations behave on downstream RL tasks.
>
> The word encoder and decoder were interchangeably used. The idea was that even though we are encoding the observation, we are trying to learn this encoder to decode the latent endogenous state. However, to avoid confusion *we have revised the paper to consistently use the word decoder.*
>
> > What is alpha in Theorem 1?
>
> **Alpha:** Representation learning methods can learn the underlying state only up to some permutation of the state labels. The alpha is a bijection that maps the true states to the abstract state/representation learned by the decoder. We have added a description of it in the paper.
>
> > For Theorem 1..."These upper bound provide the desired result which shows that not only can we learn the right representation and near-optimal policy but also do without the online episodes scaling with ln |Φ|." How can that interpretation be made from the theorem?
>
> **Clarifying Theorem 1:** Firstly, Theorem 1 shows that we learn the right representation, i.e., the learned decoder $\hat{\phi}(x)$ is same as a relabeling of the true state $\phi^\star(s)$ with high probability. Secondly, Theorem 1 shows that the learned policy $\hat{\pi} \circ \hat{\phi}$ is near-optimal as $V(\pi^\star) - V(\hat{\pi}\circ\hat{\phi}) < \epsilon$. Lastly, Theorem 1 states that the number of RL episodes scales as $O(S, A, H, \epsilon_\circ, \delta_\circ)$ and so there is no dependence on $\ln |\Phi|$. Note that RL episodes are the only online episodes, as the video data is provided in offline manner. **Therefore, the online episodes do not scale with $\ln |\Phi|$ and this is how we can make that interpretation from Theorem 1.**
>
> > For theorem 2, it also unclear how the interpretations can be deduced from the theorem itself.
>
> **Clarifying Theorem 2:** Theorem 2 states that for any algorithm $\mathscr{A}_1$ that learns a decoder from video data and for any downstream RL algorithm $\mathscr{A}_2$, there exists an MDP $M$ with exogenous noise (or with IID noise but no margin) where no matter how much video data is given to $\mathscr{A}_1$ or how many online episodes are used by $\mathscr{A}_2$, we cannot learn a near-optimal policy. *This is because Theorem 2 states that we have $V_M(\pi^\star) - V_M(\hat{\pi}) > \epsilon$ for any given $\epsilon>0$.*, i.e., the value of the policy $\hat{\pi}$ is more than $\epsilon$ smaller than the optimal policy. **Therefore, the learned policy $\hat{\pi}$ is not near-optimal.** Crucially, this holds for all choices of algorithm $\mathscr{A}_1$ and algorithm $\mathscr{A}_2$ and holds even when an infinite amount of data is given.
>
> Please let us know if you need any further clarification on the Theoretical statements.
>
> **Typos and writing:** We thank you for the useful writing suggestions and for pointing out typos. We have revised the paper to fix these. Specifically, we have revised the paper to say ACRO performs better instead of optimally.

---

> > ### Comment · Reviewer_TsKD · 2023-11-22
> > **Some elements are clarified but the main concerns remain**
> >
> > Thanks for the improvements to the paper.
> >
> > However, there are still many elements that are unclear. For instance:
> > - In Theorem 1, $\alpha$ is now introduced as a "bijection mapping". Why do yo need this one to one mapping?
> > - In theorem 1, what does it mean that $\phi^*(x)=s$? Does it mean that you have access to the true state?
> >
> > In addition, the interpretation and the contribution provided by the theorems are still unclear.
> >
> > As mentioned by the other reviewers, the experiments also have some remaining flows and altogether I maintain my score of 5.

---

> > > ### Author Response · Authors · 2023-11-22
> > > **Important Clarification: Our method does not assume access to the true states or true decoder**
> > >
> > > We thank you for your quick reply. We address the questions below:
> > >
> > > **We don’t have access to the true state at any point.** The theorem uses $\phi^\star$ in order to state our mathematical guarantees for the method. However, *the method itself never has access to true states or the true decoder $\phi^\star$ at any point*. The term $\phi^\star(x) = s$ is an event that states that the endogenous state of a given observation $x$ is $s$.
> > >
> > > The reason we need the bijective mapping $\alpha$ is because a good representation only needs to map observations emitted from different endogenous states into different clusters/abstract states. **The exact label that the representation assigns to each abstract state is irrelevant.** E.g., the underlying state labels can be $s_1=0, s_2=1$, but the learned representation can label it as $s_1=1, s_2=0$. In fact, there is no information available to predict the exact labels of the states, and therefore, the best any representation can do is to recover the states up to some relabeling. This is what $\alpha$ describes in the theorem. Using bijective mappings for evaluating representations is standard in the literature, for example, please see Section 4.2.2 in [https://proceedings.mlr.press/v70/hu17b/hu17b.pdf](https://proceedings.mlr.press/v70/hu17b/hu17b.pdf).
> > >
> > > Informally, Theorem 1 statement says that there exists a bijective mapping $\alpha$ such that for any endogenous state $s$, if one samples an observation $x$ from this state $s$, then with high probability the learned decoder $\hat{\phi}$ will map $x$ to $\alpha(s)$ which is a relabeling of the true state $s$. As discussed above, **mapping to a relabeling of the true state is the best a decoder can do, and the theorem shows that the probability of this happening is very high.**
> > >
> > > Finally, we emphasize that we need terms such as $\alpha$ and $\phi^\star$ in Theorem 1 statement only to state the mathematical guarantees for the method. **The method itself neither knows this unknown mapping $\alpha$ and nor does it have access to the true decoder $\phi^\star$ or true state $s$.**
> > >
> > > We will be happy to provide any further clarifications and answer questions.

---

### Official Review · Reviewer_tXAJ · 2023-10-31

**Soundness:** 3 good
**Presentation:** 3 good
**Contribution:** 3 good
**Rating:** 8
**Confidence:** 3

**Summary:**

This paper studies representation learning from videos in the context of reinforcement learning. In particular, this work focuses on representation learning in the presence of noice, either iid or exogenous. The theoretical results show that while the current methods should be able to work well with iid noise, the agent may need exponentially more samples when exogenous noise is present.
The experiments are conducted on GridWorld and VizDoom, and show that existing representation learning methods such as ACRO [1], temporal contrastive learning, VQ-VAE can learn with iid noise.

[1] Agent-Controller Representations: Principled Offline RL with Rich Exogenous Information, Islam et al, https://arxiv.org/abs/2211.00164

**Strengths:**

- The work presents thorough theoretical analysis of representation learning from videos with iid and exogenous noise and arrives at an interesting conclusion
- The experimental results shed light on how temporal contrastive method performs compared to models that output images

**Weaknesses:**

The experiments feel a little bit detached from the theoretical results: the are no experiments with iid noise, and the results with exogenous noise seem to mainly point to the fact that some representation methods are better than others, not that exogenous noise breaks everything. Only in Figure 6 do we see exogenous noise breaking forward modeling, while ACRO still works.

I'm willing to raise my score if the authors present results with iid noise or justify the absence these results.

Suggestions and comments:
Section 4.2: "remembering all of them can easily overcome the network’s capacity focusing on the agent’s state can better help the future predictions." reads weird.
Page 2: "probne" should be "prone".
Above Equation 2, given $(x^{(i)}, k^{(i)}$ is missing a parenthesis
Assumption 1: noisy-free should be noise-free
Justification for Assumption 3: missing parenthesis in P_for

Suggested additional related work:
1. INFOrmation Prioritization through EmPOWERment in Visual Model-Based RL, Bharadhwaj et al, https://arxiv.org/pdf/2204.08585.pdf
2. Joint Embedding Predictive Architectures Focus on Slow Features, Sobal et al, https://arxiv.org/abs/2211.10831
3. Learning Invariant Representations for Reinforcement Learning without Reconstruction, Zhang et al, https://arxiv.org/abs/2006.10742

**Questions:**

Can authors explain the connection between the theoretical part and experiments better? What do the results say in relation to the theoretical conclusions?

In Figure 4, the results seem to show that forward modeling and VAE are actually able to handle the exogenous noise. This is contrary to the theoretical result, is that right?
Only in Figure 6 do we see that indeed when noise is strong enough the forward modeling objective fails.

---

> ### Author Response · Authors · 2023-11-22
> **New Experiments with IID Noise and Highlighting failure with exogenous noise. Major Clarification**
>
> We thank the reviewer for their constructive comments and feedback on the thoroughness of our theoretical contribution and the insightful results from our experimental analysis. We address the major concerns raised by the reviewer below:
>
> > the are no experiments with iid noise
>
> **New Experiments with IID noise:** *We have included experiments with IID noise in the Appendix in Figure 7.*
>
> > results with exogenous noise seem to mainly point to the fact that some representation methods are better than others, not that exogenous noise breaks everything. Only in Figure 6 do we see exogenous noise breaking forward modeling, while ACRO still works.
> In Figure 4, the results seem to show that forward modeling and VAE are actually able to handle the exogenous noise. This is contrary to the theoretical result, is that right? Only in Figure 6 do we see that indeed when noise is strong enough the forward modeling objective fails.
>
> **Major Clarification:** We want to clarify that Theorem 2 implies that in the *worst case*, all video-based representation learning methods will be exponentially inefficient. This worst case analysis does not imply that these methods cannot succeed in any domain, especially, where the exogenous noise is mild. Therefore, results in Figure 4, where the exogenous noise is somewhat mild (e.g., the size of noisy pixel is small) **do not contradict the theory.** In comparison, the size of exogenous noise is much bigger relative to the image in Figure 2 and Figure 6, and there we see that all video based representations take a big hit in performance.
>
> In particular, *Figure 6 is a good representation of how complete failure happens as the noise increases*. When the exogenous noise is limited, i.e., pixel size of exogenous noise is 4, we can see that forward modeling— the best video-based representation method, works reasonably. However, as the size of exogenous noise increases, eventually forward modeling stops working well whereas ACRO which uses trajectory data still continues to work well. **This is consistent with Theorem 2.** Additionally, **we have added Figure 8 in the Appendix** where we also increase the number of exogenous noise variables while keeping their size fixed. In this new figure, we **observe a similar decay in performance of video representation approaches relative to ACRO.**
>
> Finally, we note that temporal contrastive learning is far more susceptible to exogenous noise than forward modeling, even though in the worst case they both fail (Theorem 2). *We have revised the paper to include a new proof of this in Appendix B.4.*
>
> > Can authors explain the connection between the theoretical part and experiments better? What do the results say in relation to the theoretical conclusions?
>
> **Connection Between Theory and Experiments:** Theorem 1 and Theorem 2 are connected to our empirical results as follows. Firstly, Theorem 1 establishes that both temporal contrastive and forward modeling should work empirically when there is no noise or just IID noise. This is what Figures (2a), (4a), and (7) show as well. Secondly, Theorem 2 establishes that in the presence of exogenous noise and in *the worst case* we will expect an exponential gap between video-based representation learning and trajectory-based representation learning. We notice this most prominently in Figure 6 as discussed above. However, even in Figure 2d, we see significant loss in performance in the presence of exogenous noise. In particular, we also observe that temporal contrastive learning is particularly broken (in both Figure 2d and 4d) by even small amount of exogenous noise. Lastly, we have **included a new proof in Appendix B.4 which explains why temporal contrastive approach is much more susceptible to exogenous noise** than forward modeling, although they both fail when there is sufficient exogenous noise.
>
> **Writing and Related work:** Thanks for pointing out the typos and related work. *We have addressed these in the revision. Specifically, the InfoPower paper has also been added to the introduction.*

---

> ### Comment · Reviewer_tXAJ · 2023-11-22
> **Response**
>
> Thank you for addressing my comments running iid experiments! I raise my score to 8. I overall think this is a good paper, but I still feel the experiments section could be made more clear. Maybe one possible change would be adding a list of questions/hypotheses the experiments aim to investigate at the beginning of section 5 (this is a suggestion for camera ready).

---

> > ### Author Response · Authors · 2023-11-22
> > **Thank you**
> >
> > Thank you for the quick reply and suggestions. We will add a list of questions/hypotheses that the experiments aim to investigate at the beginning of section 5 in the camera ready.

---

### Official Review · Reviewer_sJQE · 2023-11-01

**Soundness:** 3 good
**Presentation:** 3 good
**Contribution:** 3 good
**Rating:** 8
**Confidence:** 3

**Summary:**

The paper introduces theoretical analysis for pre-trained representation learning using video data and focuses on two settings: where there is iid noise in the observation and where there is also exogenous noise in the observations.

More specifically the paper investigates three methods for video pre-training - autoencoding, temporal contrastive learning, and forward modeling, and introduces two main theorems. The first theorem provides an upper bound for the setting where there is only iid noise, and the second provides a lower bond when the observations also include exogenous noises. The first theorem leads to the conclusion that learning a representation from videos is provably correct when there is no exogenous noise, while the second means that learning is exponentially hard when there is exogenous noise (in contrast to learning from trajectory data, where the corresponding actions are available). The proofs were provided for temporal contrastive learning, and forward modeling, while evaluation and comparison to learning form trajectory data (ACRO) are provided for all three learning procedures (vector quantized variational autoencoder, temporal contrastive learning, and forward modeling).

**Strengths:**

This work introduces, for the first time, theoretical analysis and justification for pre-trained representation learning of policies from video data (under certain assumptions). In addition, the paper validates the theoretical analysis in practice, by experimenting on two challenging visual domains (GridWorld and ViZDoom).

The paper is well organized and clear to read and understand.

**Weaknesses:**

Although tested empirically, the paper does not provide a theoretical analysis for autoencoder-based approaches. Adding this analysis would make this work more complete.
In addition, the observation that temporal contrastive representation fails in the presence of exogenous noise is only empirical, justified with intuition, and lacks a more formal poof.

The analysis is restricted to training a fixed representation using only video data, without any fine-tuning stage of the learned representation. It would be nice to see an analysis of the common scenario of the fine-tuning stage.

The evaluation for iid noise with varying strength is missing (evaluation similar to Figure 6 but with iid noise). This evaluation is important for reliably comparing the performance of the setting with iid noise to that with exogenous noise.

**Questions:**

I would like to ask the following questions:

1. Why are encoder-based approaches harder to analyze?

2. If Assumption 3 (Margin Assumption) holds also for the exogenous noise in addition to the endogenous states, would learning from video data still be exponentially worse than learning from trajectory data? If the answer is yes, it means that learning a representation from videos is provably correct for cases where the margin assumption holds for all the transitions in the data.

3. In addition to the intuition, is it possible to prove the observation that temporal contrastive representation falls short in the presence of exogenous noise, compared to the forward model?

---

> ### Author Response · Authors · 2023-11-22
> **New experiments for IID setting and New proof showing why temporal contrastive is very susceptible to exogenous noise.**
>
> We thank the reviewer for their feedback. We have revised the paper to add a new proof and additional experiments. We address the main questions raised in the review below:
>
> > If Assumption 3 (Margin Assumption) holds also for the exogenous noise in addition to the endogenous states, would learning from video data still be exponentially worse than learning from trajectory data? If the answer is yes, it means that learning a representation from videos is provably correct for cases where the margin assumption holds for all the transitions in the data.
>
> **Important Clarification about Margin Assumption:** We want to clarify that even if Assumption 1 (data collection) and Assumption 3 (margin) holds in the presence of exogenous noise, **we won’t get an efficient bound due to challenges with Assumption 2 (realizability)**. In fact, Theorem 2 statement says that the lower bound MDP satisfies Assumption 3 which is the margin assumption.
>
> To understand this, let’s take the example of the Forward Modeling algorithm. Given a current observation x, it predicts the next observation x’. In Block MDP setting where there is no exogenous noise, we can write $P(x’ \mid x) = P(x’ \mid \phi^\star(x))$ where $\phi^\star(x)$ is the endogenous state decoder that maps an observation to one of the $|S|$ states. This means, we can model this conditional distribution with a model class $f(x’ \mid \phi(x))$ where the decoder $\phi$ can take $|S|$ or more values. However, when there is also exogenous noise, we cannot write $P(x’ \mid x)$ as $P(x’ | \phi(x))$ where $\phi$ only takes $O(|S|)$ many values. In fact, we would need the decoder $\phi$ to take $N$ values where $N$ linearly scales with the space of exogenous noise which can be exponentially large. Hence, if we try to satisfy realizability then we will need the output size $N$ of the decoder $\phi$ to scale with the size of exogenous noise which leads to poor sample complexity for RL. Alternatively, if we use a decoder with a small output capacity, then we cannot satisfy realizability which results in failure to accurately learn the distribution $P(x' \mid x)$. Similar failures will happen for other methods.
>
>
> > In addition to the intuition, is it possible to prove the observation that temporal contrastive representation falls short in the presence of exogenous noise, compared to the forward model?
>
> **Proof of why temporal contrastive is very susceptible to exogenous noise:** This is an interesting question. Firstly, please note that Theorem 2 establishes that _any algorithm_ including temporal contrastive and forward modeling, is exponentially inefficient in the worst case when there is exogenous noise. However, one can ask about what happens in an average case setting? We argue that in general, temporal contrastive is far more susceptible to exogenous noise than forward modeling as shown by our empirical results.
>
> **We demonstrate this with a new proof that is provided in subsection B.4 in Appendix.** We provide a brief sketch here and request the reviewer to look at the revised paper for details. We construct an MDP with a $(m+2)$-bit observation whose $l+1$ bits are generated by the exogenous noise and remaining by endogenous state. As $l$ increases, the exogenous noise becomes more dominant. We prove that for any value of $l$, temporal contrastive cannot distinguish between the correct decoder $\phi^\star$ and a bad decoder $\phi^\star_\xi$ that maps to the exogenous noise. In contrast, forward modeling correctly prefers the correct decoder over the bad decoders if $l < m/2$, i.e, when noise is limited. However, in the presence of sufficient exogenous noise ($l >= m/2$), eventually both methods will fail, consistent with the worst case scenario described in Theorem 2.
>
> > The evaluation for iid noise with varying strength is missing
>
> **IID Experiments:** We have **included results with IID noise in Figure  7 in the Appendix.**
>
> > “Although tested empirically, the paper does not provide a theoretical analysis for autoencoder-based approaches....
> > The analysis is restricted to training a fixed representation using only video data, without any fine-tuning stage of the learned representation. It would be nice to see an analysis of the common scenario of the fine-tuning stage.
>
> **Autoencoder and fine-tuning representation**: To our knowledge, there is no finite-sample result for autoencoder and it is an open question of how to do so. Empirically, we observe the performance of autoencoder is unpredictable. Fine-tuning representation is also an interesting direction to extend this work, but will require more assumptions on the setup to be made such as the choice of fine-tuning algorithm and optimization routine before doing analysis. In contrast, Theorem 1 applies irrespective of the choice of downstream PAC RL method. We leave these interesting questions for future work.

---

> > ### Comment · Reviewer_sJQE · 2023-11-23
> >
> > Thank you for revising the paper with additional experiments and proofs. I recognize your effort to address my concerns. The additional experiments definitely strengthen the contribution of the paper, and I especially appreciate the effort to provide proof as to why temporal contrastive learning is more susceptible to exogenous noise.
> >
> > Since the authors basically answered most of my concerns, I would like to raise my score to 8.

---

### Author Response · Authors · 2023-11-22
**General Response: Paper Revised, New Experiments and Additional proof and Ablations**

We would like to thank the reviewers for their time and helpful feedback. In light of this discussion, we have revised the paper and made the following changes:

**Additional Results with IID Noise:** We have included experiments with IID noise in Figure 7.

**Additional Results on a New Environment:** We have included experiments on a new Vizdoom environment called Defend the Center (see Figure 9).

**New proof showing that temporal contrastive learning is especially susceptible to exogenous noise:** A common observation in all experiments is that temporal contrastive learning fails in the presence of even small amounts of exogenous noise. In contrast, forward modeling is more robust to exogenous noise, although as predicted by Theorem 2, both approaches fail when there is sufficient exogenous noise (e.g., see Figure 6 and Figure 8). We have provided a new proof that explains this behavior in Appendix B.4.

**Writing Improvement:** We have also made writing improvements and fixed typos that were mentioned in the reviews.

All important edits are in red color. The above empirical results further buttress our findings.

We have also addressed the concerns raised by each reviewer below. We are happy to answer any further questions.

---

### Meta-Review · Area_Chair_Ks49 · 2023-12-09

**Metareview:**

This paper studies the problem of representation learning from videos for downstream reinforcement learning tasks. In particular, the authors focus on the problem of learning the latent state of the MDP given only access to video data. The authors perform a theoretical analysis of various algorithms for learning in this setting, including autoencoding, temporal contrastive learning, and forward modeling, establishing upper and lower bounds in various settings. They also perform experiments on these representation learning methods to support their theoretical findings.

The reviewers generally agreed that the problem being studied was an important one, and that the theoretical insights identified interesting conclusions. Learning from videos has proven to be an effective strategy for scaling reinforcement learning to real-world tasks since large amounts of offline video data are often available. The authors' theoretical analysis suggest that issues such as exogenous noise can significantly impede learning in these settings, which can be helpful for future work in this direction. Finally, there were some concerns about the clarity of the exposition, including use of undefined terminology such as "endogenous state", which the authors should address.

**Justification For Why Not Higher Score:**

The paper has some clarity issues that remain to be addressed.

**Justification For Why Not Lower Score:**

The paper is studying an important problem and proposes an interesting theoretical analysis.

---

### Decision · Program_Chairs · 2024-01-16

Accept (spotlight)